# Waves in SKRIPS: WAVEWATCH III coupling implementation and a case study of cyclone Mekunu

Rui Sun[1], Alison Cobb[1], Ana B. Villas Bôas[2], Sabique Langodan[3], Aneesh C. Subramanian[4], Matthew R. Mazloff[1], Bruce D. Cornuelle[1], Arthur J. Miller[1], Raju Pathak[3], and Ibrahim Hoteit[3]

[1]Scripps Institution of Oceanography, California, USA
[2]Department of Geophysics, Colorado School of Mines, Colorado, USA
[3]Physical Sciences and Engineering Division, King Abdullah University of Science and Technology (KAUST), Thuwal, Saudi Arabia
[4]Department of Atmospheric and Oceanic Sciences, University of Colorado Boulder, Colorado, USA

**Correspondence:** Rui Sun (rus043@ucsd.edu)

**Abstract.** In this work, we integrated the WAVEWATCH III model into the regional coupled model SKRIPS (Scripps–KAUST Regional Integrated Prediction System). The WAVEWATCH III model is implemented with flexibility, meaning the coupled system can run with or without the wave component. In our implementations, we considered the effect of Stokes drift, Langmuir turbulence, sea surface roughness, and wave-induced momentum fluxes. To demonstrate the impact of coupling we performed a case study using a series of coupled and uncoupled simulations of tropical cyclone Mekunu, which occurred in the Arabian Sea in May 2018. We examined the model skill in these simulations and further investigated the impact of Langmuir turbulence in the coupled system. Because of the chaotic nature of the atmosphere, we ran an ensemble of 20 members for each coupled and uncoupled experiment. We found that the characteristics of the tropical cyclone are not significantly different due to the effect of surface waves when using different parameterizations, but the coupled models better capture the minimum pressure and maximum wind speed compared with the benchmark stand-alone WRF model. Moreover, in the region of the cold wake, when Langmuir turbulence is considered in the coupled system, the sea surface temperature is about 0.5°C colder and the mixed layer is about 20 meters deeper. This indicates the ocean model is sensitive to the parameterization of Langmuir turbulence in the coupled simulations.

## 1 Introduction

Ocean surface waves play a key role in mediating exchanges of momentum, heat, and gases across the air–sea boundary (Fan et al., 2009; D'Asaro et al., 2014). The importance of surface waves in mediating air–sea interactions has been studied for decades (Fairall et al., 2003; Chen et al., 2007b). Surface waves can enhance upper ocean mixing through Langmuir turbulence, and neglecting the Langmuir mixing process may contribute to a shallow bias in mixed layer depth (MLD) (Li et al., 2016). In addition, waves determine the sea surface roughness, which affects wind stress that is important for short-term forecasting of tropical and sub-tropical cyclones (Olabarrieta et al., 2012).

Several regional coupled model studies have considered the effect of waves in air–sea interactions (e.g., Chen et al., 2007a; Warner et al., 2010; Liu et al., 2011; Wu et al., 2019a; Lewis et al., 2019; Sauvage et al., 2022). Because of the importance of air–sea heat fluxes on the energy budget of a tropical cyclone (TC) (Emanuel, 1991), many of these studies have focused on TCs and demonstrated increased accuracy in simulated intensity of TCs when coupled (e.g., Bender and Ginis, 2000; Chen et al., 2007b; Warner et al., 2010; Wu et al., 2019a; Lewis et al., 2019; Li et al., 2022). Studies have also shown a strong coupled feedback in conditions where the heat content of the upper ocean layer is low, and a weak feedback when the ocean has a thick mixed layer (Mogensen et al., 2017). Saxby et al. (2021) highlight outstanding challenges, with high-resolution convection-permitting atmosphere-only and coupled configurations both accurately simulating TCs in the Bay of Bengal, suggesting that many of the deficiencies originate in the atmospheric model, but improvements could also be gained by coupling to a wave model.

The sea state is highly complex and variable in TC conditions, with Langmuir turbulence playing an important role in the upper-ocean mixing (Rabe et al., 2015; Reichl et al., 2016a, b). This turbulence is associated with coherent Langmuir circulation structures that exist and evolve over a range of spatial and temporal scales in the surface ocean (Langmuir, 1938; McWilliams et al., 1997; Thorpe, 2004). These structures arise through an interaction between ocean surface waves and the background Eulerian current. Langmuir turbulence enhances turbulent entrainment, deepening the mixed layer and leading to sea surface cooling, which in turn affects the air–sea heat fluxes that modulate the development of TCs. Studies of idealized TCs suggest including Langmuir turbulence in model simulations may cool the sea surface temperature (SST) by 0.5-0.7°C and increase the mixed layer depth by up to 20 m (Reichl et al., 2016b; Blair et al., 2017).

Because of the importance of ocean surface waves on air–sea interaction, we implemented a regional coupled ocean–wave–atmosphere model, with the capability of investigating the impact of surface waves on air–sea interaction. The goal of this work is twofold. First, we demonstrate the integration of the wave model WAVEWATCH III to the Scripps–KAUST Regional Integrated Prediction System (SKRIPS, Sun et al., 2019), which is a regional coupled ocean–atmosphere model that has been used to investigate extreme heat wave events on the shore of the Red Sea (Sun et al., 2019), North Pacific atmospheric rivers (Sun et al., 2021), and sea-ice evolution in the Southern Ocean (Cerovečki et al., 2022). The second goal is to evaluate the implementations of ocean surface waves in the coupled system, especially for Langmuir turbulence that alleviates the model bias (Li et al., 2016; Li and Fox-Kemper, 2017). The coupled model is also sensitive to the parameterization of Langmuir turbulence because it increases ocean mixing and cools down the SST during the simulation. Here, we perform a series of coupled and uncoupled numerical simulations of tropical cyclone Mekunu in the Arabian Sea. Because of the chaotic nature of the atmosphere, we ran an ensemble of 20 members for each coupled and uncoupled experiment. The Arabian Sea is investigated in this work because of its rich and diverse ecosystem, its economic impact on the surrounding countries, and its important role in international trade. Continued climate warming is expected to further amplify the risk of cyclones in the Arabian Sea (Dube et al., 1997; Evan et al., 2011; Evan and Camargo, 2011) and increase socio-economic implications for coastal communities in that region (Henderson-Sellers et al., 1998; Murakami et al., 2017; Bhatia et al., 2018). We investigated Cyclone Mekunu because it was the strongest tropical cyclone in the north Indian Ocean in 2018 (Government of India, 2018). It had a clear signature in SST cooling and MLD deepening, which can be used for testing the model. We investigated the sensitivities of

the coupled model to the parameterizations of surface wave driven mixing to examine the effect of surface waves on air–sea interactions.

The rest of this paper is organized as follows. The implementation of the coupled model is described in Section 2. An overview of cyclone Mekunu, the design of the experiments, and the validation data are presented in Section 3. Section 4 details the numerical simulation results and Section 5 discusses the sensitivity of the simulation results to parameterizing Langmuir turbulence based on the evolving wave state. Section 6 concludes the paper with a summary of the main findings.

## 2 Methodology

### 2.1 Model Description

In this work, the version 6.07 (Tolman, 1991; WW3DG, 2019) of the WAVE-height, WATer depth and Current Hindcasting third generation wave model (WAVEWATCH III, hereinafter, WW3) is integrated into the SKRIPS model. The SKRIPS model (Sun et al., 2019) is a regional coupled ocean–atmosphere model: the oceanic model component is the MIT general circulation model (MITgcm) (Marshall et al., 1997; Campin et al., 2019) and the atmospheric model component is the Weather Research and Forecasting (WRF) model (Skamarock et al., 2019). The Earth System Modeling Framework (ESMF) (Hill et al., 2004) is used as the coupler to drive the coupled simulation. The National United Operational Prediction Capability (NUOPC) layer in the ESMF is used to simplify the implementations of component synchronization, execution, and other common tasks in the coupling (Hill et al., 2004).

The schematic description of the coupled model is shown in Fig. 1. In the coupling process, all model components send data to ESMF: MITgcm sends SST and ocean surface velocity; WRF sends surface atmosphere fields, including (1) net surface longwave and shortwave radiative fluxes, (2) surface latent and sensible heat fluxes, (3) 10-m wind speed, (4) precipitation, and (5) evaporation; WW3 sends the wave variables to ESMF, including the (1) bulk wave parameters (i.e., significant wave height, peak wavelength and mean wavenumber), (2) surface Stokes drift, (3) Langmuir turbulence parameters (i.e., Langmuir number and enhancement factor), and (4) momentum flux terms due to surface waves. Then all model components read the data they need from ESMF: MITgcm reads surface atmospheric variables and wave variables; WRF reads SST, ocean surface velocity, and wave variables; WW3 reads wind speed and surface current velocity. The surface current velocity sent to WRF and WW3 is consistent in our model, using the current velocity in the first layer of MITgcm. We used the surface current based on previous literature (Warner et al., 2008, 2010; Couvelard et al., 2020), but this may overestimate the strength of surface currents impacting the wave model, as suggested by Fan et al. (2009), who used the current velocity at $L/4\pi$ ($L$ is the mean wavelength). The wind speed sent to WW3 and MITgcm is the relative 10-m wind speed from WRF based on the Monin-Obukhov similarity theory (Monin and Obukhov, 1954; Renault et al., 2020), then WW3 and MITgcm use the relative 10-m wind speed without correcting the current velocity in the simulations.

Similar to our previous work Sun et al. (2019), the MITgcm model uses the surface atmospheric variables received from ESMF to prescribe surface forcing, including (1) total net surface heat flux, (2) surface wind stress, and (3) freshwater flux. The total net surface heat flux is computed by adding surface latent heat flux, sensible heat flux, net shortwave radiation flux,

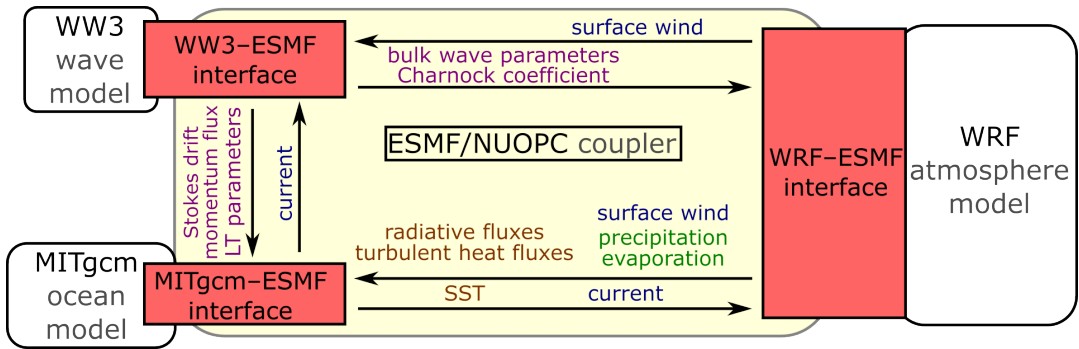

**Figure 1.** The schematic description of the SKRIPS regional coupled ocean–atmosphere model. The yellow block is the ESMF/NUOPC coupler; the white blocks are the ocean and atmosphere components; the red blocks are the implemented MITgcm–ESMF, WRF–ESMF, and WW3–ESMF interfaces.

and net longwave radiation flux. The surface latent and sensible heat fluxes are computed using the COARE 3.0 bulk algorithm in WRF (Fairall et al., 2003). The implementations of the wave effects are discussed in the latter sections. The Stokes forces, the Langmuir turbulence parameters, and the momentum fluxes are detailed in Sections 2.2, 2.3, and 2.4, respectively. The sea surface roughness parameterizations are summarized in Section 2.5.

To implement the WW3–ESMF interface, we followed our previous implementations (Sun et al., 2019) and the WW3–ESMF interface (WW3DG, 2019) for the prototype case. We separated the WW3 main program into three subroutines that handle initialization, execution, and finalization. These subroutines are used by the ESMF/NUOPC coupler that controls the wave component in the coupled run. During the simulation, WW3 receives and sends boundary fields via subroutine calls by the WW3–ESMF interface, shown in Fig. 1. In addition, WW3 grid information is provided to the coupler in the initialization subroutine. To carry out the coupled simulation on HPC (high-performance computing) clusters, the WW3–ESMF interface runs in parallel via MPI (message passing interface) communications. We have also updated the MITgcm–ESMF and WRF–ESMF interfaces by including the inputs and outputs associated with the wave model. The wave component is implemented with flexibility, meaning the coupled system can run with or without the wave component. It is noted that ESMF online re-gridding options are also implemented for the wave component when exchanging boundary fields, but it is not used in this work because we aim to present the implementation of wave components. The online re-gridding option will be used when using a higher resolution ocean model for the Arabian Sea operational model.

## 2.2 Stokes Forces in MITgcm

The contribution of surface waves to the ocean momentum balance can be described by the wave-averaged momentum equations as follows (Suzuki and Fox-Kemper, 2016; Wu et al., 2019a):

$$\frac{\partial \boldsymbol{u}}{\partial t} + (\boldsymbol{u} \cdot \nabla)\boldsymbol{u} = -\frac{1}{\rho_w}\nabla p + \boldsymbol{D}^u + \boldsymbol{b} - \boldsymbol{f} \times \boldsymbol{u} - \underbrace{(\boldsymbol{u}^S \cdot \nabla)\boldsymbol{u}}_{\text{Stokes advection}} - \underbrace{\boldsymbol{f} \times \boldsymbol{u}^S}_{\text{Stokes Coriolis}} - \underbrace{u_j^L \nabla u_j^S}_{\text{Stokes shear}}, \tag{1}$$

where $t$ is time; $\nabla = (\partial x, \partial y, \partial z)$; $\rho_w$ is the density of water; $p$ is the pressure; $\boldsymbol{b} = -g\rho/\rho_w \hat{\boldsymbol{z}} = b\hat{\boldsymbol{z}}$ is the buoyancy term; $\hat{\boldsymbol{z}}$ is the vertical unit vector; $\boldsymbol{D}^u$ is the diffusion; $\boldsymbol{f} = f\hat{\boldsymbol{z}}$ is the Coriolis parameter; $-u_j^L \nabla u_j^S$ is the Stokes shear force; $\boldsymbol{u} = (u_1, u_2, u_3) = (u, v, w)$ is the wave-filtered Eulerian velocity (Eulerian velocity of the flow solved in MITgcm); $\boldsymbol{u}^S$ is the Stokes drift; $\boldsymbol{u}^L = \boldsymbol{u} + \boldsymbol{u}^S$ is the wave-filtered Lagrangian velocity. Here, the Einstein summation convention is used (e.g, $u_j^L \nabla u_j^S = u_1^L \nabla u_1^S + u_2^L \nabla u_2^S + u_3^L \nabla u_3^S$), although vector notation is used when it is unambiguous.

The tracer advection equation can be written as (Suzuki and Fox-Kemper, 2016; Wu et al., 2019a):

$$\frac{\partial c}{\partial t} + (\boldsymbol{u} \cdot \nabla)c = -\underbrace{(\boldsymbol{u}^S \cdot \nabla)c}_{\substack{\text{Stokes} \\ \text{advection}}} + D^c, \tag{2}$$

where $c$ is a scalar quantity, such as potential temperature and salinity; $D^c$ is the diffusion.

The Stokes-Advection and Stokes-Coriolis terms are implemented in MITgcm by modifying the source term of the governing equations. The profiles of Stokes velocity are determined based on Breivik et al. (2014). Considering the effect of Langmuir turbulence, the Stokes shear term in Eq. (1) is parameterized according to the literature (Li et al., 2016; Li and Fox-Kemper, 2017; Li et al., 2017), detailed in Section 2.3. It is noted that our implementations and tests aim to demonstrate the impact of Langmuir turbulence on the ocean, and thus the divergence of the Stokes drift is not considered in our governing equations as discussed in Wu et al. (2019a, b). There are also other options to better approximate the Stokes velocity profiles (e.g. Breivik et al., 2016; Romero et al., 2021) that remain to be tested in future work.

## 2.3 Parameterization of Langmuir Turbulence

Considering the impact of the surface waves, the Stokes drift provides a source of the turbulent kinetic energy (TKE) through the vortex force and modified pressure (Craik and Leibovich, 1976), or more cleanly the Stokes shear force (Suzuki and Fox-Kemper, 2016; Li and Fox-Kemper, 2017) as mentioned in Eq. (1). Evidence of this enhanced vertical mixing has been documented from observations and large-eddy simulations (McWilliams et al., 1997; D'Asaro, 2001; Van Roekel et al., 2012; D'Asaro et al., 2014). In this work, we aim to implement the Stokes shear force in the coupled model and investigate its effect on the coupled system. Although it is not explicitly accounted for in KPP (K-profile parameterization) (Large et al., 1994), KPP might have implicitly incorporated some effects by tuning the parameters to ocean observations (Reichl et al., 2016b). In addition, our implemented model is about 8 km resolution (0.075°), and the horizontal gradients of the Stokes drift are several orders of magnitude smaller than vertical gradients. Following Suzuki and Fox-Kemper (2016), we only consider the effects of Stokes shear force due to Langmuir turbulence because of this scale separation.

Although there are many unknowns about the role of Langmuir mixing in ocean modeling, there are exists many parameterizations that aim to represent these processes and alleviate model bias. Within the KPP scheme, we implemented three Langmuir turbulence parameterizations (Van Roekel et al., 2012; Li et al., 2016; Li and Fox-Kemper, 2017; Li et al., 2017): (1) VR12-MA; (2) LF17; (3) LF17-ST. Both VR12-MA and LF17 parameterize the Langmuir turbulence based on the parameters computed from WW3: in VR12-MA the KPP turbulent velocity scale is multiplied by an enhancement factor; in LF17 the KPP turbulent velocity scale is treated in the same way as VR12-MA, and the entrainment buoyancy flux is also considered.

On the other hand, LF17-ST parameterizes the Langmuir turbulence similarly to LF17, but parameters are computed using the 10-m winds instead of using the output from WW3. VR12-MA and LF17 are implemented because they are used in a variety of case studies and substantially improve the shallow biases of mixed layer depth (Li et al., 2016; Li and Fox-Kemper, 2017; Li et al., 2019). We aim to compare the performance of VR12-MA and LF17 to demonstrate the impact of entrainment on the simulations. We also used the well-validated LF17-ST implementation by Schultz et al. (2020) to validate LF17 in the coupled simulations due to the similarity of these two options. Because LF17-ST does not need bulk wave parameters as input, it can be also used in uncoupled MITgcm simulations (Schultz et al., 2020) or coupled simulations without waves to parameterize the Langmuir turbulence.

For all the implemented schemes, the turbulent velocity scale $w_s$ is modified by multiplying with an enhancement factor $\varepsilon$ as:

$$w_s = \frac{\kappa u^*}{\phi} \varepsilon \tag{3}$$

where $\kappa$ is the von Karman constant; $u^*$ is the friction velocity; $\phi$ is the stability function defined by Large et al. (1994); the enhancement factor is defined as:

$$\varepsilon = |\cos(\alpha)| \sqrt{1 + (1.5 \mathrm{La}_{\mathrm{SLP}})^{-2} + (5.4 \mathrm{La}_{\mathrm{SLP}})^{-2}}, \tag{4}$$

where $\alpha$ is the angle between wind and Langmuir cells; $\mathrm{La}_{\mathrm{SLP}}$ is the Langmuir number (Van Roekel et al., 2012; Li et al., 2016, 2019). Here we used the projected Langmuir number defined in Eq. (6) of Li et al. (2019) based on the surface averaged Stokes velocity. More details of the projected Langmuir number can be found in Appendix A.

The enhanced turbulent velocity scale affects the vertical viscosity, tracer diffusivity, and nonlocal flux in KPP. In particular, the KPP eddy diffusivity profile $\kappa_v$ is:

$$\kappa_v = w_s h G(\sigma), \tag{5}$$

where $h$ is the boundary layer depth; $h$ is the KPP boundary layer depth; $G(\sigma)$ is the shape function, and $\sigma = -z/h$ is the normalized depth. The boundary layer depth is determined based on the bulk Richardson number:

$$Ri_b(z) = \frac{(B_r - B(z))|z|}{|\boldsymbol{u}_r - \boldsymbol{u}(z)|^2 + V_t^2(z)}, \tag{6}$$

where $B_r$ and $\boldsymbol{u}_r$ are the buoyancy and velocity averaged in the surface layer; $B(z)$ and $\boldsymbol{u}(z)$ are the local buoyancy and local velocity; $V_t$ is the unresolved vertical shear proportional to $w_s$ (Large et al., 1994; Li et al., 2016). The KPP boundary layer depth is defined at the smallest KPP boundary layer depth $h$ that reaches the critical bulk Richardson number $Ri_{cr} = 0.3$.

Different from VR12-MA, LF17 parameterized the entrainment flux due to Langmuir turbulence by revising the bulk Richardson number:

$$Ri_b(z) = \frac{(B_r - B(z))|z|}{|\boldsymbol{u}_r - \boldsymbol{u}(z)|^2 + V_{tL}^2(z)}, \tag{7}$$

the turbulence shear term $V_{tL}^2$ is defined as:

$$V_{tL}^2(z) = \frac{C_v N(z) w_s(z)|z|}{Ri_c} \left[ \frac{0.15w_*^3 + 0.17u_*^3(1+0.49\text{La}_{\text{SL}}^{-2})}{w_s(z)^3} \right]^{\frac{1}{2}}, \tag{8}$$

where dimensionless coefficient $C_v = \max(2.1 - 200 \times \max(0, N), 1.7)$; $N(z)$ is the local buoyancy frequency; $Ri_c$ is the critical Richardson number; $w_* = (-B_0/h)^{1/3}$ is the convective velocity scale; $B_0$ is the surface buoyancy flux. Here $\text{La}_{\text{SL}}$ is the Langmuir number defined in Eq. (5) of Li et al. (2019) and is detailed in Appendix A.

In LF17-ST, the enhancement coefficient and entrainment flux are calculated similarly to LF17, but the Langmuir turbulence coefficient La is determined by the Stokes velocity parameterized from 10-m wind and mixed layer depth. The details can be found in Eq. (25) in Li et al. (2017). In this work, the implementations of LF17-ST in MITgcm followed the code provided by Schultz et al. (2020).

## 2.4   Momentum Flux in MITgcm

The surface boundary condition is also modified by waves. The surface wind stress $\tau_a$ is modified by subtracting the part that goes into wave growth $\tau_{aw}$ and adding the wave-to-ocean momentum flux due to wave breaking $\tau_{ow}$. Hence the momentum flux in MITgcm is (Jenkins, 1989; Weber et al., 2006; Janssen, 2012):

$$\tau_{oc} = \tau_a - \tau_{aw} + \tau_{ow}, \tag{9}$$

where

$$\tau_{aw} = \rho_w g \int_0^{2\pi} \int_0^\infty \frac{k}{\omega} S_{in} \,\mathrm{d}\omega d\theta, \tag{10}$$

and

$$\tau_{ow} = \rho_w g \int_0^{2\pi} \int_0^\infty \frac{k}{\omega} S_{ds} \,\mathrm{d}\omega d\theta. \tag{11}$$

Here, $S_{in}$ and $S_{ds}$ are the wind input and wave dissipation source terms, respectively; $k$ is the wave number; $\omega$ is the angular wave frequency; $\theta$ is the wave direction. In the coupled model, $\tau_a$ is calculated in MITgcm (Large and Yeager, 2004) because

WRF does not directly output the momentum flux terms. The parts that go into wave growth $\tau_{aw}$ and wave breaking $\tau_{ow}$ are calculated in WW3 (Tolman, 1995; Arduin et al., 2003).

## 2.5   Ocean Roughness Closures

The parameterization of ocean roughness in WRF is also important. When WRF is not coupled to WW3, the bottom roughness length $z_0$ is computed with the formulation proposed by Smith (1988), which is a combination of the formulae described by Liu

et al. (1979) and Charnock (1955). When coupled with WW3 we parameterize the surface roughness based on the Charnock coefficient calculated from WW3 to make the surface roughness consistent. We have also implemented a few other ocean

roughness closure models that have been used in COAWST Olabarrieta et al. (2012): DGHQ (Drennan et al., 2003, which is based on wave age), TY2001 (Taylor and Yelland, 2001, which is based on wave steepness), and OOST (Oost et al., 2002, which considers both the effects of wave age and steepness). These models parameterize $z_0$ using the bulk wave parameters from WW3. We implemented these options in the WRF Mellor–Yamada–Nakanishi–Niino (MYNN) surface layer scheme. More detailed descriptions of these closure models and sensitivity analysis are presented in Appendices B and C.

## 3 Experimental Design

### 3.1 Overview of the event

Cyclone Mekunu formed in the southeast Arabian Sea on May 20, 2018, and then propagated northwest before making landfall in southwest Oman on May 26. Categorized as ESCS (extremely severe cyclonic storm), cyclone Mekunu was the second cyclonic storm over the Arabian Sea in 2018 and the strongest tropical cyclone in the north Indian Ocean that year. The peak maximum sustained surface wind speed was 170-180 km/h gusting to 200 km/h (95 knots) and the lowest estimated central pressure was 960 hPa on May 25th (Government of India, 2018). Salalah, the capital city of southern Oman's Dhofar province, received 278.2 millimeters (10.95 inches) of rain in just 24 hours ending around 10:30 a.m. on May 26, over double the city's average annual rainfall of about five inches, with a total of 617 millimeters of rainfall during May 23-27 (Government of India, 2018). In addition to the extremely heavy rainfall in Oman and Yemen, cyclone Mekunu caused heavy rainfall that created desert lakes over the "Empty Quarter" in Saudi Arabia. The warm, sandy, and wet soil was the perfect environment for the outbreak of desert locusts, posing a serious risk to food security and livelihoods (Salih et al., 2020).

### 3.2 Model Setups

To illustrate the coupled model capabilities, we perform the following types of model runs:

1. CPL.AOW: coupled ocean–wave–atmosphere (MITgcm–WW3–WRF) simulations.

2. CPL.AO: coupled ocean–atmosphere (MITgcm–WRF) simulations. The ocean–atmosphere model is not coupled to the wave model, aiming to demonstrate the impact of the wave model on the simulation results.

3. ATM.DYN: stand-alone atmosphere (WRF) simulations. The atmosphere model is not coupled to the wave or ocean models. The SST forcing is from the HYCOM/NCODA $1/12°$ daily global analysis data (the Global Ocean Forecast System, Version 3.0 (Chassignet et al., 2007), hereinafter, HYCOM). Compared with CPL.AO and CPL.AOW, this run serves as a benchmark that aims to demonstrate the impact of waves and coupled air–sea interactions on the simulation results.

The grid spacing and computational domain are outlined in Table 1. To generate the grids, we choose the latitude–longitude (cylindrical equidistant) map projection for MITgcm, WW3, and WRF. The model domain extends from 0 to 30.6°N and from 30°E to 78°E. The horizontal grid has 408×640 (lat×long) cells and the spacing is 0.08° in both directions. We use identical

horizontal grids for all model components to eliminate the complication of regridding winds near steep orography and complex coastlines, although the regridding capability is implemented in SKRIPS. There are 40 sigma layers in the atmosphere model (top pressure is 50 hPa) and 50 z-layers in the ocean model ($dz = 4$ m in the top). The wave model has a spectral grid of 48 directions (7.5° resolution) and 32 frequencies exponentially spaced from 0.0343 to 1.1 Hz. Because of the chaotic nature of the atmosphere, we generated 20-member ensembles for each run by adding small random perturbations to the initial SST (<0.01°C) at every grid point in the coupled model. The random perturbations are added without any spatial or temporal correlation, aiming to demonstrate the internal variability of the model.

The initial conditions, boundary conditions, and forcing terms of the simulations are also outlined in Table 1. In the coupled runs, the ocean model uses the HYCOM data as initial and boundary conditions for ocean temperature, salinity, and horizontal velocities (Chassignet et al., 2007). At each time step, the boundary conditions for the ocean are updated by linearly interpolating between the daily HYCOM data. A restoring layer with a width of 13 grid cells is applied at the lateral boundaries to enforce the boundary conditions. The inner and outer boundary relaxation timescales are 10 and 0.5 days, respectively. The atmosphere is initialized using the NCEP (National Centers for Environmental Prediction) FNL (Final) Operational Global Analysis data. The same data also provide the boundary conditions for air temperature, wind speed, and humidity. The atmospheric boundary conditions are updated based on linearly interpolating between 6-hourly NCEP FNL data. The 'specified' zone in WRF prescribes the lateral boundary values, and the 'relaxation' zone is used to nudge the solution from the domain interior toward the boundary condition value. Here we use the default width of one point for the specified zone and four points for the relaxation zone. In the wave model, the wave spectra at the offshore boundary come from the global wave modeling system described by Rascle and Ardhuin (2013). To initialize the wave model, we allowed the wave field to spin-up for 19 days from May 01, 2018 and then we analyze the period from May 20, 2018. On the other hand, we did not spin up MITgcm or WRF, trying to initialize the coupled model using the analysis data directly. This may cause an initial shock in the coupled simulation, but large initial shocks are not observed in the simulations. We performed downscaled hindcasts in this work, which allows us to focus on the impacts of air–sea interactions during the tropical cyclone event by minimizing the boundary errors.

The time step of the ocean model is 120 seconds. The horizontal sub-grid mixing is parameterized using nonlinear Smagorinsky viscosities, and the K-profile parameterization is used for vertical mixing processes (Large et al., 1994) with modifications accounting for Langmuir mixing (Van Roekel et al., 2012; Li et al., 2016; Li and Fox-Kemper, 2017; Li et al., 2017). The time step of the atmosphere model is 30 seconds. The Morrison 2-moment scheme (Morrison et al., 2009) is used to resolve the microphysics; the updated version of the Kain–Fritsch convection scheme (Kain, 2004) is used for cumulus parameterization; the Mellor–Yamada–Nakanishi–Niino 2.5-order closure scheme (Nakanishi and Niino, 2004, 2009) is used for the planetary boundary layer (PBL) and the surface layer (SL); the Rapid Radiation Transfer Model for GCMs (RRTMG; Iacono et al. (2008)) is used for longwave and shortwave radiation transfer through the atmosphere; the Noah land surface model is used for the land surface processes (Tewari et al., 2004). The selection of WRF physics schemes is the same as our previous work (Sun et al., 2021). The wave model uses a global integration time step of 600 s, spatial advection time step of 60 s, spectral advection time step of 60 s, and minimum source term time step of 10 s. In CPL.AOW and CPL.AO, the coupling interval is 120 seconds

to allow for capturing the diurnal cycle of air–sea fluxes. We output the simulation results every three hours to demonstrate the tropical cyclone evolution.

When the effects of surface waves are considered in CPL.AOW, the model setup is as follows. The Stokes-Coriolis and the Stokes-Advection in Eq. (1) are considered; the impact of Langmuir turbulence is parameterized in the same way as Li and Fox-Kemper (2017); the ocean surface roughness is parameterized using the Charnock coefficient (CHNK) from WW3; the wind stress in the ocean model is treated as mentioned in Eq. (9). We have compared the coupled model with and without wave effects in Section 4, then we further illustrate the sensitivity of the coupled model to the wave effects in Section 5 and Appendix C. It is noted that when the Stokes-Coriolis and the Stokes-Advection are not explicitly considered in the experiments, the model setups are consistent with Li et al. (2016), assuming the simulated velocity is Lagrangian.

**Table 1.** The computational domain, WRF physics schemes, initial condition, boundary condition, and forcing terms used in the present simulations.

| run | CPL.AOW and CPL.AO | ATM.DYN |
|---|---|---|
| model region | 0 to 30.6°N; 30°E to 78°E | |
| horizontal resolution | 408×640 (lat×long) | |
| grid spacing | 0.075° × 0.075° (lat×long) | |
| vertical levels | 40 (atmosphere) 50 (ocean) | 40 (atmosphere only) |
| initial and boundary conditions | NCEP FNL (atmosphere) HYCOM/NCODA (ocean) | NCEP FNL (atmosphere only) |
| ocean surface conditions | from MITgcm | HYCOM/NCODA |
| atmospheric forcings for ocean model | from WRF | not necessary |
| microphysics | Morrison 2-moment scheme | |
| convection | Kain–Fritsch scheme | |
| PBL and surface layer | Mellor–Yamada–Nakanishi–Niino 2.5-order scheme | |
| longwave radiation | Rapid Radiation Transfer Model for GCMs (RRTMG) | |
| shortwave radiation | Rapid Radiation Transfer Model for GCMs (RRTMG) | |
| land surface | Noah land surface model | |

## 3.3 Validation Data

To evaluate the performance of the simulations, the model outputs are compared with available data. The track of the tropical cyclone, the tropical central pressure, and the maximum wind speed are validated against IBTrACs data (Knapp et al., 2010, 2018). Here we use the IBTrACS–World Meteorological Organization version. IBTrACS provides a compilation of

historical TC data as recorded by meteorological centers and/or forecast agencies and in this case the data from Indian Meteo-
rological Department (IMD) are used.

The simulated SST fields are validated against the HYCOM data. Because the HYCOM data are the initial and boundary
conditions in the coupled model, this aims to show the error increase from the initial condition. We used bilinear interpolation
to map the validation data onto the model grid to compare the results in a consistent way. When interpolating SST, only the
values on ocean points are used. The SST is also validated by using in-situ observations from the satellite-tracked drifters of
the Global Drifter Program (GDP, from https://www.aoml.noaa.gov/envids/gld/index.php) (Lumpkin and Pazos, 2007).

## 4    Comparing coupled and uncoupled models

In this section, the ensemble coupled simulation results (i.e., CPL.AOW and CPL.AO) are compared with the results from the
uncoupled runs (i.e., ATM.DYN) to assess the performance of the models, the impact of coupled feedbacks, and the effect of
the waves. We compared the ensemble-averaged characteristics of the tropical cyclone (e.g., track, intensity, and wind speed),
the changes in the ocean (e.g., sea surface temperature and mixed layer depth), and the waves generated by the tropical cyclone.
By comparing the coupled run with uncoupled runs, we aim to (1) demonstrate the capability of the coupled model and (2)
illustrate the impact of including ocean–wave–atmosphere interactions on simulating this tropical cyclone event.

### 4.1    Cyclone Track, Intensity, Wind Speed

First, we examine the characteristics of cyclone Mekunu obtained from CPL.AOW, CPL.AO, and ATM.DYN to demonstrate
the capability of the coupled model. The tracks of the tropical cyclone, defined by the positions of the low pressure center, are
presented in Fig. 2, where it can be seen that all models can qualitatively match the observed evolution and track. Although the
translation speed of the tropical cyclone from CPL.AOW is somewhat slower (CPL.AOW: 245 km/day; IBTrACS: 254 km/day),
the distances between the cyclone centers for all model runs and IBTrACS data are less than 250 km until May 26, shown in
Fig. 3(a).

The characteristics of cyclone Mekunu (i.e., cyclone central pressure and maximum wind speed) obtained in the ensemble
simulations are compared quantitatively with IBTrACS data in Fig. 3(b) and Fig. 3(c). From May 22 to May 27, the root-
mean-square-errors (RMSEs) of the cyclone low pressure center are 9.53, 9.25, and 10.55 hPa for CPL.AOW, CPL.AO, and
ATM.DYN, respectively (ensemble standard deviations are 2.85, 2.67, and 2.66 hPa). The RMSEs of the maximum wind
speed are 9.06, 8.81 and 9.81 m/s for CPL.AOW, CPL.AO, and ATM.DYN (ensemble standard deviations are 3.13, 3.00,
and 2.80 m/s). In addition, the ensemble mean lowest pressures in CPL.AOW, CPL.AO, and ATM.DYN are higher than the
IBTrACS data by 4.9, 5.3, and 17.3 hPa, respectively. The overestimation in ATM.DYN is more significant than one standard
deviation between May 25 to 26. For the maximum wind speed, CPL.AOW and CPL.AO underestimate the maximum wind
speed by about 6.2 m/s and 5.9 m/s, while in ATM.DYN the underestimation is as large as 14.1 m/s. The ATM.DYN does not
capture the intensification of the TC between May 24 and 26 that is present in the IBTrACS observations and in the coupled
simulations.

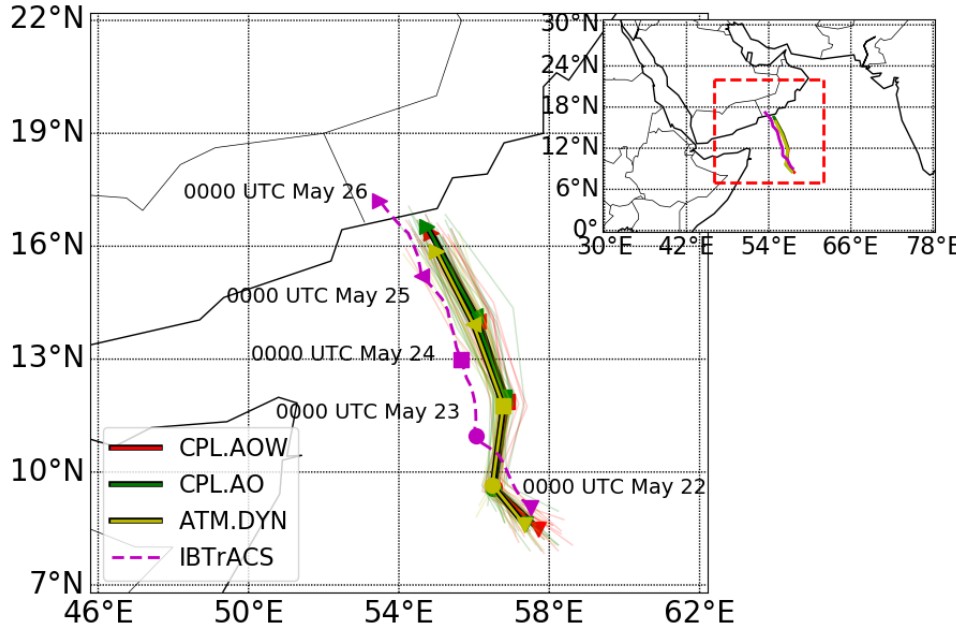

**Figure 2.** Comparison between the tracks of cyclone Mekunu obtained from IBTrACS data and the simulations. The thick solid lines indicate the tropical cyclone tracks obtained from averaging all ensemble members; the thin solid lines indicate the tropical cyclone tracks obtained from individual ensemble members. The text and markers highlight the time and locations of the cyclone at specific times.

To highlight the surface fluxes from the simulation, we show the snapshots of the wind speed and latent heat fluxes (LHFs) in Fig. 4. Instead of plotting the entire computational domain, we highlight the region around the center of the tropical cyclone (from 7°N to 22°N and from 46°E to 62°E). The 10-m wind speeds obtained in CPL.AOW and CPL.AO are generally consistent, except for the region near the center of the cyclone. Figure 4(c) shows weaker wind speed and smaller area of high wind speeds (indicated by the contour of 15 m/s) in ATM.DYN because the uncoupled run underestimates the intensity of the cyclone. In addition, we present the LHFs because they are the major component of the net surface heat fluxes and they are associated with the water vapor uptake. It can be seen in Fig. 4(d-f) that the LHFs are weak along the cyclone tracks due to the cold wake (shown in Fig. 5) but are generally consistent in CPL.AOW and CPL.AO. Near the center of the tropical cyclone in Fig. 4, the LHFs in ATM.DYN are weaker than the coupled runs by a few hundred of W/m$^2$. This is because the tropical cyclone is weaker in ATM.DYN, which will be further discussed in Section 4.2. The largest differences in cyclone track, intensity, and wind speed are between the uncoupled (ATM.DYN) and coupled (CPL.AO, CPL.AOW) simulations, with the latter being closer to IBTrACS observations.

In summary, both CPL.AOW and CPL.AO runs better simulate the tropical cyclone characteristics than ATM.DYN in comparison with the IBTrACS data. For the cyclone central pressure and wind speed, CPL.AOW is better for extreme conditions but is outperformed by CPL.AO for their RMSEs of throughout the event. CPL.AO also better simulates the track of the tropical cyclone. The ATM.DYN also underestimates the latent heat loss from the ocean compared with CPL.AOW and CPL.AO.

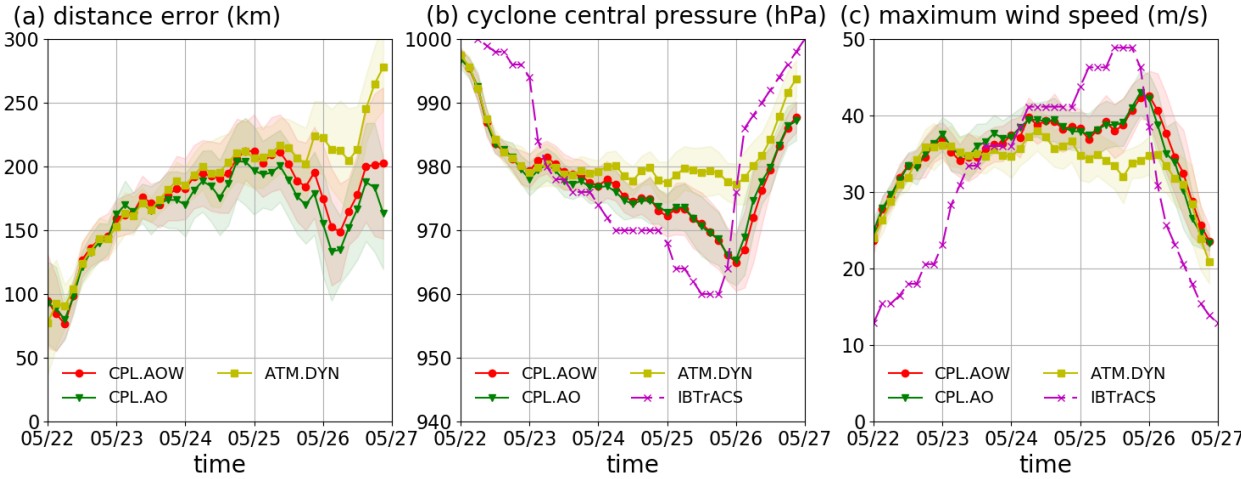

**Figure 3.** The characteristics of cyclone Mekunu obtained from the simulations plotted as functions of time. The solid lines indicate the ensemble averaged simulation results; the shaded areas indicate the standard deviation of the results. Panel (a) shows the distance errors in comparison with IBTrACS data; Panel (b) shows the cyclone central pressure; Panel (c) shows the maximum wind speed. The simulations start on May 20, but the results are not presented before the pressure starts to drop on May 22.

The differences between the tropical cyclones simulated in the coupled and uncoupled simulations are associated with the SST cooling in the simulations, which are further discussed in Section 4.2. The differences between CPL.AOW and CPL.AO are further investigated in Section 5.

## 4.2 SST and Mixed Layer Depth

To highlight the impact of the tropical cyclone on the ocean, we plot the evolution of SST and ocean MLD from CPL.AOW and ATM.DYN in Fig. 5. The results obtained from CPL.AO are not presented here, but in Section 5 we investigate the effect of wave coupling on SST and MLD in the coupled system. Figures 5(b) and 5(c) show the differences of ensemble averaged SST between May 20 and May 27 (May 27 SST minus May 20 SST). This aims to highlight the development of an SST cold wake during this event. It can be seen that the SST cools down by a maximum of 4°C along the TC track, which can impact the ocean and air–sea interactions (Price, 1981; Stramma et al., 1986; Pasquero et al., 2021). It can be seen in Fig. 5 that the SST cooling in CPL.AOW is weaker than that in HYCOM, indicating the SST is warmer throughout the simulation in CPL.AOW. Contributed by the warmer SST, the intensity of the tropical cyclone is also stronger in CPL.AOW than ATM.DYN. Due to chaotic nature of the atmosphere, it is still unknown why the warmer SST in CPL.AOW improves the simulation of tropical cyclone characteristics. In addition, although we compared the simulation results using available data, the lack of in-situ observations in this region makes it challenging to validate the SST used the simulations. It should be noted that CPL.AOW also captures the SST warming in the Arabian Gulf and the Gulf of Aden compared with HYCOM data.

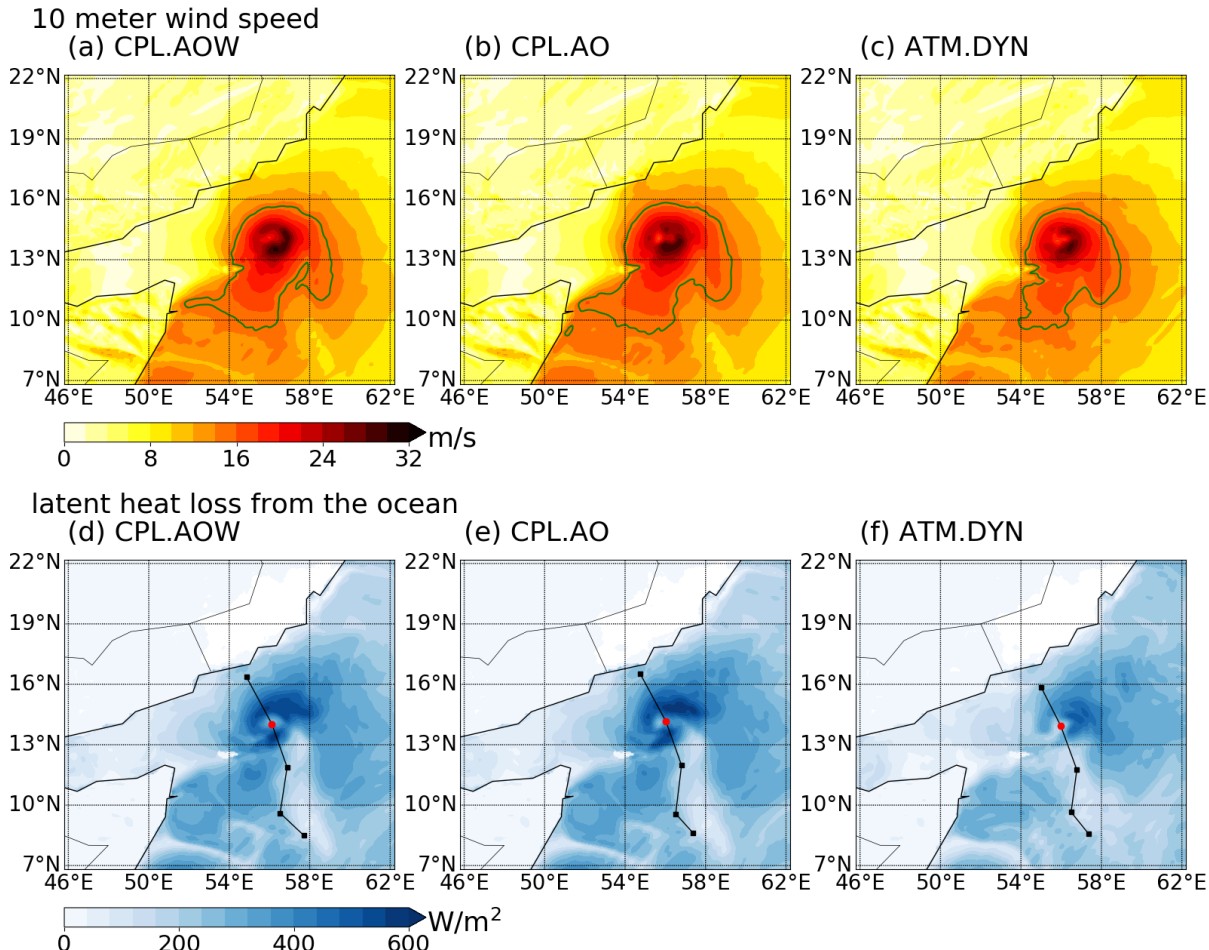

**Figure 4.** The snapshots of the wind speed and latent heat fluxes at 00 UTC May 25 obtained in the simulations. The ensemble averaged fields are plotted and we highlighted the region between 7°N to 22°N and from 46°E to 62°E. In Panels (a-c) the 15 m/s contour of wind speed is used to highlight the size of the tropical cyclone. The black solid lines indicate the ensemble-averaged track of tropical cyclone in the simulations shown in Fig. 2. The red dots indicate the ensemble-averaged locations of the center of tropical cyclones at the snapshot; the black dots indicate the ensemble-averaged locations of the center of tropical cyclones each day at 00 UTC.

The evolution of the mixed layer depth during the event is shown in Fig. 6 to demonstrate the impact of the tropical cyclone on ocean mixing. Here we are using $\Delta\rho = 0.03\ \mathrm{kg/m^3}$ to define the MLD. The initial MLD is about 30-40 meters along the track of the tropical cyclone. To highlight the evolution of the mixed layer, the differences of ensemble averaged MLD between May 20 and May 27 (May 27 MLD minus May 20 MLD) are plotted in Figs. 6(b) and 6(c). It can be seen that the MLD deepens by approximately 30-40 meters (the standard deviation is about 10 meters) along the track of the tropical cyclone, which is almost a 100% increase compared to its initial value in Fig. 6(a). It is noted that CPL.AOW has stronger MLD deepening than HYCOM, but weaker SST cooling. We hypothesize that this is because (1) the parameterization of the ocean mixing layer is

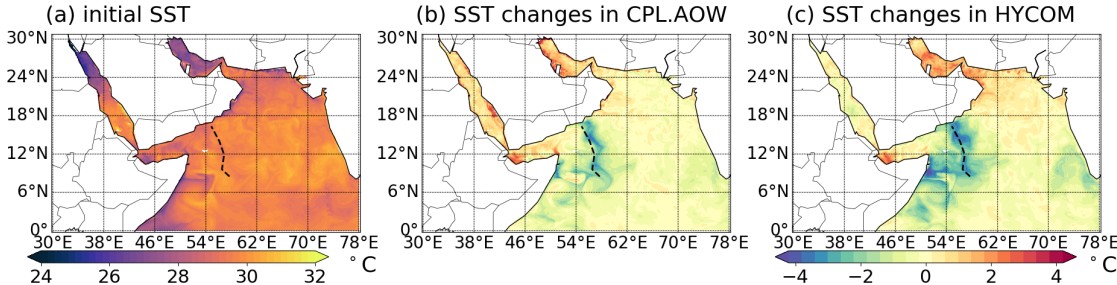

**Figure 5.** The evolution of SST during the tropical cyclone event. Panel (a) shows the SST when the simulation is initialized at 00 UTC May 20; Panel (b) illustrates the ensemble averaged SST changes between 00 UTC May 20 and 00 UTC May 27 in CPL; Panel (c) shows the SST changes in the HYCOM analysis for the same period.

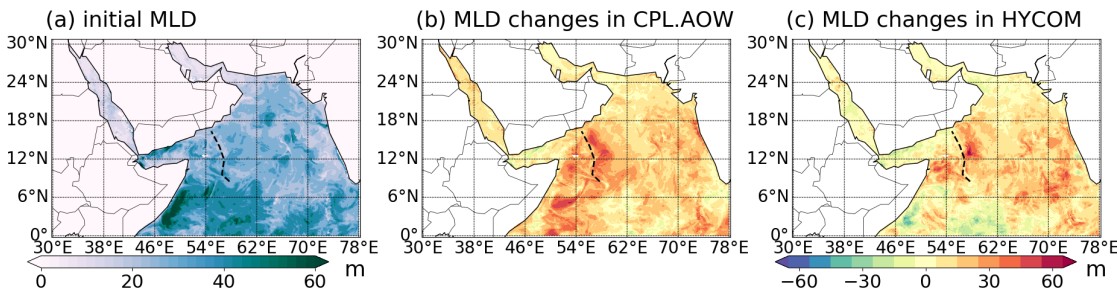

**Figure 6.** The evolution of MLD during the tropical cyclone event. Panel (a) shows the MLD at 00 UTC May 20 when the simulation is initialized; Panel (b) shows ensemble averaged MLD changes between 00 UTC May 20 and 00 UTC May 27 in CPL; Panel (c) shows the MLD changes in the HYCOM analysis for the same period.

different when the effects of Langmuir turbulence are considered in CPL.AOW; (2) the atmosphere forcing used in the coupled model has a higher spatial and temporal resolution that makes the SST and MLD different.

### 4.3 Waves

Ocean surface waves are expected to affect air–sea interactions. The ensemble mean significant wave height ($H_s$) and the ensemble standard deviation obtained from the coupled simulation are shown in Fig. 7. The snapshots of the ensemble-averaged $H_s$ are presented in Fig. 7(a-c). On May 26, the ensemble averaged $H_s$ is as high as 8 m near the eye wall of the tropical cyclone. Figure 7(b) and (c) shows that alternating regions of high and low waves can be observed between 12°N to 24°N and from 60°E to 75°E. The spatial pattern of high and low beams of $H_s$ is due to surface wave refraction by ocean currents. We have performed uncoupled WW3 simulations to investigate these beams and more details can be found in Sun et al. (2022).

Near the eye wall of the tropical cyclone, the standard deviation of $H_s$ from the ensembles is approximately 3 m, showing greater variance near the eye wall (Fig. 7(d-f)), while the spatial variability of $H_s$, with alternating high and low beams, is

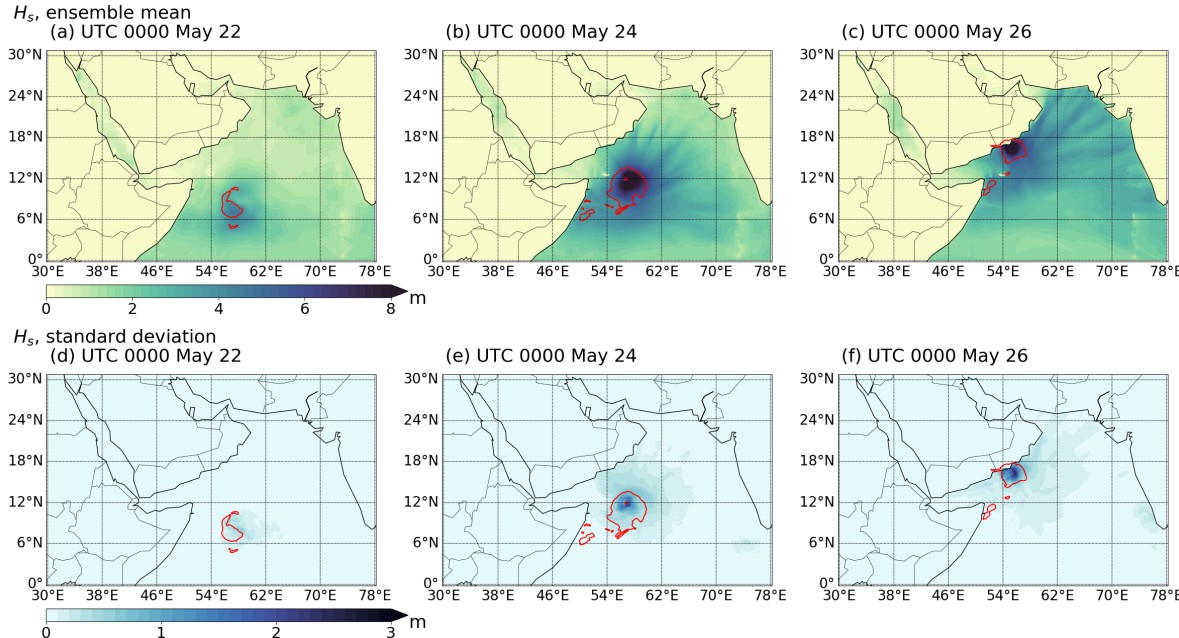

**Figure 7.** Snapshots of the significant wave height $H_s$ at 00 UTC May 22 24, and 26, 2018. Panel (a-c) show the ensemble averaged $H_s$ obtained from CPL.AOW; Panel (d-f) show the standard deviation of $H_s$ of the ensembles from CPL.AOW. The 15 m/s contour of wind speed is used to highlight the location of the tropical cyclone.

consistent throughout the ensemble. Although CPL.AOW captures the overall spatial variability of $H_s$, the exact location of the beams deviates from the altimetry observations, since the central location of the tropical cyclone is not well captured by CPL.AOW. The comparison of the modeled $H_s$ with altimeter data is shown in Appendix D.

## 5   Sensitivity Analysis to Wave Coupling

To explore the effects of the surface waves, coupled simulations were run using three recent parameterizations of Langmuir turbulence. We compared the characteristics of the tropical cyclone (e.g., track, intensity, and wind speed) and the changes in the ocean (e.g., SST and MLD). In the sensitivity analysis, we compare the simulation results without Langmuir turbulence (NoLT) and those with Langmuir turbulence (LF17, VR12-MA, and LF17-ST). This aims to illustrate the sensitivity of the coupled model to Langmuir turbulence, which may deepen the MLD and cool the SST during a tropical cyclone event. Note

that the coupled run using LF17 is identical to CPL.AOW in Section 4. To evaluate the effect of Langmuir turbulence on the ocean, we also performed the simulations using spectral nudging in WRF in addition to the "free runs" (simulations without spectral nudging). The spectral nudging is performed because of the uncertainties of the atmosphere model, especially for the tracks of the cyclones. By restraining the uncertainty of the atmosphere using spectral nudging, we are able to highlight the impact of Langmuir turbulence on the ocean. In the present study, WRF nudges the model fields to NCEP FNL data and

the wavenumber for spectral nudging is about 600 km. Although the simulations with spectral nudging have smaller internal variability, they underestimate the intensity of the tropical cyclone in the simulations.

    Similar to the simulation results shown in Section 4, the characteristics of the cyclones (i.e., tracks, cyclone central pressure, wind speed) are not significantly different from the simulations using different parameterizations. The cyclone characteristics in the sensitivity analysis are detailed in Appendix C. In this section, we only highlight the sensitivity of SST, ocean mixed layer,

and other surface fluxes to the parameterization of Langmuir turbulence. We also performed a similar sensitivity analysis for different surface roughness parameters that may impact the atmosphere surface variables. The sensitivity analyses of Stokes-Advection, Stokes-Corolis, and wave-induced momentum fluxes are also performed. However, these results are summarized in appendix C because they are not significantly different.

## 5.1   SST and ocean mixed layer

To highlight the SST differences between the simulations, we plotted the SST differences between the runs with and without Langmuir turbulence in Figure 8, with regions of significant SST changes ($P < 0.05$) highlighted. It can be seen that in CPL.LF17 and CPL.LF17-ST the SST cooling near the tracks of the cyclone is stronger by about $0.5°C$ in comparison with CPL.NoLT due to the effect of Langmuir turbulence. When the spectral nudging is added to reduce the randomness of the atmosphere model, the cyclones within the ensemble simulations are more similar and thus the SST cooling is more signifi-

cant. The results obtained using LF17 and LF17-ST are generally consistent, because they use a similar way to calculate the enhancement coefficient and entrainment flux. Their differences are because of different options to parameterize the Langmuir number La. On the other hand, when using VR12-MA, we observed weaker SST cooling compared with the simulation without Langmuir turbulence. Though it is demonstrated in Reichl et al. (2016b) that VR12-MA is not adequate to parameterize the Langmuir turbulence, this non-intuitive SST change needs to be documented and discussed. After we examine the vertical

profiles of the ocean, we hypothesize that too much diffusivity is added to the ocean current velocity when using VR12-MA in this case. The reduction of vertical gradient in ocean current velocity reduces the ocean mixing in KPP, and contributes to a weaker SST cooling. More details on the SST cooling are presented in Section 5.2.

    The comparison between the MLDs obtained from the simulations is shown in Fig. 9. Again, we highlight the regions with significant MLD changes ($P < 0.05$) for both "free run" and those with spectral nudging. Due to Langmuir turbulence, the MLDs

in CPL.LF17 and CPL.LF17-ST are deeper by a maximum of about 20 meters than that of CPL.NoLT. When using VR12-MA, the MLD is shallower again due to the reduction of turbulent shear from the parameterization of Langmuir turbulence in this case. It is noted that the largest SST and MLD changes are not centered on the location of the tropical cyclone. We hypothesize that this is because (1) the SST and MLD changes need some time to develop and (2) the winds on the right-front quadrant of the cyclone are strongest (Moon et al., 2004; Fan et al., 2009).

## 5.2   The Vertical Profiles

To investigate the SST warming and cooling in the wake zone due to Langmuir turbulence, we examined the vertical profiles of the ocean aiming to illustrate the impact of different Langmuir turbulence options. Here we averaged the quantities of interest

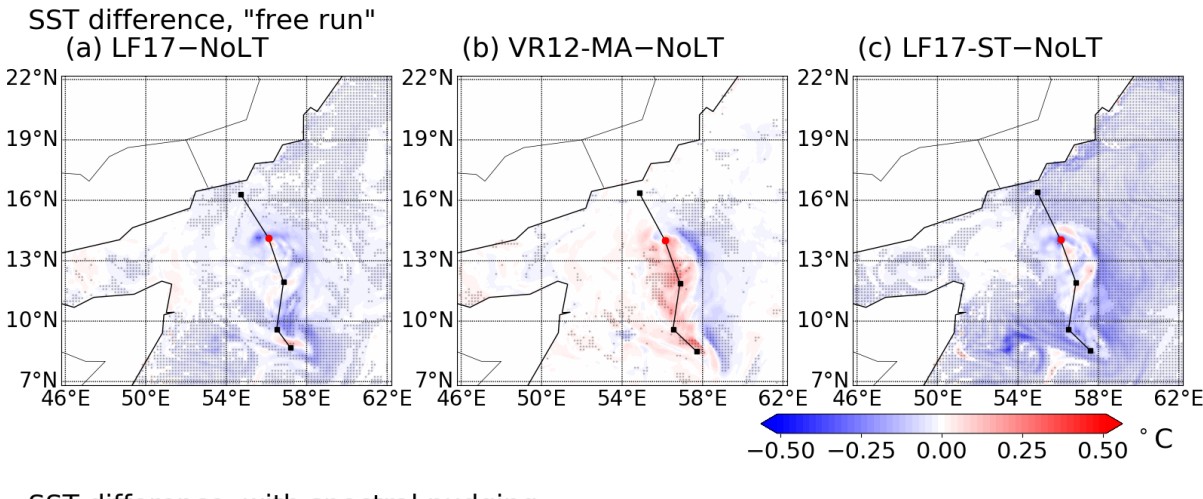

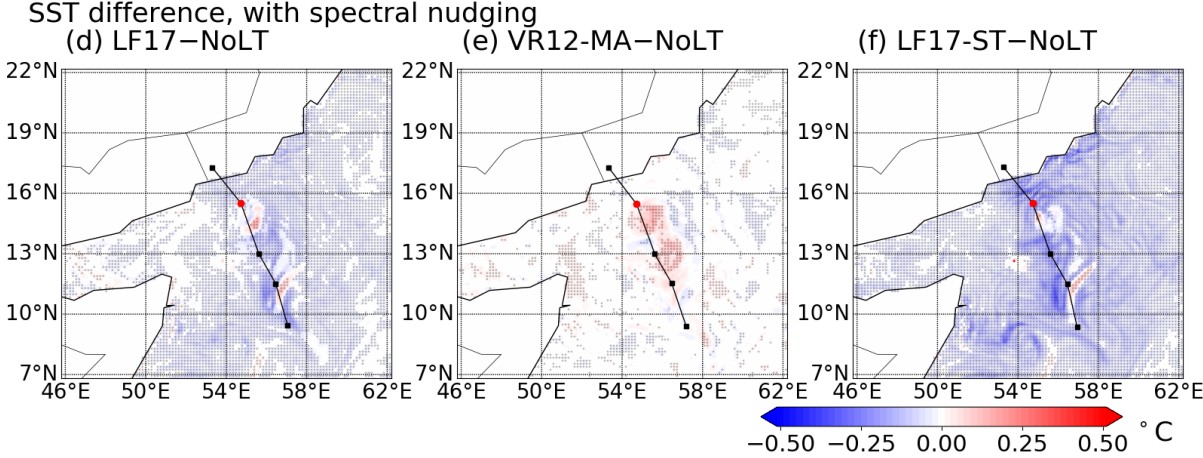

**Figure 8.** The snapshots of the ensemble averaged SST difference. Panels (a-c) show the SST difference between the simulations with Langmuir turbulence (CPL.LF17, CPL.VR12-MA, and CPL.LF17-ST) and without Langmuir turbulence (CPL.NoLT). Panels (d-f) show the same differences in the simulations with spectral nudging. The markers indicate the regions where the SST difference is significant (P < 0.05).

in the region between 11°N to 14°N and from 55°E to 57°E because large SST and MLD changes are observed in this region. Due to the internal variability of the atmosphere, we only compare the simulation results when spectral nudging is used.

To analyze the impact of different Langmuir turbulence options, in Fig. 10(a-d) we plotted the domain-averaged bulk Richardson number, buoyancy difference, vertical density gradient, and current velocity, which are the dominant terms in Eq. (6). The bulk Richardson number is plotted because it is used in the MITgcm KPP scheme to determine the boundary layer depth, which is crucial to parameterize vertical mixing. The critical Richardson number $Ri_{cr}$ is 0.3, meaning the ocean is assumed dynamically unstable and turbulent when $\mathrm{Ri} < 0.3$. It can be seen that when VR12-MA is applied, the Richard-

son number increases compared with CPL.NoLT (no Langmuir turbulence), indicating the parameterized turbulence is getting

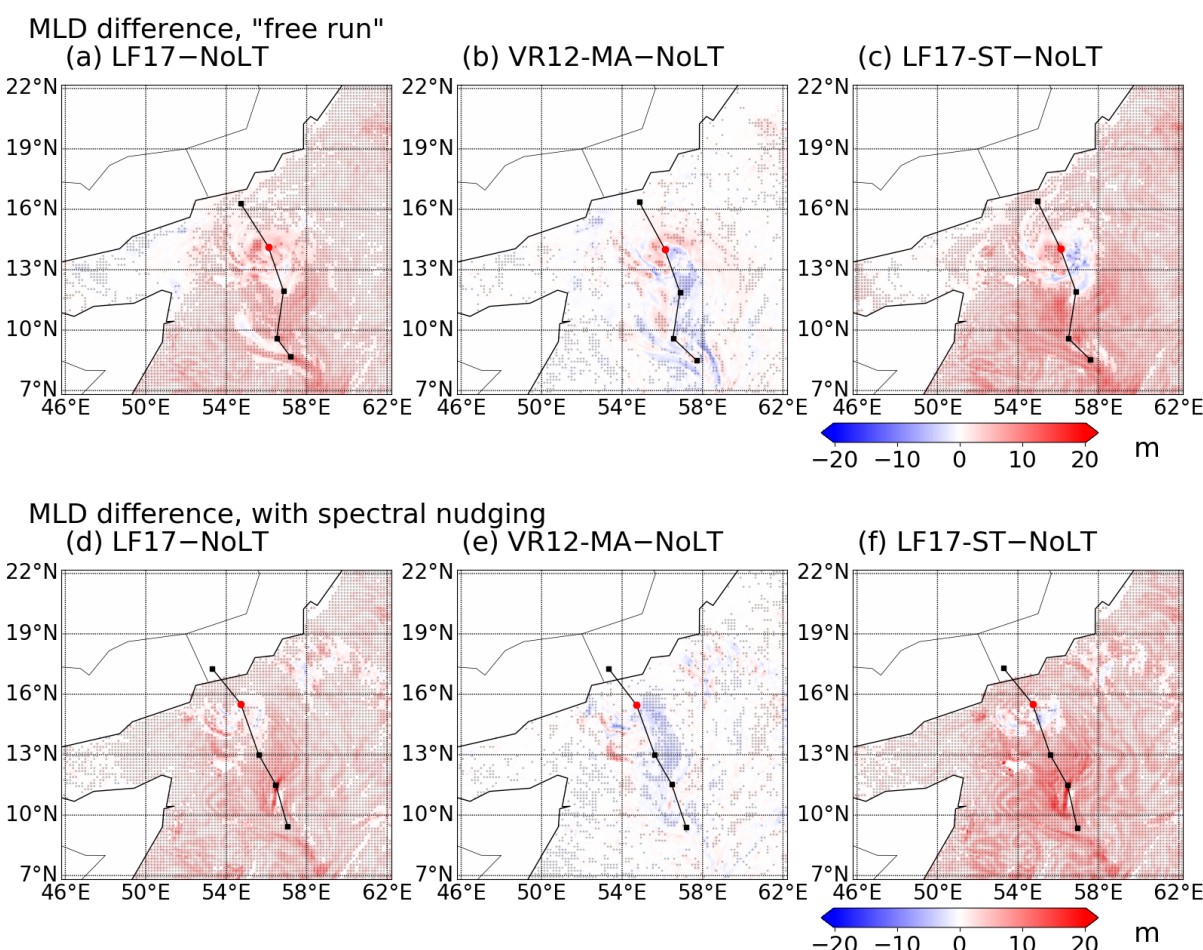

**Figure 9.** The snapshots of the ensemble averaged MLD difference. Panels (a-c) show the MLD difference between the simulations with Langmuir turbulence (CPL.LF17, CPL.VR12-MA, and CPL.LF17-ST) and without Langmuir turbulence (CPL.NoLT). Panels (d-f) show the same differences in the simulations with spectral nudging. The markers indicate the regions where the MLD difference is significant (P < 0.05).

weaker. Examining each component of the Richardson number in Eq. (6), it can be seen that buoyancy and vertical density gradient terms do not change significantly, shown in Fig. 10(b) and 10(c); while the changes of horizontal current speed can be seen in Fig. 10(d) near the surface.

When VR12-MA is applied, the Langmuir enhancement factor (see Eq. (3)) is used to amplify the KPP diffusivity term (see Eq. (5)). This reduces the vertical gradient of horizontal current, shown in Fig. 10(d). When the velocity gradient is reduced in VR12-MA, the $|\boldsymbol{u}_r - \boldsymbol{u}(z)|^2$ term in Eq. (6) decreases, and thus the Richardson number increases. This Richardson number increase results in a decrease in estimated MLD, and thus the SST cooling is weaker than the simulation without Langmuir turbulence (CPL.NoLT). To verify this, we run an ensemble of simulations (CPL.VR12-MA-NoU) that do not enhance the

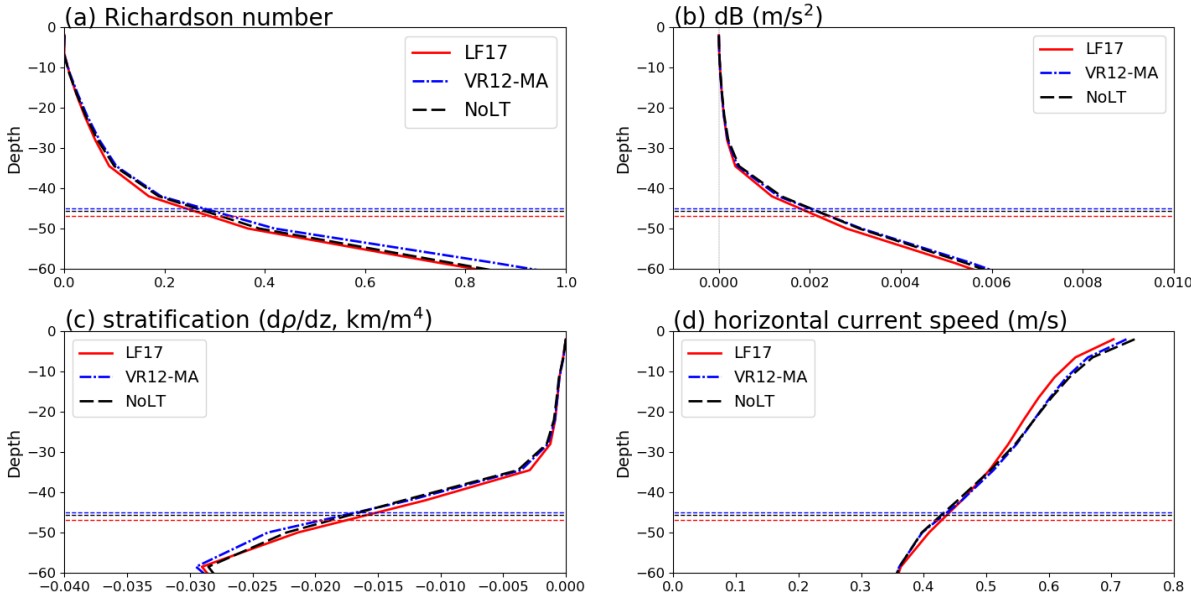

**Figure 10.** The snapshots of the vertical profiles at 00 UTC May 25 obtained in the simulations. Panel (a-d) show the Richardson number, buoyancy term, density changes, and horizontal velocity. The solid lines indicate the vertical profiles obtained from CPL.NoLT, CPL.VR12-MA, and CPL.LF17. The horizontal dashed lines indicate the KPP boundary layer in MITgcm.

KPP diffusivity for horizontal currents, then we observed cooler SST due to enhanced mixing by Langmuir turbulence. The
420 simulation results of this verification test are detailed in Appendix D. It is noted that the drawbacks of VR12-MA are also
discussed in Reichl et al. (2016b), showing that using Lagrangian currents $\boldsymbol{u}^L$ on parameterizing Langmuir turbulence can also
alleviate the bias when using VR12-MA.

On the other hand, when using LF17 in the simulations, the same enhancement factor $\varepsilon$ as VR12-MA is added, but the $V_{tl}(z)$
term is used in Eq. (7) for parameterizing the Richardson number. Although the velocity gradient $|\boldsymbol{u}_r - \boldsymbol{u}(z)|^2$ is also smaller,
shown in Fig. 10(d), the entrainment flux $V_{tl}(z)$ decreases the Richardon number. This implies stronger vertical mixing due
to the Langmuir entrainment by the tropical cyclone. Hence the SST cooling in the near wake region of the tropical cyclone
is stronger when LF17 is used than VR12-MA. This shows parameterizing the Langmuir turbulence using LF17 gives more
realistic results than VR12-MA.

## 6   Discussion and Summary

This work described the integration of WAVEWATCH III into the SKRIPS regional coupled model. The implementation
allows using or not using the wave model in the SKRIPS model. The parameterizations of the surface waves are implemented
in MITgcm to account for the impact of waves on the ocean as well as the Stokes-Advection and Stokes-Coriolis forces in the
coupled model.

To test the coupled ocean–wave–atmosphere model, we performed a series of simulations of cyclone Mekunu in the Arabian Sea, which is a representative tropical cyclone case. In order to model the uncertainty due to the atmospheric internal variabilities, we added small perturbations to the initial SST to generate 20 ensembles for each test case. Then we compared the fully coupled simulations (CPL.AOW) with coupled runs without the wave model (CPL.AO) and stand-alone atmosphere simulations (ATM.DYN). The characteristics of the tropical cyclone (e.g., track, intensity, and wind speed) obtained in the two coupled simulations are similar within uncertainty. However, the stand-alone atmosphere model sees the SST cooling from the HYCOM analysis that is stronger than in the coupled runs. Compared with the coupled simulations, in ATM.DYN the tropical cyclone has higher pressure and lower wind speed, making it less consistent with the observations.

We further tested the sensitivity of simulated characteristics of cyclone to wave coupling. The simulation results show that the characteristics of the tropical cyclone (e.g., intensity, pressure, maximum wind speed) are not sensitive to wave coupling compared with the internal variability of the model as resolved by the ensemble simulations. When the effect of Langmuir turbulence is parameterized using the LF17 and LF17-ST options that account for the Langmuir entrainment, the maximum SST cooling is about $0.5°C$ cooler and the maximum mixed layer deepening is about 20 m along the track of the tropical cyclone, indicating the surface waves play an important role in modulating the response of the upper ocean to tropical cyclone surface forcing. On the other hand, when the effect of Langmuir turbulence is parameterized using the VR12-MA, the SST cooling and MLD deepening are weaker due to the changes of current shear in the coupled simulation.

The results presented here motivate further studies to evaluate and improve this and other regional or high-resolution coupled models for investigating dynamical processes and forecasting applications, especially the interaction between ocean and waves and their feedback with the atmosphere.

*Code availability.* The source code of the coupled model is maintained on Github (https://github.com/iurnus/scripps_kaust_model) and Zenodo (https://doi.org/10.5281/zenodo.4014267). The code documentation is available at https://github.com/iurnus/scripps_kaust_model_doc

## Appendix A: Langmuir Numbers

In this work, we used the Langmuir number summarized in Li et al. (2019). The Langmuir number $\mathrm{La}_{\mathrm{LP}}$ is defined from the surface layer (a fraction of the mixed layer, upper 20% in their definition) averaged Stokes drift $\langle u^S \rangle_{\mathrm{SL}}$ and a reference Stokes drift $u_{\mathrm{ref}}^S$ near the base of the mixed layer:

$$\mathrm{La}_{\mathrm{SL}} = \left[ u_* / (\langle u^S \rangle_{\mathrm{SL}} - u_{\mathrm{ref}}^S) \right]^{\frac{1}{2}}. \tag{A1}$$

The Langmuir number $\mathrm{La}_{\mathrm{LP}}$ is used in the scaling of Langmuir-enhanced entrainment in LF17 in Eq. (8).

The projected Langmuir number $\mathrm{La}_{\mathrm{SLP}}$ considers the misalignment between wind and waves:

$$\mathrm{La}_{\mathrm{SLP}} = \left[ \frac{u_* \cos\theta_{\mathrm{wl}}}{\langle u^S \rangle_{\mathrm{SL}} \cos(\theta_{\mathrm{ww}} - \theta_{\mathrm{wl}})} \right]^{\frac{1}{2}}, \tag{A2}$$

where $\theta_{\mathrm{ww}}$ is the misalignment between wind and waves and $\theta_{\mathrm{wl}}$ between wind and Langmuir cells. The Langmuir number $\mathrm{La}_{\mathrm{SLP}}$ is used in VR12-MA and the scaling of Langmuir-enhanced diffusivity in LF17 in Eq. (4).

## Appendix B: Ocean Roughness Closures

In this work, we implemented three different ocean roughness closure models in (Olabarrieta et al., 2012): (1) DGHQ model based on wave age (Drennan et al., 2003); (2) TY2001 model based on wave steepness (Taylor and Yelland, 2001); and (3) OOST model that considers both the effects of wave age and steepness (Oost et al., 2002). These options are implemented in the MYNN surface layer scheme in WRF.

In Taylor and Yelland (2001), the ocean surface roughness is parameterized as:

$$\frac{z_0}{H_s} = 1200(H_s/L_p)^{4.5}, \tag{B1}$$

where $z_0$ is the ocean roughness; $L_p$ is the wavelength at the peak of the wave spectrum; $H_s$ is the significant wave height.

Drennan et al. (2003) proposed a wave age-based formula to characterize the ocean roughness. The wind friction velocity is also considered in this formula:

$$\frac{z_0}{H_s} = 3.35(u^*/C_p)^{3.4}, \tag{B2}$$

where $u^*$ is the wind friction velocity; $C_p$ is the wave phase speed at the peak frequency.

Oost et al. (2002) also derived the following expression for the ocean roughness based on the experimental data based on wave age and wavelength:

$$\frac{z_0}{L_p} = \frac{25}{\pi}(u^*/C_p)^{4.5}. \tag{B3}$$

We also implemented another option that uses the Charnock coefficient (CHNK) calculated from WAVEWATCH III. In the present study, we used the ST4 option in WW3 and thus the Charnock coefficients are calculated based on Ardhuin et al. (2010).

In this manuscript we used WAVEWATCH III version 6.0.7 compiled with the following switches:
F90 NOGRB NOPA LRB4 SCRIP SCRIPNC NC4 TRKNC DIST MPI PR3 UQ FLX0 LN1 ST4 STAB0 NL1 BT4 DB1
MLIM TR0 BS0 IC2 IS2 REF1 IG0 XX0 WNT2 WNX1 CRT1 CRX1 TIDE O0 O1 O2 O2a O2b O2c O3 O4 O5 O6 O7

**Appendix C:  Tropical Cyclone Characteristics Using Different Setups**

The comparison between the simulations of cyclone Mekunu as resulting from the coupled models using different setups. Here we tested different setups in parameterizing Langmuir turbulence and sea surface roughness. For the Langmuir turbulence we test VR12-MA, LF17, LF17-ST, and no Langmuir turbulence; for the sea surface roughness we test CHNK, TY2001, DGHQ, and OOST. The simulation using CHNK is the same as CPL.AOW in Section 4. Finally, we examine the impact of Stokes forces and wind stress.

Figure A1 shows that the tracks of tropical cyclones from the coupled simulations are generally consistent within the ensemble spread, although the track from CPL.AOW (CHNK) is slightly closer to IBTrACS than the other simulations. The distance error, simulated cyclone central pressure, and maximum wind speed shown in Figs. A2 and A3 are also close. Note that the differences between the ensemble-mean pressure and wind speed in the coupled models are smaller than the standard deviations shown in Fig. A2 and Fig. A3. The snapshots of the 10-m wind speed and latent heat fluxes in Fig. A5 aim to illustrate the sensitivity of the surface atmosphere to parameterizing surface roughness. It can be seen that the 10-m wind speed and latent heat loss are different when using TY2001, DGHQ, and OOST in comparison with using CHNK in CPL.AOW. It can be seen that the simulations using TY2001, DGHQ, and OOST has stronger wind obtained from the simulations, similar to the findings in Olabarrieta et al. (2012).

The impact of Stokes forces and wind stress on the characteristics of the tropical cyclone is shown in Fig. A4. The coupled simulation results without using Stokes-Advection and Stokes-Coriolis are shown using NoStokes; the results without correcting the wind stress due to the waves are shown using NoStress. It can be seen in the simulation results that the effects of Stokes forces and wind stress are not significant. Noted that the NoStokes experiment is consistent with the implicit scheme for in Li et al. (2016).

In addition to the experiment performed without Stokes-Advection and Stokes-Coriolis, we performed a test to further illustrate the difference when Langmuir turbulence options are applied. Simulations are performed to test VR12-MA, LF17, and LF17-ST options in comparison with no Langmuir turbulence NoLT. The SST differences for these simulations are shown in Fig. A7. It can be seen that SST differences in Panel (a-c) are generally consistent with those shown in Fig. 8, except for the regions near the track of the tropical cyclone where the uncertainty is large. The SST cooling and warming patterns obtained from the spectral nudging experiments in Panel (d-f) are generally consistent with those shown in Fig. 8. This demonstrates that the simulation results are not sensitive to the implicit or explicit options for Stokes-Advection and Stokes-Coriolis.

## Appendix D: Impact of Velocity Shear in KPP

To illustrate the impact of velocity shear in KPP, we performed a 20-member ensemble simulation using VR12-MA scheme, but do not enhance the velocity scale when calculating the KPP diffusivity. The simulation results of this experiment (CPL.VR12-MA-NoU) are shown in Fig. A6. In this experiment, It can be seen in this figure that when the diffusivity is not enhanced, the MLD deepens by about 20 m and SST cools down by about 0.5°C. This is indicating that the diffusivity of the velocity scale in VR12-MA makes an impact on the mixing layer and SST when applied to parameterize the Langmuir turbulence.

## Appendix E: The Significant Wave Height

To evaluate the simulation performance of surface waves, we compared the modeled $H_s$ with along-track $H_s$ measurements from the Jason-3 and SARAL/AltiKa altimeters. We use quality-controlled, unfiltered, and not resampled, along-track $H_s$ measurements provided by the Institut Français de Recherche pour l'Exploitation de la MER (IFREMER; ftp://ftp.ifremer.fr/ifremer/cersat/products/swath/altimeters/waves/, Queffeulou and Croizé-Fillon (2013)).

The comparison of the significant wave height $H_s$ with the altimeter data is shown in Fig. A8. We also plotted the simulation results obtained in Sun et al. (2022). WAV.WND indicates the simulation using stand-alone WAVEWATCH III model driven by ERA5 wind only; WAV.CUR indicates the simulation using stand-alone WAVEWATCH III model driven by ERA5 wind and HYCOM currents. It can be seen from Fig. A8 that the coupled model captures the focusing and defocusing of the waves. However, because of the error in the location of the tropical cyclone, the patterns of the $H_s$ are not completely consistent with the observational data.

## Appendix F: Validation Against Drifter Data

To illustrate the impact of Langmuir turbulence on the upper ocean, a sensitivity analysis of the SST and the ocean mixed layer is performed. First we compare the SST cooling obtained in the simulations with the observational data from Global Drifter Program (Lumpkin and Centurioni, 2019). In Fig. A9 we show the SST changes throughout the event along the drifter tracks. It can be seen that a drifter closest to the track of the TC (highlighted in Fig. A9) recorded a cooling of 3.81°C, while the SST cooling in both CPL.NoLT and CPL.LF17 are not significantly different (CPL.NoLT: 2.18°C; CPL.LF17: 2.17°C; standard deviation: 0.37°C). It is noted that the SST cooling in HYCOM is 3.23°C, which is closer to the drifter data than the coupled simulations. However, because the in-situ observations are few in this region, future work still needs to be done to evaluate the performance of the coupled model.

*Author contributions.* RS worked on the coding tasks for integrating WW3 to the coupled system, wrote the code documentation, and performed the ensemble simulation. RS and AC drafted the initial manuscript. RS, ABVB, and SL implemented the WW3 wave model.

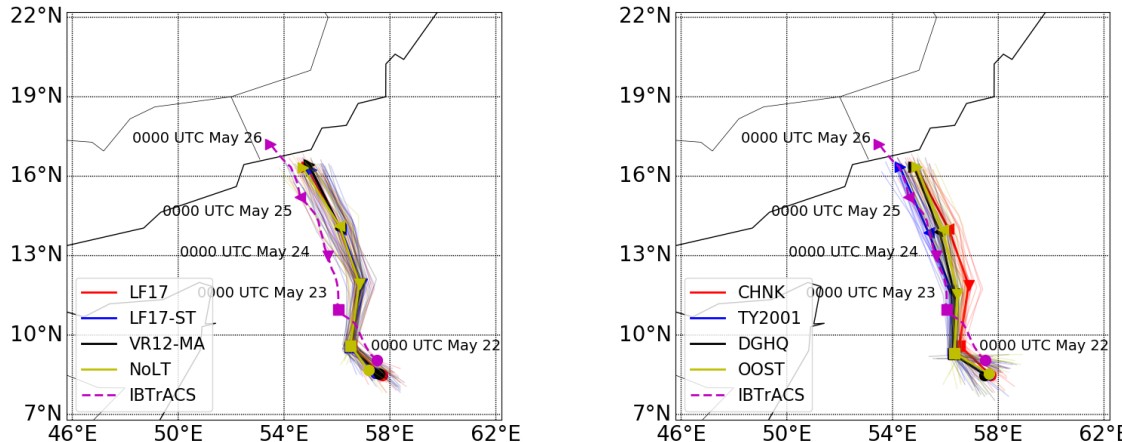

**Figure A1.** The tracks of cyclone Mekunu from the coupled simulations using different options to parameterize (a) Langmuir turbulence and (b) surface roughness. The thick solid lines indicate the locations of the center of the tropical cyclone obtained from averaging all ensemble members. The thin solid lines in the background denote the tracks of each ensemble member. The dashed lines denote the track of the tropical cyclone in IBTrACS data. The text and markers highlight the time and locations of the cyclone at specific times.

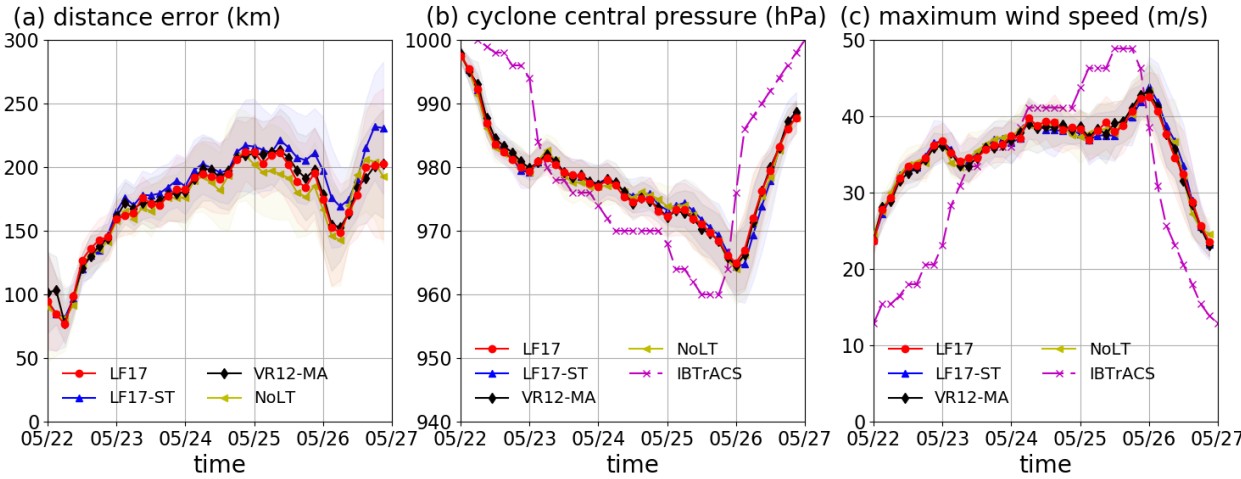

**Figure A2.** The characteristics of cyclone Mekunu obtained from the coupled simulations using different options to parameterize Langmuir turbulence. The solid lines indicate the ensemble averaged simulation results; the shaded areas indicate the standard deviation of the results. Panel (a) shows the distance errors in comparison with IBTrACS data; Panel (b) shows the cyclone central pressure; Panel (c) shows the maximum wind speed.

All authors designed the computational framework and the numerical experiments. All authors discussed the results and contributed to the writing of the final manuscript.

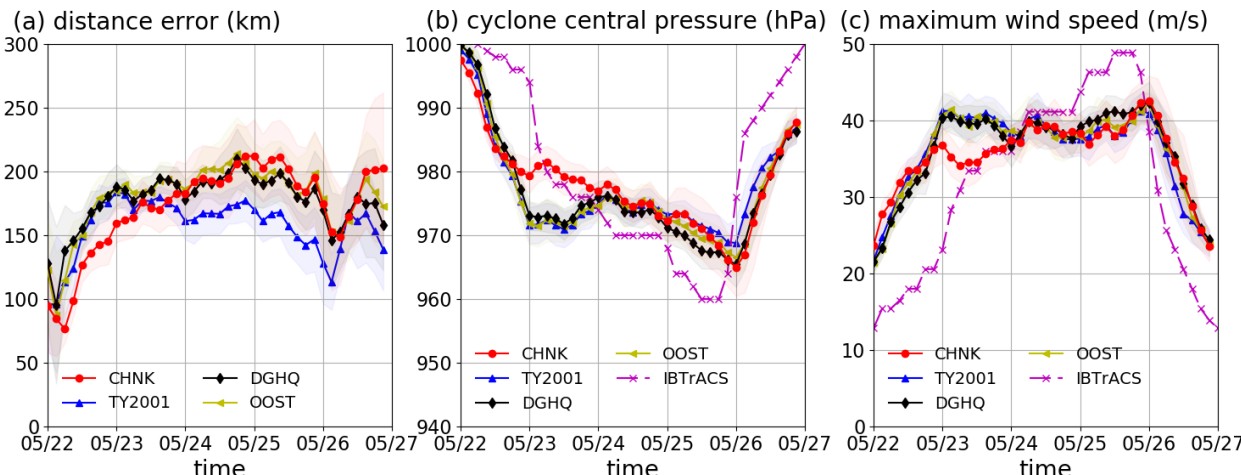

**Figure A3.** The characteristics of cyclone Mekunu obtained from the coupled simulations using different options to parameterize surface roughness. The solid lines indicate the ensemble averaged simulation results; the shaded areas indicate the standard deviation of the results. Panel (a) shows the distance errors in comparison with IBTrACS data; Panel (b) shows the cyclone central pressure; Panel (c) shows the maximum wind speed.

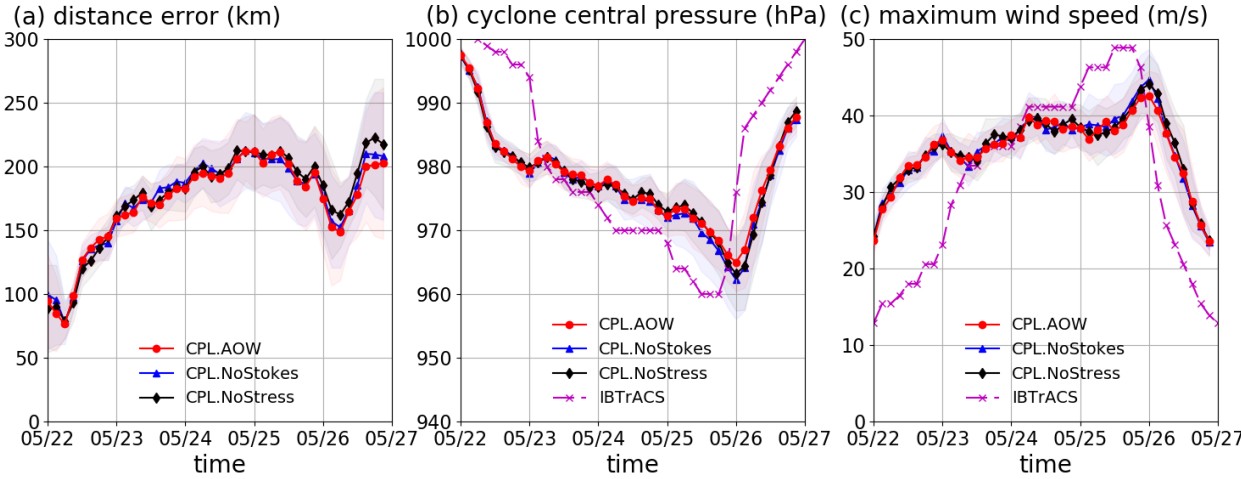

**Figure A4.** The characteristics of cyclone Mekunu obtained from the coupled simulations that do not consider the effect of Stokes forces (Stokes-Advection and Stokes-Coriolis) and wind stress corrections. The solid lines indicate the ensemble averaged simulation results; the shaded areas indicate the standard deviation of the results. Panel (a) shows the distance errors in comparison with IBTrACS data; Panel (b) shows the cyclone central pressure; Panel (c) shows the maximum wind speed.

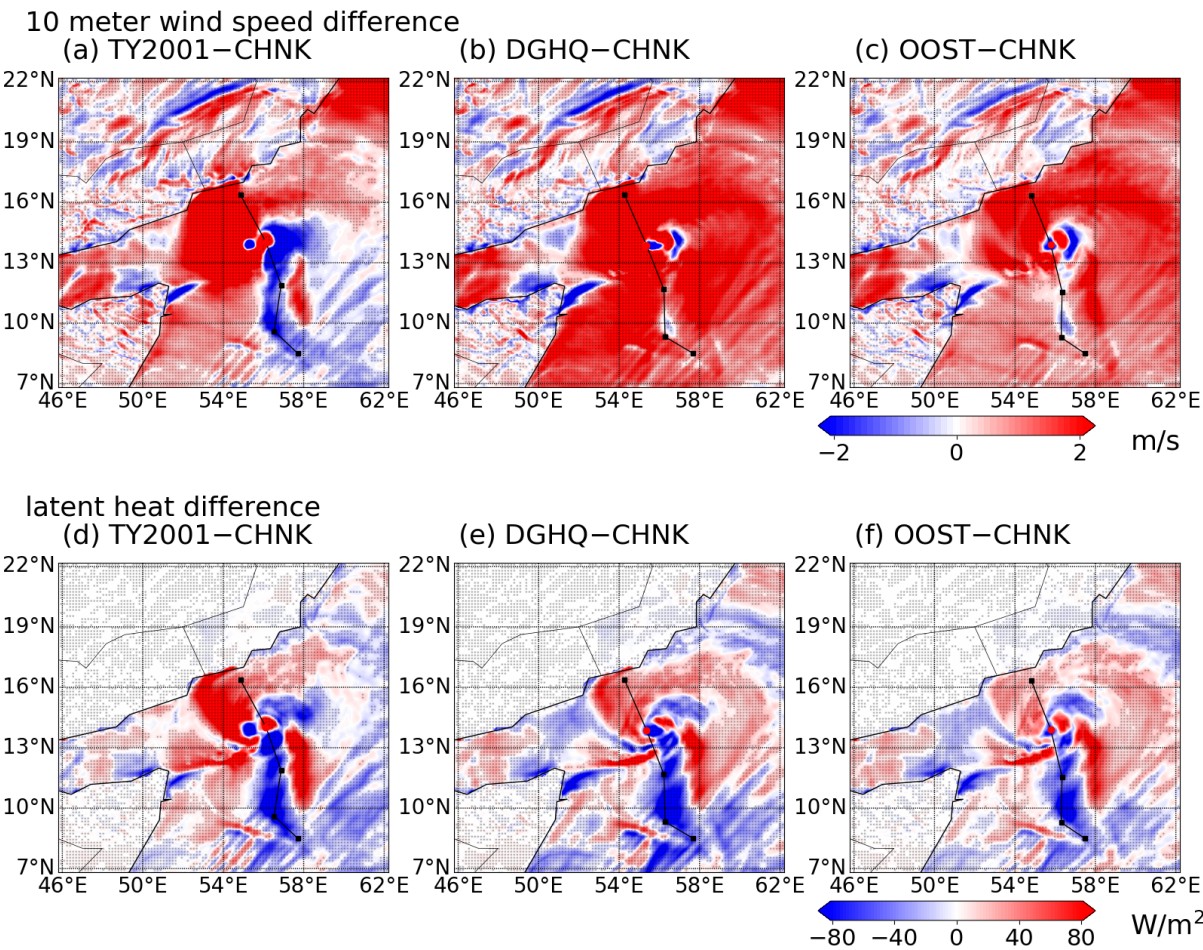

**Figure A5.** The snapshots of the ensemble averaged difference in 10-m wind speed and latent heat flux. Panels (a-c) show the 10-m wind speed difference between the simulations using different options to parameterize the surface roughness (TY2001, DGHQ, and OOST) compared with CPL.AOW (CHNK). Panels (d-f) show the latent heat differences for the same simulations. The markers indicate the regions where the differences are significant (P < 0.05).

*Competing interests.* No competing interests are present.

*Acknowledgements.* We gratefully acknowledge the research funding (grant number: OSR-2016-RPP-3268.02) from KAUST (King Ab-
dullah University of Science and Technology). We also appreciate the computational resources on supercomputer Shaheen II and the
assistance provided by KAUST Supercomputer Laboratory. RS and ACS were supported by ONR ASTRAL research initiative (N00014-
23-1-2092). ABVB was supported by NASA award 80NSSC19K1004 through the S-MODE program. ACS was supported by NOAA
Grant NA18OAR4310405 and ONR MISOBOB research initiative (N00014-17-S-B001). BDC and MRM were supported by NOAA Grant

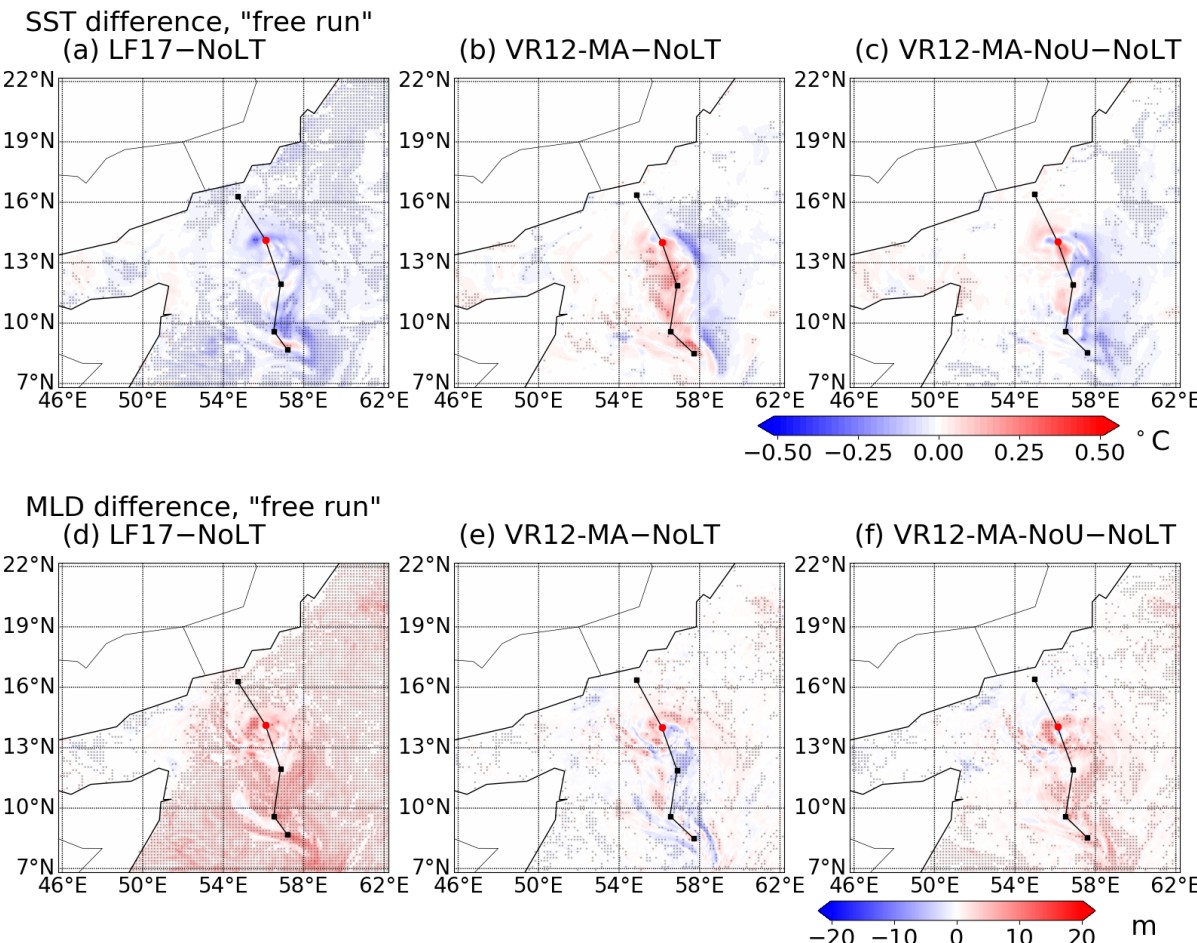

**Figure A6.** The snapshot of the ensemble averaged SST and MLD difference. Panels (a-b) show the SST difference between the simulations with Langmuir turbulence (CPL.LF17, CPL.VR12-MA) and without Langmuir turbulence (CPL.NoLT). Panel (c) shows the SST difference using VR12-MA but does not enhance the diffusion coefficient for current velocity. Panels (d-f) show the differences in MLD. The markers indicate the regions where the SST difference is significant (P < 0.05).

NA21OAR4310257, NA18OAR4310403, and NA22OAR4310597. AJM was partly supported by the National Science Foundation (OCE-2022868). We thank Baylor Fox-Kemper, Qing Li, and Zhihua Zheng for discussing the implementation of the Langmuir turbulence parameterizations.

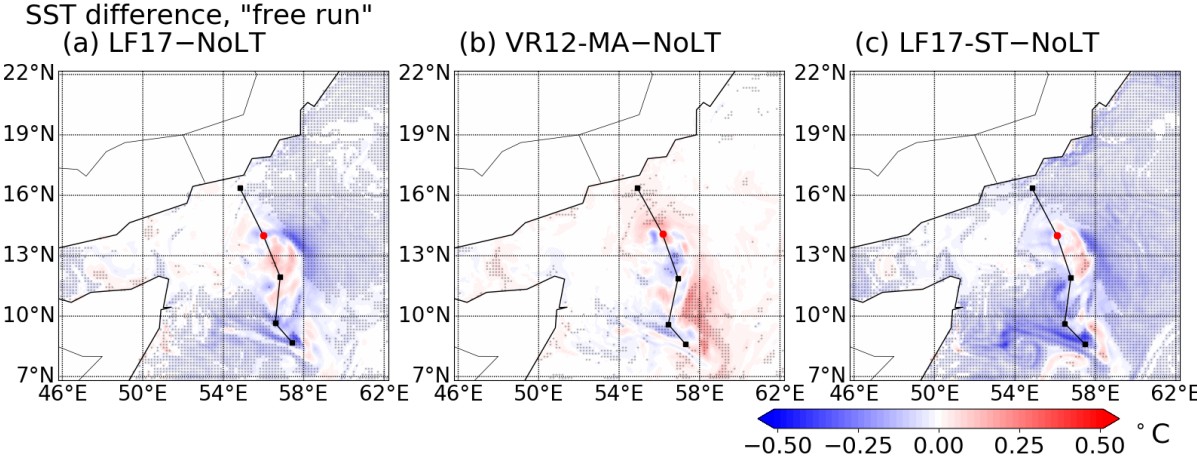

SST difference, "free run"

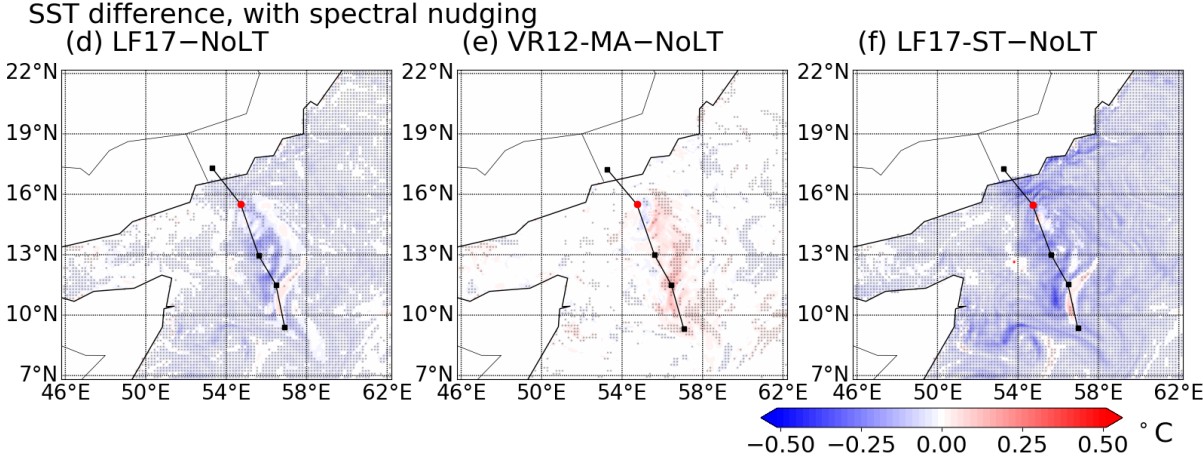

SST difference, with spectral nudging

**Figure A7.** The snapshots of the ensemble averaged SST difference. Panels (a-c) show the SST difference between the simulations with Langmuir turbulence (CPL.LF17, CPL.VR12-MA, and CPL.LF17-ST) and without Langmuir turbulence (CPL.NoLT). Panels (d-f) show the same differences in the simulations with spectral nudging. The markers indicate the regions where the SST difference is significant (P < 0.05).

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

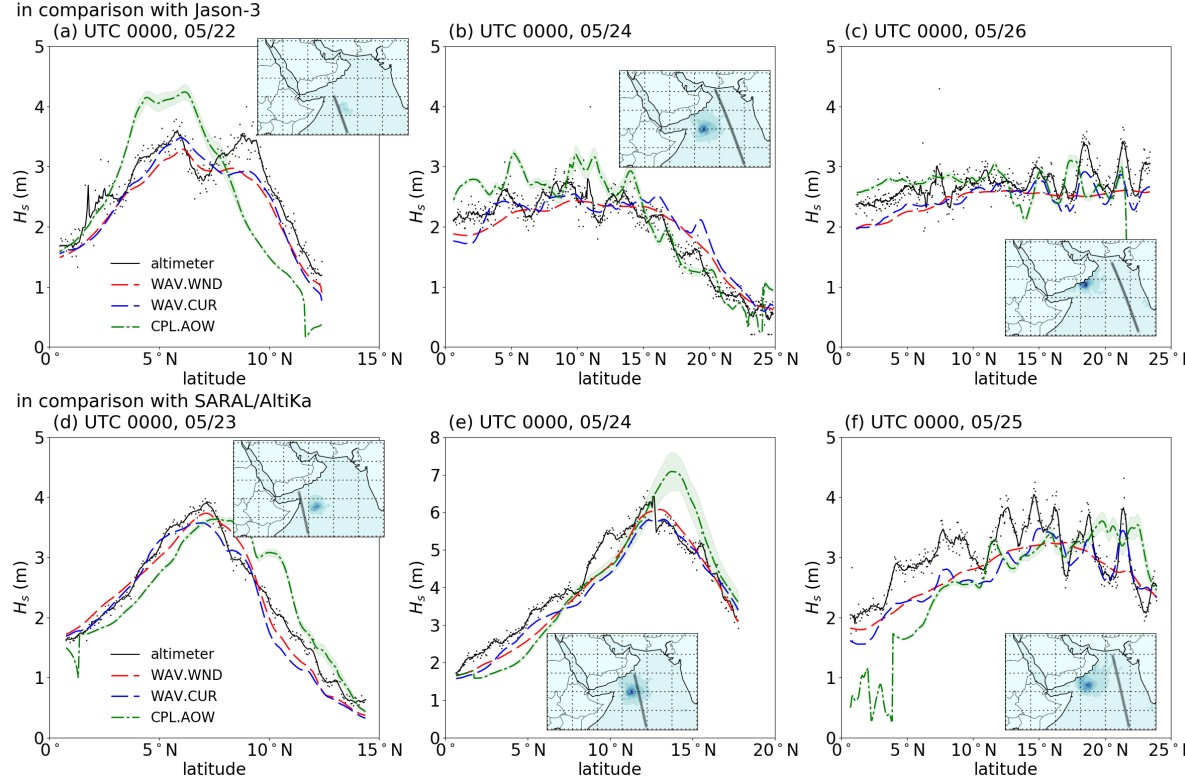

**Figure A8.** The ensemble averaged significant wave height in comparison with the altimeter data. Panels (a-c) show the comparison with Jason-3 data; Panels (d-f) show the comparison with SARAL data. The simulation results from Sun et al. (2022) are also presented. The shaded areas indicate the standard deviation of wave height in the ensemble simulations.

Bender, M. A. and Ginis, I.: Real-case simulations of hurricane–ocean interaction using a high-resolution coupled model: Effects on hurricane intensity, Monthly Weather Review, 128, 917–946, 2000.

Bhatia, K., Vecchi, G., Murakami, H., Underwood, S., and Kossin, J.: Projected response of tropical cyclone intensity and intensification in a global climate model, Journal of Climate, 31, 8281–8303, 2018.

Blair, A., Ginis, I., Hara, T., and Ulhorn, E.: Impact of Langmuir turbulence on upper ocean response to Hurricane Edouard: Model and observations, Journal of Geophysical Research: Oceans, 122, 9712–9724, 2017.

Breivik, Ø., Janssen, P. A., and Bidlot, J.-R.: Approximate Stokes drift profiles in deep water, Journal of Physical Oceanography, 44, 2433–

2445, 2014.

Breivik, Ø., Bidlot, J.-R., and Janssen, P. A.: A Stokes drift approximation based on the Phillips spectrum, Ocean Modelling, 100, 49–56, 2016.

Campin, J.-M., Heimbach, P., Losch, M., Forget, G., edhill3, Adcroft, A., amolod, Menemenlis, D., dfer22, Hill, C., Jahn, O., Scott, J., stephdut, Mazloff, M., baylorfk, antnguyen13, Doddridge, E., Fenty, I., Bates, M., Martin, T., Abernathey, R., samarkhatiwala, Smith, T.,

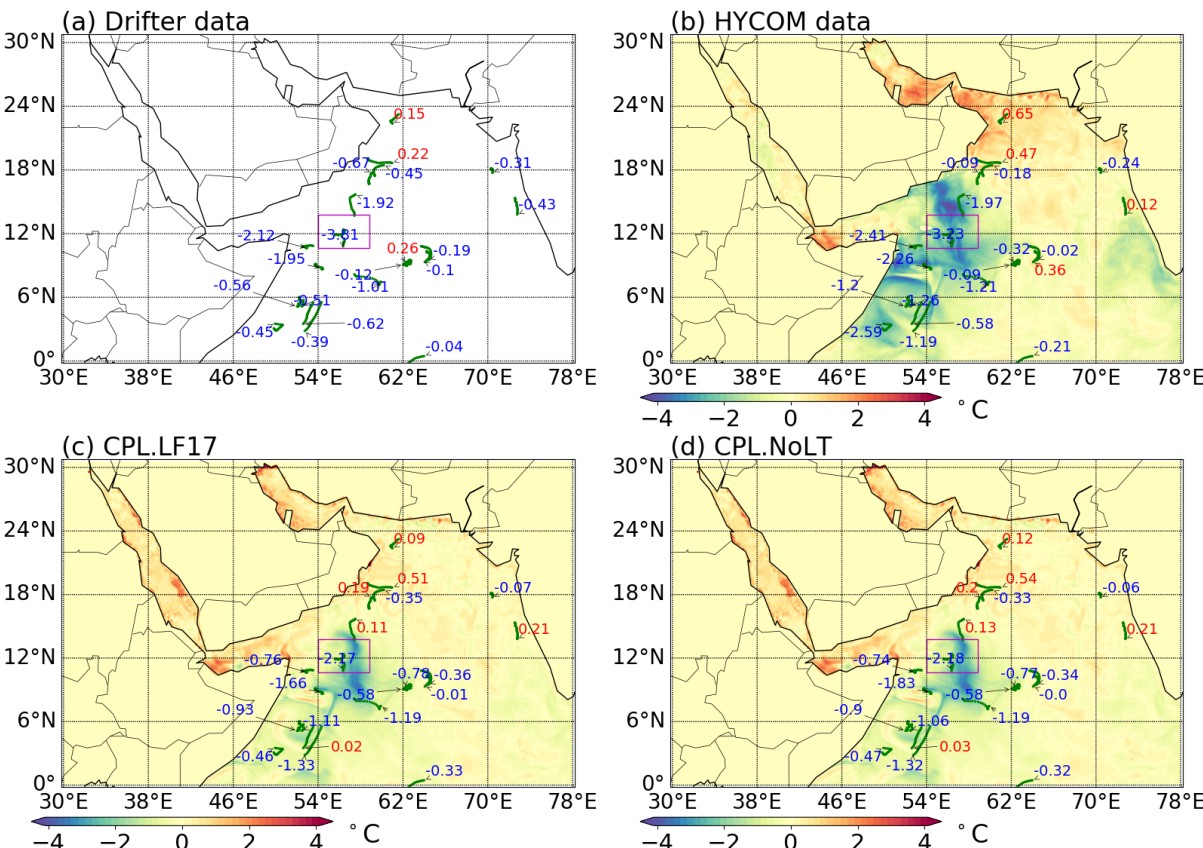

**Figure A9.** Evolution of the SST during the tropical cyclone event in comparison with drifter data. Panel (a) shows the SST changes during the event from drifter data; Panels (b-d) show the ensemble averaged SST changes from HYCOM, CPL.LF17, and CPL.NoLT, respectively. The red numbers indicate SST warming during the event; the blue numbers indicate SST cooling during the event.

Lauderdale, J., hongandyan, Deremble, B., raphael dussin, Bourgault, P., dngoldberg, and T., A. T.: MITgcm/MITgcm: checkpoint67m, https://doi.org/10.5281/zenodo.3492298, 2019.

Cerovečki, I., Sun, R., Bromwich, D. H., Zou, X., Mazloff, M. R., and Wang, S.-H.: Impact of downward longwave radiative deficits on Antarctic sea-ice extent predictability during the sea ice growth period, Environmental Research Letters, 17, 084 008, 2022.

Charnock, H.: Wind stress on a water surface, Quarterly Journal of the Royal Meteorological Society, 81, 639–640, 1955.

Chassignet, E. P., Hurlburt, H. E., Smedstad, O. M., Halliwell, G. R., Hogan, P. J., Wallcraft, A. J., Baraille, R., and Bleck, R.: The HYCOM (hybrid coordinate ocean model) data assimilative system, Journal of Marine Systems, 65, 60–83, 2007.

Chen, S. S., Price, J. F., Zhao, W., Donelan, M. A., and Walsh, E. J.: The CBLAST-Hurricane program and the next-generation fully coupled atmosphere–wave–ocean models for hurricane research and prediction, Bulletin of the American Meteorological Society, 88, 311–318, 2007a.

Chen, S. S., Price, J. F., Zhao, W., Donelan, M. A., and Walsh, E. J.: The CBLAST-Hurricane program and the next-generation fully coupled atmosphere–wave–ocean models for hurricane research and prediction, Bulletin of the American Meteorological Society, 88, 311–318, 2007b.

Couvelard, X., Lemarié, F., Samson, G., Redelsperger, J.-L., Ardhuin, F., Benshila, R., and Madec, G.: Development of a two-way-coupled ocean–wave model: assessment on a global NEMO (v3. 6)–WW3 (v6. 02) coupled configuration, Geoscientific Model Development, 13, 585    3067–3090, 2020.

Craik, A. D. and Leibovich, S.: A rational model for Langmuir circulations, Journal of Fluid Mechanics, 73, 401–426, 1976.

D'Asaro, E. A.: Turbulent vertical kinetic energy in the ocean mixed layer, Journal of Physical Oceanography, 31, 3530–3537, 2001.

D'Asaro, E. A., Thomson, J., Shcherbina, A., Harcourt, R., Cronin, M., Hemer, M., and Fox-Kemper, B.: Quantifying upper ocean turbulence driven by surface waves, Geophysical Research Letters, 41, 102–107, 2014.

Drennan, W. M., Graber, H. C., Hauser, D., and Quentin, C.: On the wave age dependence of wind stress over pure wind seas, Journal of Geophysical Research: Oceans, 108, 2003.

Dube, S. K., Rao, A. D., Sinha, P. C., Murty, T. S., and Bahulayan, N.: Storm surge in the Bay of Bengal and Arabian Sea the problem and its prediction, Mausam, 48, 283–304, 1997.

Emanuel, K. A.: The theory of hurricanes, Annual Review of Fluid Mechanics, 23, 179–196, 1991.

Evan, A. T. and Camargo, S. J.: A climatology of Arabian Sea cyclonic storms, Journal of Climate, 24, 140–158, 2011.

Evan, A. T., Kossin, J. P., and Ramanathan, V.: Arabian Sea tropical cyclones intensified by emissions of black carbon and other aerosols, Nature, 479, 94–97, 2011.

Fairall, C., Bradley, E. F., Hare, J., Grachev, A., and Edson, J.: Bulk parameterization of air–sea fluxes: Updates and verification for the COARE algorithm, Journal of Climate, 16, 571–591, 2003.

Fan, Y., Ginis, I., and Hara, T.: The effect of wind–wave–current interaction on air–sea momentum fluxes and ocean response in tropical cyclones, Journal of Physical Oceanography, 39, 1019–1034, 2009.

Government of India, Ministry of Earth Sciences, I. M. D.: Extremely severe cyclonic storm, "MEKUNU" over the Arabian Sea (21–27 May 2018): A report., Tech. rep., New Delhi: India Meteorological Department. Retrieved from http://www.rsmcnewdelhi.imd.gov.in/index. php?option=com_content&view=article&id=198:preliminary-report-2018&catid=12:publications&Itemid=540&lang=en, 2018.

Henderson-Sellers, A., Zhang, H., Berz, G., Emanuel, K., Gray, W., Landsea, C., Holland, G., Lighthill, J., Shieh, S.-L., Webster, P., et al.: Tropical cyclones and global climate change: A post-IPCC assessment, Bulletin of the American Meteorological Society, 79, 19–38, 1998.

Hill, C., DeLuca, C., Balaji, Suarez, M., and Silva, A.: The architecture of the earth system modeling framework, Computing in Science & Engineering, 6, 18–28, 2004.

Iacono, M. J., Delamere, J. S., Mlawer, E. J., Shephard, M. W., Clough, S. A., and Collins, W. D.: Radiative forcing by long-lived greenhouse 610    gases: Calculations with the AER radiative transfer models, Journal of Geophysical Research: Atmospheres, 113, 2008.

Janssen, P. A.: Ocean wave effects on the daily cycle in SST, Journal of Geophysical Research: Oceans, 117, 2012.

Jenkins, A. D.: The use of a wave prediction model for driving a near-surface current model, Deutsche Hydrografische Zeitschrift, 42, 133–149, 1989.

Kain, J. S.: The Kain–Fritsch convective parameterization: an update, Journal of Applied Meteorology, 43, 170–181, 2004.

Knapp, K. R., Kruk, M. C., Levinson, D. H., Diamond, H. J., and Neumann, C. J.: The international best track archive for climate stewardship (IBTrACS) unifying tropical cyclone data, Bulletin of the American Meteorological Society, 91, 363–376, 2010.

Knapp, K. R., Diamond, H. J., Kossin, J. P., Kruk, M. C., and Schreck, C. J.: The international best track archive for climate stewardship (IB-TrACS) unifying tropical cyclone data, version 4, Tech. rep., [NI - North Indian]. NOAA National Centers for Environmental Information. https://doi.org/10.25921/82ty-9e16 [access date: Dec 01 2021]., 2018.

Langmuir, I.: Surface motion of water induced by wind, Science, 87, 119–123, 1938.

Large, W. G. and Yeager, S. G.: Diurnal to decadal global forcing for ocean and sea-ice models: the data sets and flux climatologies, Tech. rep., NCAR Technical Note: NCAR/TN-460+STR. CGD Division of the National Center for Atmospheric Research, 2004.

Large, W. G., McWilliams, J. C., and Doney, S. C.: Oceanic vertical mixing: A review and a model with a nonlocal boundary layer parameterization, Reviews of Geophysics, 32, 363–403, 1994.

Lewis, H. W., Castillo Sanchez, J. M., Arnold, A., Fallmann, J., Saulter, A., Graham, J., Bush, M., Siddorn, J., Palmer, T., Lock, A., et al.: The UKC3 regional coupled environmental prediction system, Geoscientific Model Development, 12, 2357–2400, 2019.

Li, Q. and Fox-Kemper, B.: Assessing the effects of Langmuir turbulence on the entrainment buoyancy flux in the ocean surface boundary layer, Journal of Physical Oceanography, 47, 2863–2886, 2017.

Li, Q., Webb, A., Fox-Kemper, B., Craig, A., Danabasoglu, G., Large, W. G., and Vertenstein, M.: Langmuir mixing effects on global climate: WAVEWATCH III in CESM, Ocean Modelling, 103, 145–160, 2016.

Li, Q., Fox-Kemper, B., Breivik, Ø., and Webb, A.: Statistical models of global Langmuir mixing, Ocean Modelling, 113, 95–114, 2017.

Li, Q., Reichl, B. G., Fox-Kemper, B., Adcroft, A. J., Belcher, S. E., Danabasoglu, G., Grant, A. L., Griffies, S. M., Hallberg, R., Hara, T., et al.: Comparing ocean surface boundary vertical mixing schemes including Langmuir turbulence, Journal of Advances in Modeling Earth Systems, 11, 3545–3592, 2019.

Li, Z., Tam, C.-Y., Li, Y., Lau, N.-C., Chen, J., Chan, S., Lau, D.-S. D., and Huang, Y.: How does air-sea wave interaction affect tropical cyclone intensity? An atmosphere–wave–ocean coupled model study based on super typhoon Mangkhut (2018), Earth and Space Science, 9, 2022.

Liu, B., Liu, H., Xie, L., Guan, C., and Zhao, D.: A coupled atmosphere–wave–ocean modeling system: Simulation of the intensity of an idealized tropical cyclone, Monthly Weather Review, 139, 132–152, 2011.

Liu, W., Katsaros, K., and Businger, J.: Bulk parameterization of air-sea exchanges of heat and water vapor including the molecular constraints at the interface, Journal of Atmospheric sciences, 36, 1722–1735, 1979.

Lumpkin, R. and Centurioni, L.: Global Drifter Program quality-controlled 6-hour interpolated data from ocean surface drifting buoys., Tech. rep., NOAA National Centers for Environmental Information. Dataset. https://doi.org/10.25921/7ntx-z961. Accessed Mar 01, 2022., 2019.

Lumpkin, R. and Pazos, M.: Measuring surface currents with Surface Velocity Program drifters: the instrument, its data, and some recent
results, Lagrangian analysis and prediction of coastal and ocean dynamics, 39, 67, 2007.

Marshall, J., Adcroft, A., Hill, C., Perelman, L., and Heisey, C.: A finite-volume, incompressible Navier Stokes model for studies of the ocean on parallel computers, Journal of Geophysical Research: Oceans, 102, 5753–5766, 1997.

McWilliams, J. C., Sullivan, P. P., and Moeng, C.-H.: Langmuir turbulence in the ocean, Journal of Fluid Mechanics, 334, 1–30, 1997.

Mogensen, K. S., Magnusson, L., and Bidlot, J.-R.: Tropical cyclone sensitivity to ocean coupling in the ECMWF coupled model, Journal of
Geophysical Research: Oceans, 122, 4392–4412, 2017.

Monin, A. S. and Obukhov, A. M.: Basic laws of turbulent mixing in the surface layer of the atmosphere, Contrib. Geophys. Inst. Acad. Sci. USSR, 151, e187, 1954.

Moon, I.-J., Ginis, I., and Hara, T.: Effect of surface waves on air–sea momentum exchange. Part II: Behavior of drag coefficient under tropical cyclones, Journal of the Atmospheric Sciences, 61, 2334–2348, 2004.

Morrison, H., Thompson, G., and Tatarskii, V.: Impact of cloud microphysics on the development of trailing stratiform precipitation in a simulated squall line: Comparison of one-and two-moment schemes, Monthly Weather Review, 137, 991–1007, 2009.

Murakami, H., Vecchi, G. A., and Underwood, S.: Increasing frequency of extremely severe cyclonic storms over the Arabian Sea, Nature Climate Change, 7, 885–889, 2017.

Nakanishi, M. and Niino, H.: An improved Mellor–Yamada level-3 model with condensation physics: Its design and verification, Boundary-Layer Meteorology, 112, 1–31, 2004.

Nakanishi, M. and Niino, H.: Development of an improved turbulence closure model for the atmospheric boundary layer, Journal of the Meteorological Society of Japan. Ser. II, 87, 895–912, 2009.

Olabarrieta, M., Warner, J. C., Armstrong, B., Zambon, J. B., and He, R.: Ocean–atmosphere dynamics during Hurricane Ida and Nor'Ida: an application of the coupled ocean–atmosphere–wave–sediment transport (COAWST) modeling system, Ocean Modelling, 43, 112–137, 2012.

Oost, W., Komen, G., Jacobs, C., and Van Oort, C.: New evidence for a relation between wind stress and wave age from measurements during ASGAMAGE, Boundary-Layer Meteorology, 103, 409–438, 2002.

Pasquero, C., Desbiolles, F., and Meroni, A. N.: Air-sea interactions in the cold wakes of tropical cyclones, Geophysical Research Letters, 48, e2020GL091 185, 2021.

Price, J. F.: Upper ocean response to a hurricane, Journal of Physical Oceanography, 11, 153–175, 1981.

Queffeulou, P. and Croizé-Fillon, D.: Global altimeter SWH data set, version 10., Tech. rep., Laboratoire d'Océanographie Spatiale, IFRE-MER, Plouzané, France. Available at ftp://ftp.ifremer.fr/ifremer/cersat/products/swath/altimeters/waves/, 2013.

Rabe, T. J., Kukulka, T., Ginis, I., Hara, T., Reichl, B. G., D'Asaro, E. A., Harcourt, R. R., and Sullivan, P. P.: Langmuir turbulence under hurricane Gustav (2008), Journal of Physical Oceanography, 45, 657–677, 2015.

Rascle, N. and Ardhuin, F.: A global wave parameter database for geophysical applications. Part 2: Model validation with improved source term parameterization, Ocean Modelling, 70, 174–188, 2013.

Reichl, B. G., Ginis, I., Hara, T., Thomas, B., Kukulka, T., and Wang, D.: Impact of sea-state-dependent Langmuir turbulence on the ocean response to a tropical cyclone, Monthly Weather Review, 144, 4569–4590, 2016a.

Reichl, B. G., Wang, D., Hara, T., Ginis, I., and Kukulka, T.: Langmuir turbulence parameterization in tropical cyclone conditions, Journal of Physical Oceanography, 46, 863–886, 2016b.

Renault, L., Masson, S., Arsouze, T., Madec, G., and McWilliams, J. C.: Recipes for how to force oceanic model dynamics, Journal of Advances in Modeling Earth Systems, 12, e2019MS001 715, 2020.

Romero, L., Hypolite, D., and McWilliams, J. C.: Representing wave effects on currents, Ocean Modelling, 167, 101 873, 2021.

Salih, A. A., Baraibar, M., Mwangi, K. K., and Artan, G.: Climate change and locust outbreak in East Africa, Nature Climate Change, 10, 584–585, 2020.

Sauvage, C., Seo, H., Clayson, C. A., and Edson, J. B.: Impacts of waves and sea states on air-sea momentum flux in the Northwest Tropical Atlantic Ocean: parameterization and wave coupled climate modeling, Earth and Space Science Open Archive, p. 30, https://doi.org/10.1002/essoar.10512415.1, 2022.

Saxby, J., Crook, J., Peatman, S., Birch, C., Schwendike, J., Valdivieso da Costa, M., Castillo Sanchez, J. M., Holloway, C., Klingaman, N. P., Mitra, A., et al.: Simulations of Bay of Bengal tropical cyclones in a regional convection-permitting atmosphere–ocean coupled model, Weather and Climate Dynamics Discussions, pp. 1–40, 2021.

Schultz, C., Doney, S. C., Zhang, W. G., Regan, H., Holland, P., Meredith, M., and Stammerjohn, S.: Modeling of the influence of sea ice cycle and Langmuir circulation on the upper ocean mixed layer depth and freshwater distribution at the West Antarctic Peninsula, Journal of Geophysical Research: Oceans, 125, e2020JC016 109, 2020.

Skamarock, W. C., Klemp, J. B., Dudhia, J., Gill, D. O., Liu, Z., Berner, J., Wang, W., Powers, J. G., Duda, M. G., Barker, D. M., and Huang, X.-Y.: A description of the Advanced Research WRF Version 4, Tech. rep., NCAR Technical Note: NCAR/TN-556+STR, 145 pp, doi:10.5065/1dfh-6p97, 2019.

Smith, S. D.: Coefficients for sea surface wind stress, heat flux, and wind profiles as a function of wind speed and temperature, Journal of Geophysical Research: Oceans, 93, 15 467–15 472, 1988.

Stramma, L., Cornillon, P., and Price, J. F.: Satellite observations of sea surface cooling by hurricanes, Journal of Geophysical Research: Oceans, 91, 5031–5035, 1986.

Sun, R., Subramanian, A. C., Miller, A. J., Mazloff, M. R., Hoteit, I., and Cornuelle, B. D.: SKRIPS v1.0: a regional coupled ocean–atmosphere modeling framework (MITgcm–WRF) using ESMF/NUOPC, description and preliminary results for the Red Sea, Geoscientific Model Development, 12, 4221–4244, https://doi.org/10.5194/gmd-12-4221-2019, 2019.

Sun, R., Subramanian, A. C., Cornuelle, B. D., Mazloff, M. R., Miller, A. J., Ralph, F. M., Seo, H., and Hoteit, I.: The role of air–sea interactions in atmospheric rivers: Case studies using the SKRIPS regional coupled model, Journal of Geophysical Research: Atmospheres, 126, e2020JD032 885, 2021.

Sun, R., Villas Bôas, A. B., Subramanian, A. C., Cornuelle, B. D., Mazloff, M. R., Miller, A. J., Langodan, S., and Hoteit, I.: Focusing and defocusing of tropical cyclone generated waves by ocean current refraction, Journal of Geophysical Research: Oceans, p. e2021JC018112, 710 2022.

Suzuki, N. and Fox-Kemper, B.: Understanding Stokes forces in the wave-averaged equations, Journal of Geophysical Research: Oceans, 121, 3579–3596, 2016.

Taylor, P. K. and Yelland, M. J.: The dependence of sea surface roughness on the height and steepness of the waves, Journal of physical oceanography, 31, 572–590, 2001.

Tewari, M., Chen, F., Wang, W., Dudhia, J., LeMone, M., Mitchell, K., Ek, M., Gayno, G., Wegiel, J., and Cuenca, R.: Implementation and verification of the unified NOAH land surface model in the WRF model, in: 20th conference on weather analysis and forecasting/16th conference on numerical weather prediction, vol. 1115, American Meteorological Society Seattle, WA, 2004.

Thorpe, S.: Langmuir circulation, Annu. Rev. Fluid Mech., 36, 55–79, 2004.

Tolman, H. L.: A third-generation model for wind waves on slowly varying, unsteady, and inhomogeneous depths and currents, Journal of 720 Physical Oceanography, 21, 782–797, 1991.

Tolman, H. L.: Subgrid modeling of moveable-bed bottom friction in wind wave models, Coastal engineering, 26, 57–75, 1995.

Van Roekel, L., Fox-Kemper, B., Sullivan, P., Hamlington, P., and Haney, S.: The form and orientation of Langmuir cells for misaligned winds and waves, Journal of Geophysical Research: Oceans, 117, 2012.

Warner, J. C., Sherwood, C. R., Signell, R. P., Harris, C. K., and Arango, H. G.: Development of a three-dimensional, regional, coupled wave, 725 current, and sediment-transport model, Computers & geosciences, 34, 1284–1306, 2008.

Warner, J. C., Armstrong, B., He, R., and Zambon, J. B.: Development of a coupled ocean–atmosphere–wave–sediment transport (COAWST) modeling system, Ocean Modelling, 35, 230–244, 2010.

Weber, J. E. H., Broström, G., and Saetra, Ø.: Eulerian versus Lagrangian approaches to the wave-induced transport in the upper ocean, Journal of Physical oceanography, 36, 2106–2118, 2006.

Wu, L., Breivik, Ø., and Rutgersson, A.: Ocean-wave-atmosphere interaction processes in a fully coupled modeling system, Journal of
     Advances in Modeling Earth Systems, 11, 3852–3874, 2019a.

Wu, L., Staneva, J., Breivik, Ø., Rutgersson, A., Nurser, A. G., Clementi, E., and Madec, G.: Wave effects on coastal upwelling and water
     level, Ocean Modelling, 140, 101 405, 2019b.

WW3DG: User Manual and System Documentation of WAVEWATCH III® Version 6.07, Tech. rep., WAVEWATCH III® Development

Group, NOAA/NWS/NCEP/MMA College Park, MD, 2019.