# Peer review of "Waves in SKRIPS: WAVEWATCH III coupling implementation and a case study of cyclone Mekunu"

_EGUsphere, 2022_

## Author Comment (AC1)

Review Comments 1

General comments

In this manuscript the authors assess the performance and sensitivity to different parameterizations of a regional atmosphere-ocean-wave coupled model in simulating cyclone Mekunu in Arabian Sea. The authors first compare the performance of an atmosphere-ocean-wave fully coupled simulation, an atmosphere-ocean coupled simulation, and a standalone atmosphere simulation. They conclude that both versions of the coupled simulation give better results than the standalone atmosphere simulation. They further examine the sensitivity of the coupled simulation to different options of Langmuir turbulence parameterization and found that the simulation results including the mixed layer depth and sea surface temperature are sensitive to the choice of Langmuir turbulence parameterization. The authors also report the sensitivity of the simulation results to different choices of ocean surface roughness parameterization in the appendices.

In general this is an interesting study. The results are helpful for improving our understanding of the atmosphere-ocean-wave coupling during cyclones, and are useful for the development of regional atmosphere-ocean-wave coupled models. While the manuscript is easy to read, I think it can be significantly improved by a careful revision.

We thank the reviewer for acknowledging the usefulness of our work and providing comments that helped improve this manuscript.

One of my major concerns is that the focus of this study doesn't seem clear to me. If I understand it correctly, the focus of this study is to assess the effects of ocean surface waves by incorporating a wave model WaveWatch III into a regional atmosphere-ocean coupled model SKRIPS, using cyclone Mekunu as an example. If this is the case, the comparison with a standalone atmosphere model WRF seems to distract the readers from the focus. Also, coupled model has more skill in simulating cyclones than standalone atmosphere model may not be entirely new. I'd suggest the authors focus more on the impact of ocean surface waves by the coupling with a wave model. In this sense it would be better to examine in more detail what are the impact of including the effects of Stokes forces, Langmuir turbulence, wave modulated wind stress and ocean surface roughness seen by the atmosphere as introduced in Section 2 on simulating cyclone Mekunu. The presentation of the results in Section 4 is very brief and is not focusing on the effects of waves in my opinion, whereas Section 5 only discusses the impact of different options of Langmuir turbulence parameterization, which is only one of the wave effects included in this coupled model. So the section title of both sections are very confusing. In addition, the results of different sea state dependent surface roughness closures are presented only briefly in the appendices, which is also confusing to me why the authors choose to present these materials there.

Reply:

The goal of the paper is twofold: (1) demonstrate the implementation of the coupled ocean-wave-atmosphere model; (2) evaluate the impact of the surface waves to the coupled system. We have edited Section 1 to highlight our goals:

The goal of this work is twofold. First, we demonstrate the integration of the wave model WAVEWATCH III to the Scripps–KAUST Regional Integrated Prediction System (SKRIPS, Sun et al., 2019), which is a regional coupled ocean–atmosphere model that has been used to investigate extreme heat wave events on the shore of the Red Sea (Sun et al., 2019), North Pacific atmospheric rivers (Sun et al., 2021), and sea-ice evolution in the Southern Ocean (Cerovevcki et al., 2022). The second goal is to evaluate the implementations of ocean surface waves in the coupled system, especially for Langmuir turbulence that increases ocean mixing and cools down the SST during the simulation. Here, we perform a series of coupled and uncoupled numerical simulations of tropical cyclone Mekunu in the Arabian Sea.

We have also clarified the comparison with stand-alone WRF model in Section 4. By comparing with the uncoupled runs, we can highlight the changes due to air–sea coupling and the effect of the waves:

In this section, the ensemble coupled simulation results (i.e., CPL.AOW and CPL.AO) are compared with the results from the uncoupled runs (i.e., ATM.DYN) to assess the performance of the models, the impact of coupled feedbacks, and the effect of the waves.

Section 4 provides the comparison of simulations with and without all the effects of waves (e.g., Stokes forces, Langmuir turbulence, wind stress, and ocean surface roughness). In this section we showed that the effect of the surface waves does not signifitanctly impact the characteristics of the tropical cyclone in the simulation.

Furthermore, section 5 details the sensitivity analysis of Langmuir turbulence; Appendix C presents the sensitivity analysis of other effects of the waves. We did not present each component of the wave effects because their impact on the tropical cyclone characteristics is not significant, as shown in Fig. 3. The literature also suggests that the effect of Stokes forces could cancel each other in the coupled simulation (Suzuki and Fox-Kemper, 2016). Due to the chaotic nature of the atmosphere during the tropical cyclone event, the other effects of the wave model are not significant and thus we did not examine all the ocean/atmosphere variables obtained in the simulations.

Now we have revised the name of Section 4 to "Comparing coupled and uncoupled models". We have also added the comparison of the coupled simulation with/without the Stokes forces and the wind stress terms in Appendix C.

Another major comment is on the result of VR12-MA, one of the Langmuir turbulence parameterizations tested in this study. The authors found that using VR12-MA makes the simulated mixed layer depth shallower and sea surface temperature warmer in the cyclone wake than the simulation without Langmuir turbulence parameterization. This result is not intuitive as it is expected that Langmuir turbulence enhances the vertical mixing and deepens the mixed layer. The authors provide a possible explanation in Section 5.2 by examining the regionally averaged vertical profiles, which is very interesting. This may highlight a deficiency of KPP which uses a bulk Richardson number to determine the boundary layer depth, which might be sensitive to the structure of the velocity and buoyancy profiles. I'd suggest the authors to look closer to this issue, perhaps by plotting the time evolution of these profiles in Figure 11 at a point on the cyclone track and check

how these profiles change as the cyclone passes by. I guess VR12-MA would still give stronger deepening of the mixed layer depth during the cyclone, but the mixed layer depth may be shallower after the cyclone for reasons suggested by the authors.

Reply: Now we have attached the temporal evolution of these profiles (Fig 1 and 2 below). The vertical profiles of the Richarson number and velocity are similar throughout the tropical cyclone event and thus we only present the results on day 4 in our manuscript.

We have modified the manuscript to clarify the seemingly counterintuitive results from the VR12-MA simulations. When the Langmuir turbulence effects are considered in VR12-MA, the turbulent velocity scale $w_s$ increases and initially strengthens the parameterized diffusion of temperature and salinity. However it also increases the parameterized vertical viscosity and reduces the velocity shear. This reduces the $(U_r–U)$ term in Eq. (5), enlarges the Richardon number (which determines the parameterized mixing layer), and lessens the mixing and MLD in the model. These results show the importance of assessing the full parameterization when one aspect is changed, and this includes assessing how the shear is determined. Although using VR12-MA can initially improve the simulation of mixing layer depth and SST, it is also showing counterintuitive results may occur via feedbacks from the reduction in the shear (Li et al., 2016).

To illustrate the impact of the changes of velocity shear, we performed another experiment by not enhancing the diffusion of the velocity in KPP. This new option is shown in the figure below as VR12-MA-NoU. Still, the temperature and salinity diffusion are enhanced in this experiment. It can be seen in Fig. 3 below that the MLD deepens by 20 m and SST cools down by about 0.5 degree along the wake of the tropical cyclone. This is indicating that, when enhancing the diffusion of velocity in VR12-MA, it shoals the boundary layer and reduces the vertical mixing. Now we have revised our Section 5.2 to clarify our findings using VR12 and added Fig. 3 to Appendix D.

[Figure]

Fig. 1. The temporal evolution of the Richardson number profiles. The profiles on day 2 to day 5 are presented. The horizontal dashed lines are indicating the boundary layer depth in KPP.

[Figure]

Fig. 2. The temporal evolution of the horizontal current velocity profiles. The profiles on day 2 to day 5 are presented. The horizontal dashed lines are indicating the boundary layer depth in KPP.

[Figure]

[Figure]

Fig. 3. The snapshot of the ensemble averaged SST and MLD differences. Panels (a-c) show the SST difference between the simulations with Langmuir turbulence (CPL.LF17, CPL.VR12-MA, and CPL.VR12-MA-NoU) and without Langmuir turbulence (CPL.NoLT). Panels (d-e) show the differences in MLD. The markers indicate the regions where the SST difference is significant (P < 0.05).

I'd also appreciate it if the authors could provide more detailed discussion on the results. My impression is that the authors presented a lot of figures showing the results, but the corresponding description and discussion in the text are rather brief.

Thanks. Now we have revised our paper according to the reviewer's comments and added more discussions to Sections 4 and 5.

Specific comments

L5: Why comparing with a standalone atmosphere model? The difference would be dominated by the effect of including an active ocean model? Why not comparing with the coupled model without the wave component?

We compare with a standalone atmosphere model to demonstrate the improvement of the coupled model over uncoupled models. We also compared the effects of the waves in the coupled system. Now we have revised this sentence in the abstract to avoid confusion:

We examined the model skill in these simulations and further investigated the impact of Langmuir turbulence in the coupled system.

L9: Is Langmuir turbulence the only way through which the effects of waves are included? It might be helpful to mention what wave effects are included in the coupled model.

The Langmuir turbulence is not the only thing that we included. Now we added the other effects of ocean surface waves to the abstract:

In our implementations, we considered the effect of Stokes drift, Langmuir turbulence, sea surface roughness, and wave-induced momentum flux.

L22: "Intensity" -> "Intensity of TCs"?

This text has been corrected.

L37-38: Is Langmuir turbulence the only way impact of surface waves is implemented in this study? I know this becomes clear in section 2. But it would still be helpful to discuss at least why Langmuir turbulence is emphasized here.

We focused on Langmuir turbulence because the bias of the coupled model is alleviated when it is considered. According to the literature (Li et al., 2016; Reichl et al., 2016), in the coupled ocean–atmosphere model the SST and mixing layer depth are also sensitive to Langmuir turbulence. When the effect of Langmuir turbulence is considered, the mixing layer gets deeper and the SST cooling could be stronger by 0.5 to 0.7 degrees. Now we have revised this sentence in L37-38 and emphasized the Langmuir turbulence:

The second goal is to evaluate the implementations of ocean surface waves in the coupled system, especially for Langmuir turbulence that alleviates the model bias (Li et al., 2016). The coupled model is also sensitive to the implementation of Langmuir turbulence because it increases ocean mixing and cools down the SST during the simulation.

L68: What "surface boundary fields" are exchanged here?

The exchanged surface boundary fields are detailed in L76-86. Now we have re-organized Section 2 to avoid confusion:

The schematic description of the coupled model is shown in Fig. 1. We separated the WW3 main program into three subroutines that handle initialization, execution, and finalization. These subroutines are used by the ESMF/NUOPC coupler that controls the wave component in the coupled run. During the simulation, the surface boundary fields are exchanged via subroutine calls by the WW3–ESMF interface, shown in Fig. 1.

L70-71: Not sure what do the authors mean here... Why online regridding is not needed? If online regridding is not needed, why implementing it?

The online regridding process is implemented based on ESMF coupler. It is one of the advantages of this coupling framework. In this work, we aim to present the implementation of the wave model and thus not evaluate the online regridding process. Now we have revised this sentence in Section 2:

It is noted that ESMF online re-gridding options are also implemented for the wave component when exchanging boundary fields, but it is not used in this work because we aim to present the implementation of wave components.

L73-74: Might be helpful to be specific on what inputs and outputs are included…

Now we have reorganized this section and presented the input/outputs first. Please see our revisions in Section 2.1.

L81: By "Langmuir turbulence parameters" do the authors mean "Langmuir number"?

The Langmuir turbulence parameters include the Langmuir number and the turbulence enhancement factor, which are explained in Section 2.3. Now we have added the parameters that we used to parameterize Langmuir turbulence:

Langmuir turbulence parameters (i.e., Langmuir number and enhancement factor)

L99-100: So in addition to the surface Stokes drift mentioned on L80, the integrated Stokes transport is also needed to approximate the Stokes drift profile, right? Also, the same authors (Breivik et al) have an updated method to approximate the Stokes drift profile (Breivik et al., 2016), essentially requiring the same information from WW3. Any comments on why not using this newer method?

In this paper, we aim to demonstrate our implementations of the coupled system. To this end, we used the equations proposed by Breivik to approximate the profiles of Stokes velocity. We agree with the reviewer that there are other methods (e.g., Breivik et al., 2016 and Romero et al., 2021) that can better represent the Stokes velocity. Discussing the impact of Stokes drift profiles is out of the scope of this work, but we will test the other options for approximating the Stokes profiles in future work:

There are also other options to better approximate the Stokes velocity profiles (Breivik et al., 2016 and Romero et al., 2021) that remain to be tested in future work.

L117: Might be helpful to write out the equation here rather than referring the readers to an equation in Li et al., 2019?

Now we have added the missing equations to Appendix A.

L117: "using" -> "uses"?

This text has been corrected.

L119: The entrainment flux is also affected by the enhanced turbulent velocity scale, right?

Yes. We agree with the reviewer that the entrainment flux is also affected although it is not directly parameterized in VR12-MA. Now we have revised this sentence to avoid confusion:

Different from VR12-MA, LF17 parameterized the entrainment fluxes due to Langmuir turbulence by revising the definition of the bulk Richardson number:

L126: Same as above, might be helpful to write out the equation here?

Now we have added the equations to Appendix A.

L147-148: Which other models are used in this study? Was there a comparison of different options?

Yes. We have tested all these models for this case study, but the characteristics of the tropical cyclone do not change significantly. The surface variables (e.g., 10-m wind speed and latent heat fluxes) are not sensitive to these options either. We have presented the equations used in the surface roughness parameterization options in Appendix B and the sensitivity analysis in Appendix C. The sensitivity analysis of other effects of surface waves (e.g., Stokes-Advection, Stokes-Corolis, and wave-induced momentum flux) are also included in Appendix C.

L173-L185: Are the boundary conditions of atmosphere, ocean and waves consistent with each other?

In this case, we did not downscale from a global coupled model. In this sense, the boundary conditions for the atmosphere, ocean, and wave are not consistent with each other. This is done because we want to initialize our regional model using the reanalysis data to study the physical processes of air–sea interactions in a setting closest to the observed world. In addition, it is not straightforward to initialize the WAVEWATCH III model from the bulk wave parameters of a global coupled model. However, given that the reanalyses represent the observed world, they should be reasonably consistent with each other and in balance.

L193: What do the authors mean by "derive skill from boundary conditions"

In this work, we are performing downscale analysis and downscaling the ocean, wave, and atmosphere from the global models. In each ensemble, the lateral boundary conditions are provided by the same global products that allow us to investigate the air–sea interactions within the computational domain. Now we have revised this sentence to avoid confusion:

We performed downscaled hindcasts in this work, which allows us to focus on the impacts of air–sea interactions during the tropical cyclone event by minimizing the boundary errors.

L208-209: I didn't follow this sentence.

Now we have revised this paragraph in Section 3.2:

When the effects of surface waves are considered in CPL.AOW, the model setup is as follows. The Stokes-Coriolis and the Stokes-Advection in Eq. (1) are considered; the impact of Langmuir turbulence is parameterized in the same way as Li et al., (2017); the ocean surface roughness is determined using the Charnock coefficient (CHNK) from WW3; the wind stress in the ocean model is treated as mentioned in Eq. (7). We have compared the coupled model with and without wave effects in Section 4, then we further illustrate the sensitivity of the coupled model to the wave effects in Section 5 and the Appendix.

Section 4: The purpose of this section is a bit confused. If the purpose is to validate the simulation results of the coupled model (which seems to be suggested by the section title "Results"), more details and discussions on the comparison among the three sets of simulations (CPL.AOW, CPL.AO, and ATM.DYN) seem appropriate. If the purpose is to provide a background information for the discussion on the wave effects, the authors might need to be explicit on that.

Thanks. Now we have changed the title of this section to "comparing coupled and uncoupled models". We have also revised the first paragraph in Section 4 to clarify this:

In this section, the ensemble coupled simulation results (i.e., CPL.AOW and CPL.AO) are compared with the results from the uncoupled runs (i.e., ATM.DYN) to assess the performance of the models, the impact of coupled feedbacks, and the effect of the waves.

L250-251: Might be helpful to be specific on what is better and what is worse in CPL.AOW than CPL.AO.

Thanks. Now we have added more specific discussions to Section 4.1:

In summary, both CPL.AOW and CPL.AO runs better simulate the tropical cyclone characteristic than ATM.DYN in comparison with the IBTrACS data. CPL.AOW better simulates the minimum pressure and maximum wind speed than CPL.AO, but is outperformed by CPL.AO for the RMSEs throughout the event. CPL.AO also better simulates the track of the tropical cyclone.

L271: "Fig 5(b)" -> "Fig 5(c)"?

This text has been corrected.

Figure 4 caption: What do the black and red dots mean?

Now we have added the meaning of these dots in the caption:

The red dots indicate the ensemble-averaged locations of the center of tropical cyclones at the snapshot; the black dots indicate the ensemble-averaged locations of the center of tropical cyclones each day at 00 UTC.

L297: Switch the order of "cool the SST" and "deepen the MLD"?

Now we have switched the order of "cool the SST" and "deepen the MLD".

L299: Nudging to what?

We have added this sentence in the first paragraph of Section 5:

By using spectral nudging, WRF nudges the model fields to NCEP FNL data.

L300-301: Perhaps more reasoning of why nudging is necessary here deserves more clarification.

We used spectral nudging because of the uncertainties of the atmosphere model. Although we performed a series of 20-member ensemble simulations, the standard error of the SST and MLD is still very large due to the uncertainties of the cyclone tracks. By restraining the larger scale features of the atmosphere, we are able to reduce the uncertainty of the atmosphere model and can highlight the impact of the Langmuir turbulence on the ocean. Now we have added the discussion of this to Section 5:

To evaluate the effect of Langmuir turbulence on the ocean, we also performed the simulations using spectral nudging in WRF in addition to the ``free runs'' (simulations without spectral nudging). The spectral nudging is performed because of the uncertainties of the atmosphere model, especially for the tracks of the cyclones. By restraining the uncertainty of the atmosphere using spectral nudging, we are able to highlight the impact of Langmuir turbulence on the ocean.

L318-321 and L325-326: Do the authors mean that the vertical mixing of momentum is too much in VR12-MA, which reduces the vertical gradient of ocean current and reduces vertical mixing of tracers like temperature? It is not clear to me why an enhanced vertical mixing of momentum coexists with a reduction in vertical mixing of temperature. It might be helpful to elaborate on why this is the case.

When VR12-MA is applied in the coupled model, the KPP diffusion coefficient increases (multiplied by the enhancement factor in Eq. (3)). Then the horizontal diffusion increases and reduces the vertical gradient of ocean current. Because vertical current velocity is used in the (Ur – U) term Eq. (5) to determine the Richardson number, a smaller velocity gradient increases the Richardson number and shallows the ocean mixing layer in the model. When the ocean mixing layer is shallower, the vertical mixing of the ocean is reduced in VR12-MA and SST cooling becomes weaker. The vertical profiles shown in Fig. 11 aims to demonstrate the velocity profiles and current velocity to help explain this.

For the temperature differences shown in Figs. 9 and 10, we hypothesize that in this case ocean mixing layer gets shallower and thus reduces the vertical mixing of temperature. Now we have revised our manuscript and added more discussions on this:

When VR12-MA is applied, the Langmuir enhancement factor (see Eq. (3)) is used to amplify the KPP diffusivity term (see Eq. (5)). This reduces the vertical gradient of horizontal current, shown in Fig. 11(d).  When the velocity gradient is reduced in VR12-MA, the |Ur – U| term in Eq. (5) decreases, and thus the Richardson number increases. This Richardson number increase results in a decrease in estimated MLD, and thus the SST cooling is weaker than the simulation without Langmuir turbulence (NoLT).

Figure 9, 10: The results of VR12-MA are not intuitive to me. According to Section 2.3, VR12-MA also includes the effects of Langmuir turbulence on enhancing the vertical mixing. Then why the MLD gets shallower and SST gets warmer along the cyclone track than the case without Langmuir turbulence? Are the snapshot plotted at the time indicated by the red dot?

Yes. These results are plotted at the time indicated by the red dot. While we agree this is non-intuitive, we have diagnosed the processes taking place with the implementation of VR12-MA and gained understanding. Now we have revised the manuscript and clarified the cause:

To analyze the impact of different Langmuir turbulence options, in Fig. 11(a-d) we plotted the domain-averaged Richardson number, buoyancy difference, vertical density gradient, and velocity, which are the dominant terms in Eq. (6). The Richardson number is plotted because it is used in the MITgcm KPP scheme to determine the boundary layer depth, which is crucial to parameterize vertical mixing. The threshold of critical Richardson number is 0.3, meaning the ocean is assumed dynamically unstable and turbulent when Ri<0.3.

It can be seen that when VR12-MA is applied, the Richardson number increases compared with NoLT (no Langmuir turbulence), indicating the parameterized turbulence is getting weaker. Examining each component of the Richardson number in Eq. (6), it can be seen that buoyancy and vertical density gradient terms do not change significantly, shown in Fig. 11(b) and 11(c); while the changes of horizontal current speed can be seen in Fig. 11(d) near the surface.

When VR12-MA is applied, the Langmuir enhancement factor (see Eq. (3)) is used to amplify the KPP diffusivity term (see Eq. (5)). This reduces the vertical gradient of horizontal current, shown in Fig. 11(d).  When the velocity gradient is reduced in VR12-MA, the |Ur – U| term in Eq. (5) decreases, and thus the Richardson number increases. This Richardson number increase results in a decrease in estimated MLD, and thus the SST cooling is weaker than the simulation without Langmuir turbulence (NoLT).

As we mentioned in our previous reply, to illustrate the impact of the changes of velocity shear, we performed another experiment by not enhancing the diffusion of the velocity in KPP (i.e. leaving epsilon constant in Eq. 3). This new simulation is shown in the figure below as VR12-MA-NoU. In Fig. 1 of this reply, the MLD deepens by about 20 m and SST cools down by about 0.5 degree along the wake of the tropical cyclone. This is indicating that, when enhancing the diffusion of velocity in VR12-MA, it shoals the boundary layer and reduces the vertical mixing. We have added this figure and the discussions to Appendix D:

To verify this, we run an ensemble of simulations (CPL.VR12-MA-NoU) that do not enhance KPP diffusivity for horizontal currents, then we observed cooler SST due to enhanced mixing. The simulation results of this verification test are detailed in Appendix D.

Figure 11: Why the mixed layer depth (dashed lines) is different in panel (d) from other panels?

Now we have re-plotted Figure 11 and improved its quality.

References

Breivik, Ø., J.-R. Bidlot, and P. A. E. M. Janssen, 2016: A Stokes drift approximation based on the Phillips spectrum. Ocean Modelling, 100, 49–56, https://doi.org/10.1016/j.ocemod.2016.01.005.

---

## Author Comment (AC2)

Community Comments 1

This manuscript presents a coupled atmosphere-wave-ocean model for regional studies of cyclone development including options to simulate both 2-way atmosphere-ocean and 3-way atmosphere-wave-ocean configurations.  The model formulation and wave-coupling physics are briefly discussed and then a case-study is applied to analyze the impacts of the different model configurations.  The text is clear and the presentation/writing is of good quality. The topic is presently significant for the ocean and wave modeling communities. However, the manuscript does not presently provide compelling arguments for new advances, capabilities, and/or findings related to wave/ocean coupled simulations under cyclones beyond what has been demonstrated in previous studies on the topic.

One main result is that the role of ocean coupling improves the simulation, but this could be investigated further to better explain why the improvement is found (see Major Concern 1).  Another primary result is that Langmuir turbulence parameterizations can deepen the mixed layer and decrease the SST, but the effect of this on the coupled model is inconclusive and it is not clear if/how these conclusions would extend to other cyclone simulations.  Additional analysis into the other wave processes mentioned in the model description but not analyzed would also help clarify what is learned here and what should be considered for future studies.  I have several additional important technical concerns with the model and study, which are detailed below.  At this point I cannot recommend the article for publication and recommend substantial revision.

We appreciate the reviewer for those insightful comments. In this work, we aim to demonstrate our most recent technical developments of the SKRIPS coupled model. We have implemented the Langmuir turbulence parameterizations in MITgcm using the most recent version of WAVEWATCH III. In addition, we used ESMF coupler based on NUOPC Consortium, which is also a state-of-the-art tool for developing the coupled system.

In our case study, we focused on the air-sea interactions in the Arabian Sea, also including the Red Sea and the Arabian Gulf. This region is important because of its rich and diverse ecosystem, its economic impact on the surrounding countries, and its important role in international trade. In our work, we contrast methods for implementing Langmuir turbulence. We also document the counter-intuitive SST changes when using the VR12-MA option to parameterize the Langmuir turbulence in the sensitivity analysis. In the words of Reviewer 1, "The results are helpful for improving our understanding of the atmosphere-ocean-wave coupling during cyclones, and are useful for the development of regional atmosphere-ocean-wave coupled models."

Now we have extensively revised our manuscript to highlight the innovations in our work. Please refer to the changes in Sections 1 and 2.

Major Concerns

1. Adding ocean coupling to a cyclone model is expected to improve the cyclone simulation by improving SSTs, most clearly seen in cases where simulating ocean cooling in place of static SSTs reduces the cyclone intensity (e.g., Ginis & Bender, 2000: doi 10.1175/1520-

0493(2000)128<0917:RCSOHO>2.0.CO;2).  This study presents a downscaled model, however, and not a forecast model, so that what is called an "uncoupled" case here has time evolving SST with ocean cooling.  It is unclear from the given results whether the coupling has indeed improved SST relative to the observation data (drifters or any other sources) compared to the HYCOM assimilating product SST (it might be useful to look at biases in SST, similar to Figure 8).  This step and discussion would be useful to clarify that improved SST is indeed why the cyclone intensity is improved by coupling, and not, for example, other biases that are not ruled out by present analysis.

We have validated SST changes using in-situ observations in Section 5.1, but because of the lack of in-situ observations, only one drifter in the tropical cyclone wake zone is available for us to validate the SST from the simulations. The SST and MLD changes in the coupled model are also validated against HYCOM/NCODA data in Section 4.2.

We totally agree with the reviewer that we did not clarify why the cyclone intensity is improved in the coupled model. However, it is challenging to investigate the coupling processes during the tropical cyclone event. The simulation results are sensitive to the initial condition and the WRF physics options. The uncertainty of the model is also significantly large. In addition, due to the lack of in-situ observation, the SST obtained in the coupled simulation remains to be future work.

Now we have added the following text in our manuscript to clarify this:

It can be seen in Fig. 5 that the SST cooling in CPL.AOW is weaker than that in HYCOM, indicating the SST is warmer throughout the simulation in CPL.AOW. Contributed by the warmer SST, the intensity of the tropical cyclone is also stronger in CPL.AOW than ATM.DYN. Due to chaotic nature of the atmosphere, it is still unknown why the warmer SST in CPL.AOW improves the simulation of tropical cyclone characteristics. In addition, although we compared the simulation results using available data, the lack of in-situ observations in this region makes it challenging to validate the SST used the simulations.

2. The description of model physics and equations needs some clarifications in the text.  Some specific questions/issues:

2.1 The text states that the Stokes shear term in equation (1) is "parameterized" through Langmuir turbulence. L209 then implies that this term is dropped in the resolved scale model for this reason. However, there are scale separation issues here that are not discussed. Langmuir turbulence schemes are usually interpreted as representing the impacts of the WANS equations on turbulence at scales <~1km (usual LES domain, below the model resolved grid). Since the MITgcm model also solves the larger-scale "resolved" WANS equations including Stokes advection/Coriolis, including a Langmuir turbulence parameterization is not a formally consistent reason to drop the Stokes shear term in the model equations. At the relatively fine horizontal resolution of this model, the importance of the Stokes shear force should be more carefully considered and possibly retained to avoid changing the model's resolved momentum balance (see discussions in Suzuki & Fox-Kemper, 2016, for example).

We thank the reviewer for pointing out this issue in our manuscript. In our work, we implemented the governing equations based on Wu et al., 2019 that considered the Stokes-Advection and Stokes-Corolis terms, but did not include the effect of Langmuir turbulence. Although we do not explicitly resolve all the Stokes Shear force terms, it is worth noting that some of these effects might have been implicitly incorporated by tuning the parameterizations (e.g., KPP) using ocean observations. In our work, we only consider the contribution of Stokes shear forces to TKE through Langmuir turbulence parameterizations following the work from Li and Fox-Kemper (Li et al., 2016 and Li et al., 2017).

We acknowledge that a scale separation is critical for justifying dropping the horizontal Stokes shear terms in the coupled model. In our simulations the grid resolution is about 8 km (0.075 degree) and we verified that the horizontal shear of the Stokes drift is several orders of magnitude smaller than the vertical shear. In the study case that we present, the ocean model is still far from resolving submesocales, for which one could expect the horizontal and vertical components of the Stokes shear to be comparable.

Now we have clarified the Stokes shear force in Section 2.3:

Considering the impact of the surface waves, the Stokes drift provides a source of the turbulent kinetic energy~(TKE) through the vortex force and modified pressure (Craik et al., 1976), or more cleanly the Stokes shear force (Suzuki et al., 2016) as mentioned in Eq.~(1). Evidence of this enhanced vertical mixing has been documented from observations and large-eddy simulations. Although this effect is not explicitly accounted for in KPP (K-profile parameterization)~(Large et al., 1994), KPP might have implicitly incorporated some effects of Langmuir turbulence from tuning the parameters to ocean observations (Reichl et al., 2016). In the test case discussed here, we implemented the model at about 8 km resolution (0.075 degree), for which horizontal gradients of the Stokes drift are several orders of magnitude smaller than vertical gradients. Based on this scale separation, we only consider the effect of Stokes shear force through Langmuir turbulence parametrized in KPP (Suzuki and Fox-Kemper 2016)

Although there are many unknowns about the exact physics by which Langmuir mixing enhances entrainment, there are many options to parameterize the Stokes shear force in Eq.~(1) that could alleviate the model bias from the simulations. Within the KPP scheme, we implemented three Langmuir turbulence parameterizations: (1) VR12-MA; (2) LF17; (3) LF17-ST…

Section 3.2:

When the effects of the surface waves are considered in CPL.AOW, the model setup is as follows. The Stokes-Coriolis and the Stokes-Advection in Eq. (1) are considered; the impact of Langmuir turbulence is parameterized in the same way as Li et al., 2017.

2.2. Discussions in McWilliams et al. (2014, doi: 10.1175/JPO-D-13-0122.1) and Reichl et al. (2016b) suggest from Large Eddy Simulations that the KPP bulk Richardson number and the model's parameterized vertical momentum fluxes are improved by parameterizing with the Lagrangian current and Lagrangian current shear. I am concerned that separating between

Lagrangian/Eulerian currents in some parts of the model equations and not considering this difference in parameterizations could lead to inconsistencies in how the parameterizations are applied (e.g., this seems consistent with the explanation in the text for unintuitive results when using the VR12 parameterization, perhaps using the Lagrangian current would rectify this difference).

In our coupled model, MITgcm solves the Eulerian currents. The Stokes drift is added to terms on the right-hand side of Equation (1). We agree with the reviewer that using Lagrangian currents can alleviate the problem in VR12, but investigating the effect of Lagrangian currents is out of the scope of this model development paper. Now we have added the following discussions to the effect of Lagrangian currents in Section 5.2:

The drawbacks of VR12-MA are also discussed in Reichl et al., (2016b), where the authors show that using Lagrangian currents $u^L$ on Langmuir turbulence parametrizations can alleviate the bias when using VR12-MA.

2.3. Is the Stokes drift considered for the volume conservation equation? Wu et al. (2019) express it as a non-divergent condition on the Stokes drift vector, which presumably results in a vertical component of "Stokes drift" since the horizontal components of the Stokes drift can be divergent. It should be clarified how this is dealt with in this model.

In the coupled model, we did not consider the vertical component of Stokes drift. Hence our governing equations do not include Eq. (13) and Eq. (14) as Wu et al., 2019. Now we have clarified this in Section 2:

It is noted that our implementations and tests aim to demonstrate the impact of Langmuir turbulence on the ocean, and thus the divergence of the Stokes drift is not considered in our governing equations as discussed in Wu et al., (2019a, b).

2.4. It is not clear what is gained by including the "VR12" <w'w'> scaling as a parameterization in this study. The previous studies of LF17 (and also Reichl et al., 2016b) have shown that the ad-hoc assumption of applying <w'w'> based enhancements to the diffusivity and Vt2 in KPP are inadequate Langmuir parameterization approaches. Since the results of this scheme in the study are unintuitive, it might be best to drop this from the study (or clarify what specifically is learned by including it).

Yes, LF17 is developed based on VR12 with consideration of the entrainment. By demonstrating the differences between VR12 and LF17, we aim to illustrate the differences due to entrainment. In addition, VR12 has also been proven to be able to reduce the error in some case studies (e.g., Li et al., 2016). In our work, we documented the non-intuitive SST warming due to VR12 and discussed it.

We have revised Section 2.3 in our manuscript:

VR12-MA and LF17 are implemented because they are used in a variety of case studies and substantially improve the shallow biases of mixed layer depth (Li et al., 2016; Li et al., 2019). We aim to compare the performance of VR12-MA and LF17 to demonstrate the impact of entrainment on the simulations.

Section 5.1 is also revised:

Though it is demonstrated in Reichl et al., 2016 that VR12-MA is not adequate to parameterize the Langmuir turbulence, this non-intuitive SST change needs to be documented and discussed.

2.5. The inclusion of the LF17-ST parameterization model in the comparison is also not well motivated/discussed. Presumably the difference from the LF17 WW3 version can demonstrate what can be gained by including a wave model (sea-state dependence) for the turbulence scheme (an interesting topic), but that point is not motivated or discussed.

First, we include this option to validate our implementations of LF17 and VR12-MA. The option LF17-ST parameterizes the Langmuir turbulence in the same way as LF17, but using the surface wind to compute the enhancement factor. We have used the intermediate results from LF17-ST (implemented by Shultz et al., 2020) to validate our implementations of LF17 and VR12-MA. In addition, because SKRIPS can run with or without the wave component, LF17-ST can be used to parameterize the Langmuir turbulence in atmosphere-ocean coupled simulations without waves.

We have added the following discussion on LF17-ST in Section 2.3:

LF17-ST is implemented to validate LF17 in the coupled simulations. Because LF17-ST does not need waves, it can be also used in the coupled ocean-atmosphere simulations to parameterize the Langmuir turbulence.

We have also added the following text to discuss the differences of the simulation results in Section 5.1:

The results obtained using LF17 and LF17-ST are generally consistent, because they use a similar way to calculate the enhancement coefficient and entrainment flux. Their differences are because of different options to parameterize the Langmuir number La.

2.6. The implementation of the wave-budget terms and the Charnock coefficient in computing the wind stress is highlighted in the model formulation, but it is not discussed in the results. Some more analysis to clarify how including these terms impacts the simulations would be beneficial.

We have discussed the impact of sea surface roughness parameterization in Appendix C, but we found out that they do not have major impacts on the characteristics of the tropical cyclone. Although surface roughness is parameterized based on wave age and steepness, there are uncertainties in simulating the tropical cyclone and thus the surface variables (e.g., 10-m wind speed, latent heat fluxes) are not significantly different. Now we have added the discussions at the beginning of Section 5:

We also performed a similar sensitivity analysis for different surface roughness parameters that may impact the atmosphere surface variables. However, these results are summarized in the appendix C because they are not significantly different.

Minor Comments

1. The impression of the abstract is that the wave coupling improves the model, but this seems misleading. The "improvements" appear to come from coupling to the ocean model and the impacts of waves are less significant (but also note major concern 1).

We have revised this in our abstract:

We found that the coupled model better captures the minimum pressure and maximum wind speed compared with the stand-alone WRF model, although the characteristics of the tropical cyclone are not significantly different due to the effect of surface waves when using different parameterizations.

2. Regarding the wave momentum flux budget terms: How are the input and dissipation source terms parameterized for this study? Are these directly from WAVEWATCH? If so, clarify which source term packages are utilized and some discussion how these source terms are validated for cyclone wind speeds.

Yes. The input and dissipation source terms are directly from WAVEWATCH III based on Tolman 1995 and Ardhuin et al. 2003:

In the coupled model, tau_a is calculated in MITgcm (Large and Yeager, 2004) because WRF does not directly output the momentum flux terms. The parts that go into wave growth tau_aw and wave breaking tau_ow are calculated in WW3 (Tolman 1995; Ardhuin et al. 2003).

We have also added the setup of WAVEWATCH in Appendix C:

In this manuscript we used WAVEWATCH III version 6.0.7 compiled with the following switches:

F90 NOGRB NOPA LRB4 SCRIP SCRIPNC NC4 TRKNC DIST MPI PR3 UQ FLX0 LN1 ST4 STAB0 NL1 BT4 DB1 MLIM TR0 BS0 IC2 IS2 REF1 IG0 XX0 WNT2 WNX1 RWND CRT1 CRX1 TIDE O0 O1 O2 O2a O2b O2c O3 O4 O5 O6 O7

3. Regarding the use of HYCOM velocities to initialize the MITgcm model: Assuming the velocities are not somehow made dynamically consistent with the regridded hydrography, are any potential implications from the initial shock/adjustment times assessed?

Yes. There are potential impacts from initial shocks that are not assessed in this work. We used the HYCOM/NCODA data because we want to make the ocean state closer to the "truth". If we spin up the ocean model and assimilate the observation data, we still cannot completely get rid of the initial shocks. Now we have added the following discussion of the initial shocks in the revised manuscript:

To initialize the wave model, we allowed the wave field to spin-up for 19 days from May 01, 2018 and then we analyze the period from May 20, 2018. On the other hand, we did not spin up MITgcm or WRF, trying to initialize the coupled model using the analysis data directly. This may cause an initial shock in the coupled simulation, but we did not observe the initial shocks in the simulations.

4. Is the current passed to WW3 the same as the current passed to the atmosphere (e.g., Fig 1)? Presumably the atmosphere needs the surface current, but the current appropriate for WW3 is usually assumed at a depth related to the dominant wavelength (e.g., as in Fig 1 of Fan et al., 2009). Furthermore, the relative (and neutral) 10m wind should be used to drive WW3, adjusting for the surface current.  It is unclear from the text/diagram if this is done.

The same ocean surface current velocity is passed to WW3 and the atmosphere model WRF. We are using the current velocity in the first layer of the ocean model (z0 = 4 m) to represent the surface current velocity. It can be found in Fig. 11(d) that the vertical shear is relatively weak within the ocean boundary layer. This implementation of the current effects on waves is similar to COAWST but not consistent with Fan et al., 2009. We have modified Section 2 to clarify this point:

The surface current velocity sent to WRF and WW3 is consistent in our model, using the current velocity in the first layer of MITgcm. This may overestimate the strength of surface currents passed to WW3 compared with Fan et al., (2009) which used the currents at $L/4\pi$, where L is the mean wavelength.

For the surface wind, we did not change anything in the WW3 bulk formula, but replaced the WND forcing in WW3 by sending the 10-m wind speed (U10/V10) from WRF. In WW3 the relative wind (RWND) switch is used and thus the current velocity is considered for computing the wind stress. Now we have clarified what we did in our coupled model in Section 3:

The wind speed sent to WW3 and MITgcm is the 10-m wind speed, then WW3 and MITgcm correct the 10-m wind speed using the current velocity in the simulations.

We have also added the switch in WW3 to Appendix C:

In this manuscript we used WAVEWATCH III version 6.0.7 compiled with the following switches:

F90 NOGRB NOPA LRB4 SCRIP SCRIPNC NC4 TRKNC DIST MPI PR3 UQ FLX0 LN1 ST4 STAB0 NL1 BT4 DB1 MLIM TR0 BS0 IC2 IS2 REF1 IG0 XX0 WNT2 WNX1 RWND CRT1 CRX1 TIDE O0 O1 O2 O2a O2b O2c O3 O4 O5 O6 O7

5. The time derivative should be a partial and not material derivative in equations 1&2.

Now we have fixed the typo in the equations.

6. L185:  Why was a 19-day spin-up chosen for waves?  This seems excessive for a forced regional model including boundary conditions. It would be interesting to know if this integration time was deemed necessary.

We set up the spin-up according to the literature (Sabique et al., 2012; Boutin et al., 2021). Usually, the spin-up time takes 15 days, but we simply start the spin-up on the first day of the month (May 1st 2018) and make it a 19-day spin-up. The spin-up for the waves is necessary because we need to provide the initial and boundary conditions for the regional model.

Sabique, L., Annapurnaiah, K., Nair, T.B. and Srinivas, K., 2012. Contribution of Southern Indian Ocean swells on the wave heights in the Northern Indian Ocean–A modeling study. Ocean Engineering, 43, pp.113-120.

Boutin, G., Williams, T., Rampal, P., Olason, E. and Lique, C., 2021. Wave-sea-ice interactions in a brittle rheological framework. The Cryosphere, 15(1), pp.431-457.

7. L234: both -> all

Now we have fixed the text.

8. L270: HYCOM yields colder SSTs, but appears to yield shallower MLDs. Is the reason for this understood?

It is not straightforward to discuss the differences in SST and MLD from two different models. The models are different and the surface heat fluxes are different. HYCOM/NCODA also assimilates observations and makes the comparison more difficult. Because the goal of our paper is to present the coupled model, investigating the difference is out of our scope. We have added our hypothesis to the manuscript:

It is noted that CPL.AOW has stronger MLD deepening than HYCOM, but weaker SST cooling. We hypothesize that this is because (1) the parameterization of the ocean mixing layer is different when the effects of Langmuir turbulence are considered in CPL.AOW; (2) the atmosphere forcing used in the coupled model has a higher spatial and temporal resolution that makes the SST and MLD different.

9. L290: Are the beams physical or numerical?

The beams are due to the refraction of ocean surface currents. We have documented these beams in our previous paper using the uncoupled WAVEWATCH III model (Sun et al. 2022 paper: https://doi.org/10.1029/2021JC018112). The beams disappear when we turn off the ocean current forcing or directional shift in WAVEWATCH III model, and they are generally consistent with satellite altimetry data. Now we have added the text to clarify this in the manuscript:

The spatial pattern of high and low beams of Hs is due to surface wave refraction by ocean currents. We have performed uncoupled simulations to investigate these beams and more details can be found in Sun et al., 2022.

10. Figure 7: Panels are mislabeled in the caption.

Now we have revised the caption of this figure.

11. Figure 8: Missing units

Now we have revised this figure, added the missing units, and adjusted the location of the figures.

12. Figure 11 is poor quality, e.g., what is "drho/dr", what are units, etc.  Why not use the KPP boundary layer depth for mixing layer, rather than the mixed layer depth?

Now we have added the names and units of all the variables shown in the figure. We have also changed the style of the lines to improve the quality. Yes, we were presenting the KPP boundary layer depth in those figures, now we have clarified this.

13. WaveWatch should always be capitalized WAVEWATCH (as an acronym).

Now we have corrected this text throughout the manuscript.

---

## Author Comment (AC3)

Review Comments 2:

The manuscript presents a set of experiments aimed at demonstrating the impact of 1) coupling between the atmosphere/ocean, 2) further coupling the atmosphere/ocean model to a wave model and 3) various parameterizations of Langmuir turbulence using tropical cyclone Mekunu as a case study. The paper is clear, well written and provides a great level of detail on the various model formulation settings which is very useful for readers who want to explore similar experimentation.

The authors demonstrate significant improvement to the mean and RMSE values of cyclone central pressure, wind speed and latent heat fluxes through coupling. Though as the authors point out, the CPL.AOW model does not outperform the CPL.AO model. It would have been nice to see some discussion on why this might be. There are additional figures comparing the evolution of SST and MLD in the CPL.AOW simulation and that in the HYCOM analysis; however, there is little discussion of these figures. For instance, I'm surprised the HYCOM analysis has a great decrease in SST but smaller decrease in MLD.

The second section of "Results" (Section 5 for some reason) examines the impact different parameterizations of Langmuir turbulence have on SST and MLD. The authors demonstrate that the LF17 and LF17-ST experiments produce more accurate changes in SST with greater cooling and greater decreases in MLD relative to an experiment without Langmuir turbulence included. Interestingly, they also find that the VR12-MA experiment produces worse results than running without Langmuir turbulence. They attribute this fact to a reduction in turbulent shear and ocean mixing by the VR12-MA scheme.

Overall, the manuscript is straightforward, easy to follow and presents some interesting results that I feel many in the scientific community will find useful. I feel the manuscript only requires minor revision with perhaps a bit more discussion in the Results section and addressing the comments below.

We thank the reviewer for the comments that acknowledge our manuscript. Because this manuscript aims to demonstrate the development of the coupled model, we tried to focus on the technical development and demonstrate the capabilities of the coupled model. Now we have added more discussions on the physical insights and the SST trend in the sensitivity analysis in Sections 4 and 5.

Specific Comments:

Sort of a general note, but I'm surprised the first mention of using an ensemble comes on Line 195. I would think this would have been mentioned in the abstract or perhaps further up in the Methodology section.

Because we added random noise to each grid point, we put the setup of the 20-member ensemble simulation in Section 3.2 on model setups. The random noise is added simply to avoid digital precision error due to model internal variability. Now we have added the description of the ensemble experiment in the abstract:

Because of the chaotic nature of the atmosphere, we ran an ensemble of 20 members for each coupled and uncoupled experiment.

Section 2.3: It might be useful to have some description on why these three Langmuir parameterizations were selected. Especially since the Results section shows that the LF17 formulations are quite similar in their impact and the VR12-MA simulations are substantially different.

We tested these options because they are used in many global and regional models that reduce the error in simulating the mixing layer depth and SST (Li et al., 2016, 2017). Now we have added the description of the Langmuir turbulence parameterizations used in Section 2:

The three Langmuir turbulence parameterizations are selected because VR12-MA and LF17 have been shown to substantially improve shallow biases of mixed layer depth (Li et al., 2016, 2017, and 2019).

Line 196: "small random perturbations to the initial SST (<0.01 degC) at every grid point in the coupled model." Are they actually random or is there some amount of spatial/temporal correlation?

Now we have added the following sentence to the manuscript:

The random perturbations are added without any spatial or temporal correlation, aming to demonstrate the internal variability of the model.

Line 234-235: The language here does not reflect that there are three model being compared. Unless the purpose it to only discuss the coupled model simulations.

Now we have revised this sentence:

First, we examine the characteristics of cyclone Mekunu obtained from CPL.AOW, CPL.AO, and ATM.DYN to demonstrate the capability of the coupled model. The tracks of the tropical cyclone, defined by the positions of the low pressure center, are presented in Fig. 2, where it can be seen that all models can qualitatively match the observed evolution and track. Although the translation speed of the tropical cyclone from CPL.AOW is somewhat slower (CPL.AOW: 236 km/day; IBTrACS: 254 km/day), the distances between the cyclone centers for all model runs and IBTrACS data are less than 250 km until May 26, shown in Fig. 3(a).

Figure 3: Interesting that the models are somewhat indiscernible until ~day 3. Suppose this illustrates the time necessary for the ocean/waves to begin to have a meaningful influence on these metrics.

We started the simulations on May 20 and the model outputs are indiscernible for 5 days. In Figure 3 we did not show the indiscernible results from May 20 to 22. This is because the SST does not change significantly in the first few days. This is also consistent with our previous study in Sun et al., 2020 for atmospheric river events. Now we have added one sentence in the caption of the figure:

The simulations start on May 20, but the results are not presented before the pressure starts to drop on May 22.

Lines 261-263: Not sure what's being said here.

Now we have removed this sentence that causes confusion.

Line 288: Please describe why this figure shows that the wave height is sensitive to the wind speed.

Now we have added the contours of wind speed in this figure. The standard deviation of the wave height Hs is larger when the wind is strong. We have also revised our manuscript to clarify this:

Near the eye wall of the tropical cyclone, the standard deviation of Hs from the ensembles is approximately 3 m, showing greater variance near the eye wall (Fig. 7b),

Line 298: Would be useful to provide more description on why spectral nudging is necessary.

Now we have added more description on the spectral nudging:

To evaluate the effect of Langmuir turbulence on the ocean, we also performed the simulations using spectral nudging in WRF in addition to the ``free runs'' (simulations without spectral nudging). The spectral nudging is performed because of the uncertainties of the atmosphere model, especially for the tracks of the cyclones. By restraining the uncertainty of the atmosphere using spectral nudging, we are able to highlight the impact of the Langmuir turbulence on the ocean.

Figure 9/10: Perhaps I'm missing something, but in the figures with spectral nudging the red dot isn't centered on the largest SST/MLD differences. Should it be?

Yes. The largest changes are not centered on the red dots. We hypothesize that it is because (1) SST/MLD changes need some time to develop and (2) the winds on the right-front quadrant of the cyclone are strongest. Now we have added this to the manuscript:

It is noted that the largest SST and MLD changes are not centered on the location of the tropical cyclone. We hypothesize that this is because (1) the SST and MLD changes need some time to develop and (2) the winds on the right-front quadrant of the cyclone are strongest (Moon et al., 2014; Fan et al., 2009).

Lines 339-342: Having trouble connecting this phrase to figure 11. Perhaps demonstrate explicitly if it is seen in the figure

We have replotted Fig. 11 and improved its quality. We have also revised our discussions to the results shown in Fig. 11.

Technical Comments:

Line 41: "the Antarctica"?

We have replaced "the Antarctica" using "the Southern Ocean".

Line 167: Maybe just personal preference, but I would rephrase to "The ocean-atmosphere model is not coupled to the wave model". Same for #3 (Line 169).

Now we have rephrased it:

The ocean–atmosphere model is not coupled to the wave model, aiming to demonstrate the impact of the wave model on the simulation results.

The atmosphere model is not coupled to the wave or ocean models.

Line 176: Perhaps linearly interpolating "to" not "between"?

In MITgcm, we used the daily temperature, salinity, and current velocity to drive the coupled model. At each time step, the boundary conditions are interpolated between the HYCOM data before and after it. Now we have rephrased it:

At each time step, the boundary conditions for the ocean are updated by linearly interpolating between the daily HYCOM/NCODA analysis data.

Line 249: "Despite CPL.AOW better simulates…" Believe some words missing here.

Now we have removed this sentence to avoid confusion.

Line 267: Figures 5b, c. Not 5a.

Line 271: Fig. 5c. Not 5b.

Line 277: Fig. 6b, c. Not just Fig. 6b

Figure 7 Captions don't match the figure.

Thanks. Now we have fixed the typos with these figures. We have also gone through the manuscript carefully to check the figures.

---

## Author Comment (AC4)

Review Comments 3

The manuscript described the integration of the WW3 wave model into the SKRIPS couple framework. The cyclone Mekunu is chosen as study case in order to validate and compare a set of coupled and uncoupled simulations. Moreover, different Langmuir parametrizations have been implemented in order to investigate the impact of the wave to ocean coupling with an emphasis on SST and mixed layer depth analysis. Different surface roughness parametrizations are also compared to estimate the impact of waves in the coupled system. The simulations are validated against observations of the cyclone's characteristics and SST from drifters. Overall, the coupled simulations show better skill in reproducing the cyclonic event. The sensitivity to wave coupling on the characteristics of the cyclone is not found to be significant. However, the impact on SST and mixed layer is shown with a decrease of 0.5 degC and a deepening of the mixed layer of up to 20m in the wake of the cyclone compare to the simulation without Langmuir turbulence. Although, one of the parameterizations, VR12-MA, showed counterintuitive results with weaker SST cooling and shallower MLD. This is explained looking at vertical profiles showing a reduced horizontal shear velocity when using that specific scheme.

General comments:

Overall this is a good paper and well written. Although some of the results of the study are well known (i.e. coupled model generally have better skill in modeling cyclone events) I believe the emphasis is also put on the technical implementation of the wave model into the SKRIPS framework and of the different Langmuir and roughness parameterizations. For that reason, I believe it would be a good fit for GMD journal. However, one of the major issues in my opinion is that the manuscript needs more clarity on certain aspects and more discussion of the results which would be beneficial to the overall paper. As well as maybe a reorganization/clarification between sections 4 and 5 where the goals are sometimes unclear or confusing whether it aims to compare coupled and uncoupled, the impact of the different parametrizations or some validation of background ocean state. Some findings of the study are really interesting but often times too briefly discussed or even some figures not discussed at all. Also, as an optional comment the discussion on the appendices A and C on the roughness parametrizations may fit well in the main text.

We thank the reviewer for the comments that acknowledge our manuscript. This manuscript aims to demonstrate the development of the coupled model and we tried not to add many too discussions on the physical insights in our initial submission. Now we have added more discussions on the physical insights and the SST trend in the sensitivity analysis.

Hereafter are the details of the specific's comments.

Specific comments:

Line 80: It could be useful to reference the section 2.4 when mentioning the "momentum flux terms due to waves", it feels a bit too vague otherwise.

We thank the reviewer for pointing out this. Now we have revised Section 2.1 in the manuscript:

WW3 sends the wave variables to ESMF, including the (1) bulk wave parameters (i.e., significant wave height, peak wavelength and mean wavenumber), (2) Surface Stokes drift, (3) momentum flux terms due to surface waves, and (4) Langmuir turbulence parameters (i.e., Langmuir number and enhancement factor). The details of these wave variables are summarized in the latter sections.

Line 86: Maybe mention here that details of the calculation of the momentum stress is given later on, otherwise one can wonder why you're not mentioning it here.

Now we have added this to the end of Section 2.1:

The implementations of the wave effects are discussed in the latter sections. The surface Stokes drift forces, the Langmuir turbulence parameters, and the momentum fluxes are detailed in Sections 2.2, 2.3, and 2.4, respectively. The sea surface roughness parameterizations are summarized in Section 2.5.

Section 2.4: Mention that the momentum terms are output from WW3. Is the total air-side stress calculated inside MITgcm? If yes, why don't send the total air-side stress calculated within WRF which was calculated using wave information? And thus what kind of bulk formulae is used in MITgcm to calculate the air-side stress (line 83-84)? (similar remark can be made for heat flux). I think that a bit more clarity on how the surface flux are calculated and exchanged could be useful.

Yes. In the coupled model, we add the momentum terms from WW3 to those calculated from MITgcm. We mentioned that we are using the MITgcm momentum fluxes in our previous work (Sun et al. 2019). The heat fluxes and freshwater fluxes are calculated within WRF. We don't use the momentum fluxes in WRF directly because the WRF model does not directly output the momentum flux or friction velocity terms. Now we have added the following text to our manuscript to clarify this:

In the coupled model, tau_a is calculated in MITgcm (Large and Yeager, 2004) because WRF does not directly output the momentum flux terms. The parts that go into wave growth tau_aw and wave breaking tau_ow are calculated in WW3.

Line 147: In which scheme did you implemented these parametrizations ? Aren't some of these already available in WRF?

We implemented all these parameterizations for the MYNN scheme. They are not already available in WRF but are implemented in COAWST (Olabarrieta et al., 2012). Now we have clarified this in our manuscript:

We have also implemented a few other ocean roughness closure models that have been used in COAWST and discussed in Olabarrieta et al. (2012)...We implemented these options in the WRF Mellor--Yamada--Nakanishi--Niino (MYNN) surface layer scheme.

Line 147:150: Since you implemented several roughness parametrizations, any reasons why you didn't implemented COARE3.5 which is more recent and as also formulation of surface roughness based on sea state variables ?

We actually implemented everything available for COARE 3.5, but for the specific WRF version (v 4.1.3) we are using, COARE 3.0 is the default option and COARE 3.5 is commented out in the source code. Because of this, we used the default option COARE 3.0 in our experiments, aiming to represent our coupled model.

In Appendix A, line 373-374: please describe this options using the Charnock from WW3.

Now we have added the Charnock options that we used for WW3. We have also added the other physics options in WW3 to the manuscript:

In the present study, we used the ST4 option in WW3 and thus the Charnock coefficients are calculated based on Ardhuin et al. (2010). In this manuscript we used WAVEWATCH III version 6.0.7 compiled with the following switches:

F90 NOGRB NOPA LRB4 SCRIP SCRIPNC NC4 TRKNC DIST MPI PR3 UQ FLX0 LN1 ST4 STAB0 NL1 BT4 DB1 MLIM TR0 BS0 IC2 IS2 REF1 IG0 XX0 WNT2 WNX1 RWND CRT1 CRX1 TIDE O0 O1 O2 O2a O2b O2c O3 O4 O5 O6 O7

Table 1: Is the surface layer scheme used in WRF the same as PBL, MYNN?

Yes. We are using the MYNN scheme for the surface layer. Now we have added it to Table 1.

Line 207: Is a coupled frequency of 120 seconds necessary? An Hourly coupling or 30 min coupling would allow to capture the diurnal cycle as well, any reason behind the choice of the coupling frequency? Also wondering if the computation time is impacted by this high coupling frequency ?

The coupling process takes less than 5% of the total computational resources when using the ESMF coupler in our case (Sun et al., 2019, Table 4). In our previous work (Seo et al., 2014: https://doi.org/10.1175/JCLI-D-14-00141.1), we found that using a smaller coupling interval can better resolve the diurnal variation of SST. We used a 2-minute coupling frequency because ESMF allows us to exchange the air–sea fluxes without significantly increasing the computational costs.

Line 210-211: In line with my previous comment, I found it not clear how the surface stress is calculated. On Figure 1, WW3 send the surface roughness to WRF to get the stress but here it is mentioned that the Charnock coefficient is used. Is it actually the Charnock coefficient that is sent to WRF or the roughness length? Please clarify and modify Figure 1 if necessary.

We thank the reviewer for pointing out this in our figure and our text. In WRF the surface roughness height Z0 is used for the surface stress. In DGHQ, TY2001, and OOST, Z0 is calculated based on bulk wave parameters (e.g., significant wave height, peak wavelength, and mean wavenumber). When we use the Charnock coefficients from WW3 (e.g., CHNK), we replace the Charnock coefficients used to compute Z0 in WRF. WW3 is not sending the surface roughness length directly to WRF. We have revised this Figure 1 and revised our text in Sections 2.5 and 3.2.

Section 2.5:

When coupled with WW3 we parameterize the surface roughness based on the Charnock coefficient calculated from WW3 to make the surface roughness consistent. We have also implemented a few other ocean roughness closure models that have been used in COAWST: DGHQ (which is based on wave age), TY2001 (which is based on wave steepness), and OOST (which considers both the effects of wave age and steepness). These models parameterize Z0 using the bulk wave parameters from WW3. More detailed descriptions of these closure models and sensitivity analysis are presented in Appendices B and C.

Section 3.2:

When the effects of the surface waves are considered in CPL.AOW, the model setup is as follows. The Langmuir turbulence is parameterized in the same way as Li et al., 2017; the Stokes-Coriolis and the Stokes-Advection in Eq. (1) are considered; the ocean surface roughness is parameterized using the Charnock coefficient (CHNK) from WW3…

Section 3.2: What are the models frequency outputs (if different from one model to another)? It would be useful to add it here.

The models output the simulation results every three hours. In this work we set the same output frequency, but the output interval is flexible when using the SKRIPS model. Users can output the simulation results at different intervals for each model component. We have added the model output interval in Section 3.2:

We output the results every three hours to demonstrate the evolution of the tropical cyclone simulated by the coupled model.

Section 4: The name of the section, "Results", is fairly generic. The goal of this section needs some clarification, it includes some results between coupled simulation and some more validation part, i.e. section 4.2 or 4.3 which resemble more to a validation of the wave field and no impact of waves on air-sea interaction or between simulations are discussed yet.

Section 4 provides the comparison of simulations with and without all the effects of waves (e.g., Stokes forces, Langmuir turbulence, wind stress, and ocean surface roughness). In this section we showed that the effect of the surface waves does not signifitanctly impact the characteristics of the tropical cyclone in the simulation. Section 5 details the sensitivity analysis of Langmuir turbulence; Appendix C presents the sensitivity analysis of other effects of the waves.

Due to the chaotic nature of the atmosphere during the tropical cyclone event, the other effects of the wave model are not significant and thus we did not present the impact all the ocean/atmosphere variables obtained in main text.

Now we have revised the name of Section 4 to "Comparing coupled and uncoupled models". We have also added the comparison of the coupled simulation with/without the Stokes forces and the wind stress terms in Appendix C.

Line 250: Any hypothesis as why the CPL.AOW would give "worse" results than the CPL.AO ?

We agree that the CPL.AOW does not show better results compared with CPL.AO. The two sets of coupled runs show similar results and we have examined the contribution of different components of wave effect in Section 5 and the appendix. It is maybe because the CPL.AOW considers the effect of Langmuir turbulence that strengthens the SST cooling, and thus reduces the intensity of the cyclone. But generally speaking the differences between CPL.AOW and CPL.AO are smaller than the standard deviation of the results . Now we have added more specific discussions to the revised manuscript:

In summary, both CPL.AOW and CPL.AO runs better simulate the tropical cyclone characteristic than ATM.DYN in comparison with the IBTrACS data. CPL.AOW better simulates the minimum pressure and maximum wind speed than CPL.AO, but is outperformed by CPL.AO for the RMSEs throughout the event. CPL.AO also better simulates the track of the tropical cyclone… The differences between the tropical cyclones simulated in the coupled and uncoupled simulations are associated with the SST cooling in the simulations, which are further discussed in Section 4.2. The differences between CPL.AOW and CPL.AO are further investigated in Section 5.

Line 271: "Fig. 5(c)"

Now we have corrected this text.

Line 271: Is this a known cold bias from HYCOM? Any reason as why HYCOM would show stronger SST cooling?

The bias from HYCOM/NCODA data is unknown in this region and this specific case due to the lack of in-situ observations, although it has small bias compared with other observational data. In this case study the SST cooling in HYCOM is stronger than the coupled model, and thus the simulation results of the tropical cyclone using this colder SST will be different. This section aims to explain the difference between CPL.AOW and ATM.DYN in simulating the tropical cyclone. We have revised our manuscript here to clarify this:

It can be seen in Fig. 5 that the SST cooling in CPL.AOW is weaker than that in HYCOM, indicating the SST is warmer throughout the simulation in CPL.AOW. Contributed by the warmer SST, the intensity of the tropical cyclone is also stronger in CPL.AOW than ATM.DYN. Due to chaotic nature of the atmosphere, it is still unknown why the warmer SST in CPL.AOW improves the simulation of tropical cyclone characteristics. In addition, although we compared the simulation results using available data, the lack of in-situ observations in this region makes it challenging to validate the SST used the simulations.

Line 272:273: Does this mean it is in agreement with some observations? Please clarify this statement.

For the results obtained for the Arabian Gulf and the Gulf of Aden, we only compared our simulation results with the HYCOM/NCODA data, but not with the in-situ observations. Now we have revised this sentence to avoid confusion:

It should be noted that CPL.AOW also captures the SST warming in the Arabian Gulf and the Gulf of Aden compared with the HYCOM analysis data.

Section 4.2: the results of the MLD differences in HYCOM showed in Figure 6c are not discussed, such as any hypothesis as why HYCOM would show shallower MLD while stronger cooling. Overall this section could benefit from more explanation of the figures and results showed.

It is not straightforward to discuss the differences in SST and MLD from two different models. The models are different and the surface heat fluxes are different. HYCOM/NCODA also assimilates observations and makes the comparison more difficult. Because the goal of our paper is to present the coupled model, investigating the difference is out of our scope. We have added our hypothesis to the manuscript:

It is noted that CPL.AOW has stronger MLD deepening than HYCOM, but weaker SST cooling. We hypothesize that this is because (1) the parameterization of the ocean mixing layer is different when the effects of Langmuir turbulence are considered in CPL.AOW; (2) the atmosphere forcing used in the coupled model has a higher spatial and temporal resolution that makes the SST and MLD different.

Figure 7: Caption does not match the actual figure, please clarify.

Now we have fixed the caption for Fig. 7:

Snapshots of the significant wave height Hs at 00 UTC May 22 24, and 26, 2018. Panel (a-c) show the ensemble averaged Hs obtained from CPL.AOW; Panel (d-f) show the standard deviation of Hs of the ensembles from CPL.AOW. The 15~m/s contour of wind speed is used to highlight the location of the tropical cyclone.

Line 284: You probably meant Figure 7b or Figure 7c.

Now we have corrected this in our manuscript. We have also gone through the manuscript and fixed all issues with the index of the figures.

Line 291: Please precise Appendix C.

Now it is corrected.

Line 304: Please precise Appendix B. Also, in the text Appendix C is cited before Appendix B so maybe the order of the appendices could be revised.

We have have fixed this in the manuscript. We have also revised the appendices and changed their order.

Line 389: "which are"

We have revised this paragraph and removed this sentence.

Line 391: Please be more explicit than "CPL" here as on the figure they all have different names and none is CPL

Now we have replaced "CPL" using "CPL.AOW (CHNK)".

Line 395-396: Please specify the region, as in the Figure A4 it looks like there are some regions where the wind speed and the latent heat flux are weaker compared to CPL.CHAR.

Now we have revised this sentence:

It can be seen that the 10-m wind speed and latent heat loss are different when using TY2001, DGHQ, and OOST in comparison with using CHNK in CPL.AOW. However, the differences are not significant from the t-test (regions with $P < 0.05$ are highlighted), and analyzing these differences remains to be a future work.

Figure A4: "parameterization"; Also what do you mean by "without parameterization", isn't CPL.CHAR parameterized using the Charnock coefficient from WW3 ?

We have revised the caption of Fig. A4:

Panels (a-c) show the 10-m wind speed difference between the simulations using different options to parameterize the surface roughness (TY2001, DGHQ, and OOST) compared with CPL.AOW (CHNK).

Section 5.1: The Figure 8c showing HYCOM SST compared to drifters is not discussed although it looks like the cooling along the track is the closest compare to the drifters, is that right? However, in Section 4.2 it seems like the SST cooling in HYCOM was too strong which was one of the reasons leading to a too low intensity of the cyclone simulated ATM.DYN, please clarify or comment on that. Here again, this is interesting results and a bit more discussion on these findings could be beneficial.

Yes. The HYCOM/NCODA SST under the cyclone track is closer to the drifters than the simulations. We have revised Section 4.2 (shown in our previous replies) and added the following text in Section 5.1 to clarify this:

It is noted that the SST cooling in HYCOM/NCODA data is 3.23 degrees, which is closer to the drifter data than the coupled simulations. However, because the in-situ observations are few in this region, future work still needs to be done on investigating the performance of the coupled model.

Figure 11: Looks like the mixed layer depths (dashed lines) are different in panel (d) than the others, please correct/clarify this.

Now we have replotted Figure 11 and fixed this.

---

## Author Response (AR2)

Reviewer 1:

General comment:
The authors addressed well my comments in the first round of revisions. The text in the manuscript have also been improved. I still believe that this paper would be a good fit for GMD. I only have one minor concern that rose from one of the author's responses.

Minor comments:
You mentioned the different switches used to compile WW3, one of them concerns the correction of the wind using the ocean current (RWND). I assume the wind in WRF to be the relative wind since ocean currents are prescribed from the ocean model, so the WRF output should be relative wind as well. Thus, the wind sent to WW3 should already be the relative wind, yet by using this switch in WW3 you would correct the wind again, wouldn't that be double counting the ocean current to correct the wind, first in WRF and a second time in WW3? This remark could apply also to the MITgcm as at line 97 you mentioned the wind is corrected in the ocean model as well. Could you clarify this point?

Reply 1: We thank the reviewer for pointing out this issue in our work. We have carefully examined the WRF code and literature, then we found that WRF outputs the relative 10-m wind speed. We have re-compiled the code (removed the RWND switch in WW3) and re-ran all the simulations. Now we have revised the manuscript to clarify this:

The wind speed sent to WW3 and MITgcm is the relative 10-m wind speed from WRF based on the Monin-Obukhov similarity theory (Monin and Obukhov, 1954; Renault et al., 2020), then WW3 and MITgcm use the relative 10-m wind speed without correcting the current velocity in the simulations.

Due to the minor difference between the old and new simulation results, we have also updated all figures and some texts describing the simulation results. For example, we moved the comparison with drifter data to the appendix because the SST differences are not significantly different at the location of the drifters (and they were not significantly different from our previous version). Please refer to our annotated manuscript.

Reviewer 2:

I think this revised version of the manuscript has been improved in clarity as compared to the original version. However, I'm still not quite satisfied with the authors' response to my two major concerns in my previous review.

1. I still feel that the inclusion of the uncoupled WRF run in the comparison is a bit confusing and distracting. Since the focus is on the effects of including waves in SKRIPS (based on the title, abstract and the two goals described in the introduction), it may make more sense to start from the atmosphere-ocean coupled version of SKRIPS. The standalone WRF simulation may be used as a reference. But it would be good to focus on the comparison between CPL.AO and CPL.AOW in the main results and conclusions. If the authors also want to highlight the changes due to air-sea coupling (as mentioned in their response to my previous comments), it might be helpful to discuss in more detail what is the role of air-sea coupling here. But I think such discussion may be quite distracting.

Reply 2: We thank the reviewer for pointing this out. In our work, we implemented the coupled model with wave components. Natural experiments to examine the newly implemented model are to compare it both with coupled models without waves and also to uncoupled models. This is typical in other studies demonstrating coupled models with newly implemented wave components (e.g., Warner et al., 2010; Chen et al., 2013). The uncoupled model serves as a useful benchmark. We have revised our manuscript to clarify this, as detailed in our Reply 5.

2. The added explanation of the unintuitive results of VR12-MA in Section 5.2 is not satisfactory. In fact, the argument of reduced velocity shear due to enhanced diffusivity of momentum also applies to LF17 -- the same Langmuir enhancement factor in VR12-MA is applied in LF17 as well. So the increased bulk Richardson number due to reduced |u_r-u|^2 should also occur in LF17 -- indeed in Fig 11(d) the same reduction in surface velocity shear can be seen for LF17 too.

Reply 3: We agree with the reviewer that the same arguments apply to LF17 for the |Ur – U| term. However, in Eqs. (7) and (8), the Richardson number in LF17 included the $V_t(z)$ term to account for the effects of entrainment flux. When the entrainment flux is considered in LF17, it enhances the mixing and reduces the Richardson number, shown in Fig. 11(a). Although the near-surface velocity is also reduced in LF17, the mixing layer deepens and SST gets colder than in VR12-MA and NoLT. Now we have revised our discussion in Section 5 to clarify this:

On the other hand, when using LF17 in the simulations, the same enhancement factor as VR12-MA is added, but the term $V_t(z)$ is used in Eq. (8) for parameterizing the Richardson number. Although the velocity gradient |Ur – U| is also smaller, shown in Fig. 11(d), the entrainment flux $V_t(z)$ decreases the Richardson number. This implies stronger vertical mixing due to the Langmuir entrainment by the tropical cyclone. Hence the SST cooling in the near wake region of the tropical cyclone is stronger when LF17 is used than VR12-MA. This shows parameterizing the Langmuir turbulence using LF17 gives more realistic results than VR12-MA.

I think the suggestion from the Community Comments 1 about the inconsistency of the Lagrangian versus Eulerian currents in the momentum equation and Langmuir turbulence parameterization may explain this unintuitive result of VR12-MA. In both Li et al., 2016 and Li et al., 2019, VR12-MA (also LF17) was not used together with Stokes Coriolis. Their assumption (see the notes in Table 1 of Li et al., 2019) was that the simulated velocity is Lagrangian so implicitly it is the Lagrangian shear that is used to compute |u_r-u|^2 in KPP. Since the vertical shear of Stokes drift is strongest near the surface and often roughly aligns with the wind, the Eulerian velocity shear is much reduced when Stokes-Coriolis force is included in the momentum equation. This is consistent with what the authors saw in their simulations in Fig. 11(d) with reduced surface velocity shear in both VR12-MA and LF17 as compared to NoLT. I'd suggest the authors run two more simulations without Stokes-Coriolis and Stokes-advection terms (i.e., only Langmuir turbulence parameterization of VR12-MA and LF17). I guess one may see deeper MLD and cooler SST along the cyclone track in both cases, but more MLD deepening and SST cooling in LF17 than shown here.

Reply 4: Now we have added the experiments using NoLT, VR12-MA, LF17, and LF17-ST without using Stokes-Coriolis and Stokes-advection in Appendix C. This is consistent with the implicit option when simulating coupled ocean-wave-atmosphere interactions in Li et al., 2016 paper. We have found that the simulation results do not change significantly, as shown in Fig. 1 below. It can be seen that the SST changes are similar to what is shown in Fig. 8 of the manuscript, except for the region near the track of the tropical cyclone where the uncertainty is

large. For Panels (d-f) when spectral nudging is applied, similar patterns of SST cooling and warming are captured when the Stokes forces are implicitly considered.

[Figure]

[Figure]

Figure 1. The snapshot of the ensemble-averaged SST and MLD difference. Panels (a-c) show the SST difference between the simulations with Langmuir turbulence (CPL.LF17.IMP, CPL.VR12-MA.IMP, CPL.LF17-ST.IMP) and without Langmuir turbulence (CPL.NoLT.IMP). Panels (d-e) show the SST difference for the simulations with spectral nudging. The markers indicate the regions where the SST difference is significant (P < 0.05).

However, the investigation of potential inconsistencies between Eulerian and Lagrangian implementations of the numerics is beyond the scope of this paper. This paper aims to present the implementations of the coupled model and we offer two options: (1) explicit scheme for Stokes forces (similar to Wu et al., 2019); and (2) implicit scheme (similar to Li et al., 2016). We summarized the simulation results using the implicit option in Appendix C and added the following sentence in our manuscript in Section 3.2:

It is noted that when the Stokes-Coriolis and the Stokes-Advection are not explicitly considered in the experiments, the model setups are consistent with Li et al. (2016), assuming the simulated velocity is Lagrangian.

Specific comments:

L8-10: Given that the focus of this study is on including waves in SKRIPS (suggested by the title), I still feel it is not necessary and is really confusing to put any emphasize on the difference between coupled model and standalone WRF. Perhaps rephrase to use the difference between the stand-alone WRF and ocean-atmosphere coupled simulations as a reference to describe how large the impact of including surface waves in these experiment is?

Reply 5: In this work, we developed the coupled model with wave components, then we evaluate the model skill of the coupled model without wave and the uncoupled model. Now we have revised this sentence in the abstract by emphasizing the uncoupled model is used as a benchmark:

We found that the characteristics of the tropical cyclone are not significantly different due to the effect of surface waves when using different parameterizations, but the coupled models better capture the minimum pressure and maximum wind speed compared with the benchmark stand-alone WRF model.

We have also revised the description of the stand-alone simulation in Section 3.2:

Compared with CPL.AO and CPL.AOW, this run serves as a benchmark that aims to demonstrate the impact of waves and coupled air–sea interactions on the simulation results.

L49-54: These discussions seem to divert the purpose of this manuscript to test the coupled model? I think it is already widely accepted that accurately modeling tropical cyclones is important but challenging.

Reply 6: These discussions aim to explain why we select the Arabian Sea in our case. Now we have revised it:

The Arabian Sea is investigated in this work because of its rich and diverse ecosystem, its economic impact on the surrounding countries, and its important role in international trade. Continued climate warming is expected to further amplify the risk of cyclones in the Arabian Sea (Dube et al., 1997; Evan et al., 2011; Evan and Camargo, 2011) and increase socio-economic implications for coastal communities in that region (Henderson-Sellers et al., 1998; Murakami et al., 2017; Bhatia et al., 2018).

L80-84: I understand that the added text here is to address one of the reviewer's comments. But it seems a bit confusing and distracting. Perhaps rephrase? For example, by "overestimate the strength of surface currents" on L81, do the authors mean overestimating the currents that are dynamically important for waves in WW3, which shouldn't be the surface currents?

Reply 7: Our implementations of the currents are consistent with the COAWST model (Warner et al., 2008, 2010), but not consistent with the coupled model used by Fan et al. (2009). Therefore, we tried to clarify this in our manuscript. Now we have revised this sentence:

We used the surface current based on previous literature (Warner et al., 2008, 2010, Couvelard et al., 2020), but this may overestimate the strength of surface currents impacting the wave model, as suggested by Fan et al., (2009), who used the current velocity at $L/4\pi$ (L is the mean wavelength).

L89-90: What do the authors mean by "surface Stokes drift forces"?

Reply 8: Now we have fixed the typo in this sentence:

The Stokes forces, the Langmuir turbulence parameters, and the momentum fluxes are detailed in Sections 2.2, 2.3, and 2.4, respectively.

L95: What do the authors mean by "surface boundary fields"?

Reply 9: The surface boundary fields are mentioned in the first and second paragraphs in the same section (Line 73-80). Fig 1 also shows the surface boundary fields sent/received by WW3. Now we revised this sentence:

During the simulation, WW3 receives and sends boundary fields via subroutine calls by the WW3--ESMF interface, shown in Fig. 1.

L100-102: So the three components are all on the same grid? Perhaps mention briefly the possible uses in the future?

Reply 10: Yes. We are currently working on a higher-resolution ocean model with our collaborators. The possible uses in the future are added:
The online re-gridding option will be used when using a higher resolution ocean model for the Arabian Sea operational model.

L116: By using Breivik et al., 2014, it means the total Stokes transport (vertically integrated Stokes drift) is also needed to be passed from WW3 to MITgcm to reconstruct the Stokes drift profile, not only the surface Stokes drift as mentioned on L77?

Reply 11: In the simulations we only used the surface Stokes drift. The total Stokes transport is not sent to MITgcm because we sent the wavenumber $k_m$ in Eq. (8) to compute the Stoke drift profiles.

L115-121: It is not entirely clear from this discussion why approximation is necessary for the Stokes drift profile as it can be computed in WW3. The problem is passing the Stokes drift profile from WW3 to MITgcm, right?

Reply 12: The Stokes drift profile can be computed in WW3, but it is not the standard output. WW3 only outputs TUS (Stokes volume transport) and USS (surface Stokes drift) and therefore we used the Breivik et al. (2014) to approximate the Stokes drift in the coupled simulation. This is similar to the recent work performed by Couvelard et al. (2020).

Theoretically, ESMF can send 3D Stokes drift profiles from WW3 to MITgcm, but it may also significantly increase the computational cost in the coupling process.

L127-132: I didn't follow the reasoning here…

Reply 13: In this paragraph, we tried to explain why we revised KPP in our work. KPP does not explicitly consider the effect of Stokes shear force in Eq. (1), but it is tuned from observation data and may implicity incorporate some effects of Langmuir turbulence. Because of this, Li et

al. (2016) and many others tries to modify the KPP parameterization to reduce the error due to the effect of Langmuir turbulence.

Now we have revised this paragraph to clarify this:

Considering the effect of the surface waves, the Stokes drift provides a source of the turbulent kinetic energy (TKE) through the vortex force and modified pressure (Craik and Leibovich, 1976), or more cleanly the Stokes shear force (Suzuki and Fox-Kemper, 2016) as mentioned in Eq. (1). Evidence of this enhanced vertical mixing has been documented from observations and large-eddy simulations (McWilliams et al., 1998; D'Asaro, 2001; Van Roekel et al., 2012). In this work, we aim to implement the Stokes shear force in the coupled model and investigate its effect on the coupled system. Although it is not explicitly accounted for in KPP (K-profile parameterization, Large et al., 1994), KPP might have implicitly incorporated some effects by tuning the parameters to ocean observations (Reichl et al., 2016). Our implemented model is about 8 km resolution (0.075 deg), and the horizontal gradients of the Stokes drift are several orders of magnitude smaller than vertical gradients. Following Suzuki and Fox-Kemper (2016) we only consider the effects of Stokes shear force due to Langmuir turbulence because of this scale separation.

L133-134: Didn't follow this sentence as well…

Reply 14: This sentence is revised:

Although there are many unknowns about the role of Langmuir mixing in ocean modeling, there exists many parameterizations that aim to represent these processes and alleviate model bias.

L137: It is not very clear what "based on the waves" refer to?

Reply 15: We add this sentence to demonstrate the differences between VR12-MA/LF17 and LF17-ST. The first two options (VR12-MA/LF17) are parameterized based on wave state; the last option (LF17-ST) is parameterized based on the surface winds. The specific parameters used to parameterize the Langmuir turbulence are detailed in the latter paragraphs.

Now we have revised this paragraph to clarify the differences between VR12-MA/LF17 and LF17-ST:

Both VR12-MA and LF17 parameterize the Langmuir turbulence based on the parameters computed from WW3: in VR12-MA the KPP turbulent velocity scale is multiplied by an enhancement factor; in LF17 the KPP turbulent velocity scale is treated in the same way as VR12-MA, and the entrainment buoyancy flux is also considered. On the other hand, LF17-ST parameterizes the Langmuir turbulence similarly to LF17, but parameters are computed using the 10-m winds instead of using the output from WW3.

L143-144: Not clear why this case is necessary to validate LF17

Reply 16: First we use LF17-ST because it has been implemented and validated in MITgcm by Schultz et al. (2020). In comparison with LF17-ST, LF17 uses a similar approach to determine the critical Richardson number in KPP, but uses different physics inputs from surface winds. Because of the similarities between the two options and the work done by Schultz et al. (2020),

we decided to validate LF17 using the implemented LF17-ST option. Now we have revised this sentence:

We also used the well-validated LF17-ST implementation by Schultz et al. (2020) to validate LF17 in the coupled simulations due to the similarity of these two options.

L144-145: True only if it works…

Reply 17: This LF17-ST option has been implemented in MITgcm (Schultz et al., 2020) to investigate the Langmuir turbulence. In our work, we inherited their code but did not run any individual tests only using LF17-ST. Now we have made some minor changes to this sentence:

Because LF17-ST does not need bulk wave parameters as input, it can be also used in uncoupled MITgcm simulations (Schultz et al., 2020) or coupled simulations without waves to parameterize the Langmuir turbulence.

L186: Not the appropriate citation for computing the wind stress?

Reply 18: The equations and parameters to compute the wind stress in MITgcm follow Eq. (6a) in Large and Yeager, 2004. The formulation was from the unpublished work from E.E. Vera in 1983, but this document is not citable.

L417-421: These same arguments apply to LF17 as well -- in LF17 the same Langmuir enhancement factor is applied to amplify the KPP diffusivity term. In fact, the same reduction in near surface velocity shear can be seen for LF17 in Fig. 11(d).

Reply 19: We have replied to this comment in our Reply 3 and 4. Now I am attaching a part of our reply 3 for this specific comment.

We agree with the reviewer that the same arguments apply to LF17 for the $|U_r – U|$ term. However, in Eqs. (7) and (8), the Richardson number in LF17 includes the $V_t(z)$ term to account for the effects of entrainment flux. When the entrainment flux is considered in LF17, it enhances the mixing and reduces the Richardson number, shown in Fig. 11(a). Although the near-surface velocity is also reduced in LF17, the mixing layer deepens and SST gets colder than in VR12-MA and NoLT. Now we have revised our discussion in Section 5 to clarify this:

On the other hand, when using LF17 in the simulations, the same enhancement factor as VR12-MA is added, but the term $V_t(z)$ is used in Eq. (8) for parameterizing the Richardson number. Although the velocity gradient $|U_r – U|$ is also smaller, shown in Fig. 11(d), the entrainment flux $V_t(z)$ decreases the Richardson number. This implies stronger vertical mixing due to the Langmuir entrainment by the tropical cyclone. Hence the SST cooling in the near wake region of the tropical cyclone is stronger when LF17 is used than VR12-MA. This shows parameterizing the Langmuir turbulence using LF17 gives more realistic results than VR12-MA.